# Implicit Bias of Gradient Descent on Linear Convolutional Networks

**Suriya Gunasekar**
TTI at Chicago, USA
suriya@ttic.edu

**Jason D. Lee**
USC Los Angeles, USA
jasonlee@marshall.usc.edu

**Daniel Soudry**
Technion, Israel
daniel.soudry@gmail.com

**Nathan Srebro**
TTI at Chicago, USA
nati@ttic.edu

## Abstract

We show that gradient descent on full width linear convolutional networks of depth $L$ converges to a linear predictor related to the $\ell_{2/L}$ bridge penalty in the frequency domain. This is in contrast to fully connected linear networks, where regardless of depth, gradient descent converges to the $\ell_2$ maximum margin solution.

## 1 Introduction

Implicit biases introduced by optimization algorithms play an crucial role in learning deep neural networks [Neyshabur et al., 2015b,a, Hochreiter and Schmidhuber, 1997, Keskar et al., 2016, Chaudhari et al., 2016, Dinh et al., 2017, Andrychowicz et al., 2016, Neyshabur et al., 2017, Zhang et al., 2017, Wilson et al., 2017, Hoffer et al., 2017, Smith, 2018]. Large scale neural networks used in practice are highly over-parameterized with far more trainable model parameters compared to the number of training examples. Consequently, optimization objectives for learning such high capacity models have many global minima that fit training data perfectly. However, minimizing the training loss using specific optimization algorithms take us to not just any global minima, but some special global minima, *e.g.,* global minima minimizing some regularizer $\mathcal{R}(\boldsymbol{\beta})$. In over-parameterized models, specially deep neural networks, much, if not most, of the inductive bias of the learned model comes from this implicit regularization from the optimization algorithm. Understanding the implicit bias, *e.g.,* via characterizing $\mathcal{R}(\boldsymbol{\beta})$, is thus essential for understanding how and what the model learns.

For example, in linear regression we understand how minimizing an under-determined model (with more parameters than samples) using gradient descent yields the minimum $\ell_2$ norm solution, and for linear logistic regression trained on linearly separable data, Soudry et al. [2017] recently showed that gradient descent converges in the direction of the hard margin support vector machine solution, even though the norm or margin is not explicitly specified in the optimization problem. Such minimum norm or maximum margin solutions are of course very special among all solutions or separators that fit the training data, and in particular can ensure generalization Bartlett and Mendelson [2003], Kakade et al. [2009].

Changing the optimization algorithm, even without changing the model, changes this implicit bias, and consequently also changes generalization properties of the learned models [Neyshabur et al., 2015a, Keskar et al., 2016, Wilson et al., 2017, Gunasekar et al., 2017, 2018]. For example, for linear logistic regression, using coordinate descent instead of gradient descent return a maximum $\ell_1$ margin solution instead of the hard margin support vector solution solution—an entirely different inductive bias Telgarsky [2013], Gunasekar et al. [2018].

Similarly, and as we shall see in this paper, changing to a different parameterization of the same model class can also dramatically change the implicit bias Gunasekar et al. [2017]. In particular, we study the implicit bias of optimizing multi-layer fully connected linear networks, and linear convolutional networks (multiple full width convolutional layers followed by a single fully connected layer) using gradient descent. Both of these types of models ultimately implement linear transformations, and can implement any linear transformation. The model class defined by these networks is thus simply the class of all linear predictors, and these models can be seen as mere (over) parameterizations of the class of linear predictors. Minimizing the training loss on these models is therefore entirely equivalent to minimizing the training loss for linear classification. Nevertheless, as we shall see, optimizing these networks with gradient descent leads to very different solutions.

In particular, we show that for fully connected networks with single output, optimizing the exponential loss over linearly separable data using gradient loss again converges to the homogeneous hard margin support vector machine solution. This holds regardless of the depth of the network, and hence, at least with a single output, gradient descent on fully connected networks has the same implicit bias as direct gradient descent on the parameters of the linear predictor. In contrast, training a linear convolutional network with gradient descent biases us toward linear separators that are sparse in the frequency domain. Furthermore, this bias changes with the depth of the network, and a network of depth $L$ (with $L - 1$ convolutional layers), implicitly biases towards minimizing the $\|\widehat{\boldsymbol{\beta}}\|_{2/L}$ bridge penalty with $2/L \leq 1$ of the Fourier transform $\widehat{\boldsymbol{\beta}}$ of the learned linear predictor $\boldsymbol{\beta}$ subject to margin constraints (the gradient descent predictor reaches a stationary point of the $\|\widehat{\boldsymbol{\beta}}\|_{2/L}$ minimization problem). This is a sparsity inducing regularizer, which induces sparsity more aggressively as the depth increases.

Finally, in this paper we focus on characterizing *which* global minimum does gradient descent on over-parameterized linear models converge to, while assuming that for appropriate choice of step sizes gradient descent iterates asymptotically minimize the optimization objective. A related challenge in neural networks, not addressed in this paper, is an answer to *when* does gradient descent minimize the non-convex empirical loss objective to reach *a* global minimum. This problem while hard in worst case, has been studied for linear networks. Recent work have concluded that with sufficient over-parameterization (as is the case with our settings), loss landscape of linear models are well behaved and all local minima are global minima making the problem tractable Burer and Monteiro [2003], Journée et al. [2010], Kawaguchi [2016], Nguyen and Hein [2017], Lee et al. [2016].

**Notation** We typeface vectors with bold characters *e.g.,* $\mathbf{w}, \boldsymbol{\beta}, \mathbf{x}$. Individual entries of a vector $\mathbf{z} \in \mathbb{R}^D$ are indexed using $0$ based indexing as $\mathbf{z}[d]$ for $d = 0, 1, \ldots, D - 1$. Complex numbers are represented in the polar form as $z = |z|\mathrm{e}^{\mathrm{i}\phi_z}$ with $|z| \in \mathbb{R}_+$ denoting the magnitude of $z$ and $\phi_z \in [0, 2\pi)$ denoting the phase. $z^* = |z|\mathrm{e}^{-\mathrm{i}\phi_z}$ denotes the complex conjugate of $z$. The complex inner product between $\mathbf{z}, \boldsymbol{\beta} \in \mathbb{C}^D$ is given by $\langle \mathbf{z}, \boldsymbol{\beta} \rangle = \sum_{d=1}^{D} \mathbf{z}[d]\boldsymbol{\beta}^*[d] = \mathbf{z}^\top \boldsymbol{\beta}^*$. The $D^{\text{th}}$ complex root of $1$ is denoted by $\omega_D = \mathrm{e}^{-\frac{2\pi\mathrm{i}}{D}}$. For $\mathbf{z} \in \mathbb{R}^D$ we use the notation $\widehat{\mathbf{z}} \in \mathbb{C}^D$ to denote the representation of $\mathbf{z}$ in the discrete Fourier basis given by, $\widehat{\mathbf{z}}[d] = \frac{1}{\sqrt{D}}\sum_{p=0}^{D-1} \mathbf{z}[p]\omega_D^{pd}$. For integers $D$ and $a$, we denote the modulo operator as $a \bmod D = a - D\lfloor \frac{a}{D} \rfloor$. Finally, for multi-layer linear networks (formally defined in Section 2), we will use $\mathbf{w} \in \mathcal{W}$ to denote parameters of the model in general domain $\mathcal{W}$, and $\boldsymbol{\beta}_{\mathbf{w}}$ or simply $\boldsymbol{\beta}$ to denote the equivalent linear predictor.

## 2 Multi-layer Linear Networks

We consider feed forward linear networks that map input features $\mathbf{x} \in \mathbb{R}^D$ to a single real valued output $f_{\mathbf{w}}(\mathbf{x}) \in \mathbb{R}$, where $\mathbf{w}$ denote the parameters of the network. Such networks can be thought of as directed acyclic graphs where each edge is associated with a weight, and the value at each node/unit is the weighted sum of values from the parent nodes. The input features form source nodes with no incoming edges and the output is a sink node with no outgoing edge. Every such network realizes a linear function $\mathbf{x} \rightarrow \langle \mathbf{x}, \boldsymbol{\beta}_{\mathbf{w}} \rangle$, where $\boldsymbol{\beta}_{\mathbf{w}} \in \mathbb{R}^D$ denotes the effective linear predictor.

In multi-layer networks, the nodes are arranged in layers, so an $L$–layer network represents a composition of $L$ linear maps. We use the convention that, the input $\mathbf{x} \in \mathbb{R}^D$ is indexed as the zeroth layer $l = 0$, while the output forms the final layer with $l = L$. The outputs of nodes in layer $l$ are denoted by $\mathbf{h}_l \in \mathbb{R}^{D_l}$, where $D_l$ is the number of nodes in layer $l$. We also use $\mathbf{w}_l$ to denote the

parameters of the linear map between $\mathbf{h}_{l-1}$ and $\mathbf{h}_l$, and $\mathbf{w} = [\mathbf{w}_l]_{l=1}^L$ to denote the collective set of all parameters of the linear network.

**Linear fully connected network**  In a fully connected linear network, the nodes between successive layers $l-1$ and $l$ are densely connected with edge weights $\mathbf{w}_l \in \mathbb{R}^{D_{l-1} \times D_l}$, and all the weights are independent parameters. This model class is parameterized by $\mathbf{w} = [\mathbf{w}_l]_{l=1}^L \in \prod_{l=1}^L \mathbb{R}^{D_{l-1} \times D_l}$ and the computation for intermediate nodes $\mathbf{h}_l$ and the composite linear map $f_{\mathbf{w}}(\mathbf{x})$ is given by,

$$\mathbf{h}_l = \mathbf{w}_l^\top \mathbf{h}_{l-1} \quad \text{and} \quad f_{\mathbf{w}}(\mathbf{x}) = \mathbf{h}_L = \mathbf{w}_L^\top \mathbf{w}_{L-1}^\top \ldots \mathbf{w}_1^\top \mathbf{x}. \tag{1}$$

**Linear convolutional network**  We consider one-dimensional convolutional network architectures where each non-output layer has exactly $D$ units (same as the input dimensionality) and the linear transformations from layer $l-1$ to layer $l$ are given by the following circular convolutional operation[1] parameterized by full width filters with weights $[\mathbf{w}_l \in \mathbb{R}^D]_{l=1}^{L-1}$. For $l = 1, 2, \ldots, L-1$,

$$\mathbf{h}_l[d] = \frac{1}{\sqrt{D}} \sum_{k=0}^{D-1} \mathbf{w}_l[k]\, \mathbf{h}_{l-1}\left[(d+k) \bmod D\right] := (\mathbf{h}_{l-1} \star \mathbf{w}_l)\,[d]. \tag{2}$$

The output layer is fully connected and parameterized by weights $\mathbf{w}_L \in \mathbb{R}^D$. The parameters of the model class therefor consists of $L$ vectors of size $D$ collectively denoted by $\mathbf{w} = [\mathbf{w}_l]_{l=1}^L \in \prod_{l=1}^L \mathbb{R}^D$, and the composite linear map $f_{\mathbf{w}}(\mathbf{x})$ is given by:

$$f_{\mathbf{w}}(\mathbf{x}) = \left(\left(\left((\mathbf{x} \star \mathbf{w}_1) \star \mathbf{w}_2\right)\ldots\right) \star \mathbf{w}_{L-1}\right)^\top \mathbf{w}_L. \tag{3}$$

*Remark:* We use circular convolution with a scaling of $1/\sqrt{D}$ to make the analysis cleaner. For convolutions with zero-padding, we expect a similar behavior. Secondly, since our goal here to study implicit bias in sufficiently over-parameterized models, we only study full dimensional convolutional filters. In practice it is common to have filters of width $K$ smaller than the number of input features, which can change the implicit bias.

The fully connected and convolutional linear networks described above can both be represented in terms of a mapping $\mathcal{P} : \mathcal{W} \to \mathbb{R}^D$ that maps the input parameters $\mathbf{w} \in \mathcal{W}$ to a linear predictor in $\mathbb{R}^D$, such that the output of the network is given by $f_{\mathbf{w}}(\mathbf{x}) = \langle \mathbf{x}, \mathcal{P}(\mathbf{w}) \rangle$. For fully connected networks, the mapping is given by $\mathcal{P}_{full}(\mathbf{w}) = \mathbf{w}_1 \mathbf{w}_2 \ldots \mathbf{w}_L$, and for convolutional networks, $\mathcal{P}_{conv}(\mathbf{w}) = \left(\left((\mathbf{w}_L^\downarrow \star \mathbf{w}_{L-1}) \star \mathbf{w}_{L-2}\right)\ldots \star \mathbf{w}_1\right)^\downarrow$, where $\mathbf{w}^\downarrow$ denotes the flipped vector corresponding to $\mathbf{w}$, given by $\mathbf{w}^\downarrow[k] = \mathbf{w}[D - k - 1]$ for $k = 0, 1, \ldots, D-1$.

**Separable linear classification**  Consider a binary classification dataset $\{(\mathbf{x}_n, y_n) : n = 1, 2, \ldots N\}$ with $\mathbf{x}_n \in \mathbb{R}^D$ and $y_n \in \{-1, 1\}$. The empirical risk minimization objective for training a linear network parameterized as $\mathcal{P}(\mathbf{w})$ is given as follows,

$$\min_{\mathbf{w} \in \mathcal{W}} \mathcal{L}_{\mathcal{P}}(\mathbf{w}) := \sum_{n=1}^N \ell(\langle \mathbf{x}_n, \mathcal{P}(\mathbf{w}) \rangle, y_n), \tag{4}$$

where $\ell : \mathbb{R} \times \{-1, 1\} \to \mathbb{R}_+$ is some surrogate loss for classification accuracy, *e.g.,* logistic loss $\ell(\widehat{y}, y) = \log(1 + \exp(-\widehat{y}y))$ and exponential loss $\ell(\widehat{y}, y) = \exp(-\widehat{y}y)$.

It is easy to see that both fully connected and convolutional networks of any depth $L$ can realize any linear predictor $\boldsymbol{\beta} \in \mathbb{R}^D$. The model class expressed by both networks is therefore simply the unconstrained class of linear predictors, and the two architectures are merely different (over) parameterizations of this class

$$\{\mathcal{P}_{full}(\mathbf{w}) : \mathbf{w} = [\mathbf{w}_l \in \mathbb{R}^{D_{l-1} \times D_l}]_{l=1}^L\} = \{\mathcal{P}_{conv}(\mathbf{w}) : \mathbf{w} = [\mathbf{w}_l \in \mathbb{R}^D]_{l=1}^L\} = \mathbb{R}^D.$$

Thus, the empirical risk minimization problem in (4) is equivalent to the following optimization over the linear predictors $\boldsymbol{\beta} = \mathcal{P}(\mathbf{w})$:

$$\min_{\boldsymbol{\beta} \in \mathbb{R}^D} \mathcal{L}(\boldsymbol{\beta}) := \sum_{n=1}^N \ell(\langle \mathbf{x}_n, \boldsymbol{\beta} \rangle, y_n). \tag{5}$$

Although the optimization problems (4) and (5) are exactly equivalent in terms of the set of global minima, in this paper, we show that optimizing (4) with different parameterizations leads to very different classifiers compared to optimizing (5) directly.

In particular, consider problems (4)/(5) on a linearly separable dataset $\{\mathbf{x}_n, y_n\}_{n=1}^N$ and using the logistic loss (the two class version of the cross entropy loss typically used in deep learning). The global infimum of $\mathcal{L}(\boldsymbol{\beta})$ is 0, but this is not attainable by any finite $\boldsymbol{\beta}$. Instead, the loss can be minimized by scaling the norm of any linear predictor that separates the data to infinity. Thus, any sequence of predictors $\boldsymbol{\beta}^{(t)}$ (say, from an optimization algorithm) that asymptotically minimizes the loss in eq. (5) necessarily separates the data and diverges in norm, $\|\boldsymbol{\beta}^{(t)}\| \to \infty$. In general there are many linear separators that correctly label the training data, each corresponding to a direction in which we can minimize (5). Which of these separators will we converge to when optimizing (4)/(5)? In other words, what is the direction $\overline{\boldsymbol{\beta}}^\infty = \lim_{t\to\infty} \frac{\boldsymbol{\beta}^{(t)}}{\|\boldsymbol{\beta}^{(t)}\|}$ the iterates of our optimization algorithm will diverge in? If this limit exist we say that $\boldsymbol{\beta}^{(t)}$ *converges in direction* to the *limit direction* $\overline{\boldsymbol{\beta}}^\infty$.

Soudry et al. [2017] studied this implicit bias of gradient descent on (5) over the direct parameterization of $\boldsymbol{\beta}$. They showed that for any linearly separable dataset and any initialization, gradient descent w.r.t. $\boldsymbol{\beta}$ converges in direction to hard margin support vector machine solution:

$$\overline{\boldsymbol{\beta}}^\infty = \frac{\boldsymbol{\beta}_{\ell_2}^*}{\|\boldsymbol{\beta}_{\ell_2}^*\|}, \text{ where } \boldsymbol{\beta}_{\ell_2}^* = \underset{\boldsymbol{\beta}\in\mathbb{R}^D}{\operatorname{argmin}}\|\boldsymbol{\beta}\|_2^2 \text{ s.t. } \forall n, y_n\langle\boldsymbol{\beta}, \mathbf{x}_n\rangle \geq 1. \tag{6}$$

In this paper we study the behavior of gradient descent on the problem (4) w.r.t different parameterizations of the model class of linear predictors. For initialization $\mathbf{w}^{(0)}$ and sequence of step sizes $\{\eta_t\}$, gradient descent updates for (4) are given by,

$$\mathbf{w}^{(t+1)} = \mathbf{w}^{(t)} - \eta_t\nabla_{\mathbf{w}}\mathcal{L}_{\mathcal{P}}(\mathbf{w}^{(t)}) = \mathbf{w}^{(t)} - \eta_t\nabla_{\mathbf{w}}\mathcal{P}(\mathbf{w}^{(t)})\nabla_{\boldsymbol{\beta}}\mathcal{L}(\mathcal{P}(\mathbf{w}(t))), \tag{7}$$

where $\nabla_{\mathbf{w}}\mathcal{P}(.)$ denotes the Jacobian of $\mathcal{P}: \mathcal{W} \to \mathbb{R}^D$ with respect to the parameters $\mathbf{w}$, and $\nabla_{\boldsymbol{\beta}}\mathcal{L}(.)$ is the gradient of the loss function in (5).

For separable datasets, if $\mathbf{w}^{(t)}$ minimizes (4) for linear fully connected or convolutional networks, then we will again have $\|\mathbf{w}^{(t)}\| \to \infty$, and the question we ask is: what is the limit direction $\overline{\boldsymbol{\beta}}^\infty = \lim_{t\to\infty} \frac{\mathcal{P}(\mathbf{w}^{(t)})}{\|\mathcal{P}(\mathbf{w}^{(t)})\|}$ of the predictors $\mathcal{P}(\mathbf{w}^{(t)})$ along the optimization path?

The result in Soudry et al. [2017] holds for any loss function $\ell(u, y)$ that is strictly monotone in $uy$ with specific tail behavior, name the tightly exponential tail, which is satisfied by popular classification losses like logistic and exponential loss. In the rest of the paper, for simplicity we exclusively focus on the exponential loss function $\ell(u, y) = \exp(-uy)$, which has the same tail behavior as that of the logistic loss. Along the lines of Soudry et al. [2017], our results should also extend for any strictly monotonic loss function with a tight exponential tail, including logistic loss.

## 3   Main Results

Our main results characterize the implicit bias of gradient descent for multi-layer fully connected and convolutional networks with linear activations. For the gradient descent iterates $\mathbf{w}^{(t)}$ in eq. (7), we henceforth denote the induced linear predictor as $\boldsymbol{\beta}^{(t)} = \mathcal{P}(\mathbf{w}^{(t)})$.

**Assumptions.** *In the following theorems, we characterize the limiting predictor* $\overline{\boldsymbol{\beta}}^\infty = \lim_{t\to\infty} \frac{\boldsymbol{\beta}^{(t)}}{\|\boldsymbol{\beta}^{(t)}\|}$ *under the following assumptions:*

1. $\mathbf{w}^{(t)}$ *minimize the objective, i.e.,* $\mathcal{L}_{\mathcal{P}}(\mathbf{w}^{(t)}) \to 0$.
2. $\mathbf{w}^{(t)}$, *and consequently* $\boldsymbol{\beta}^{(t)} = \mathcal{P}(\mathbf{w}^{(t)})$, *converge in direction to yield a separator* $\overline{\boldsymbol{\beta}}^\infty = \lim_{t\to\infty} \frac{\boldsymbol{\beta}^{(t)}}{\|\boldsymbol{\beta}^{(t)}\|}$ *with positive margin, i.e.,* $\min_n y_n\langle\mathbf{x}_n, \overline{\boldsymbol{\beta}}^\infty\rangle > 0$.
3. *Gradients with respect to linear predictors* $\nabla_{\boldsymbol{\beta}}\mathcal{L}(\boldsymbol{\beta}^{(t)})$ *converge in direction.*

These assumptions allow us to focus on the question of *which specific linear predictor do gradient descent iterates converge to* by separating it from the related optimization questions of when gradient descent iterates minimize the non-convex objective in eq. (5) and nicely converge in direction.

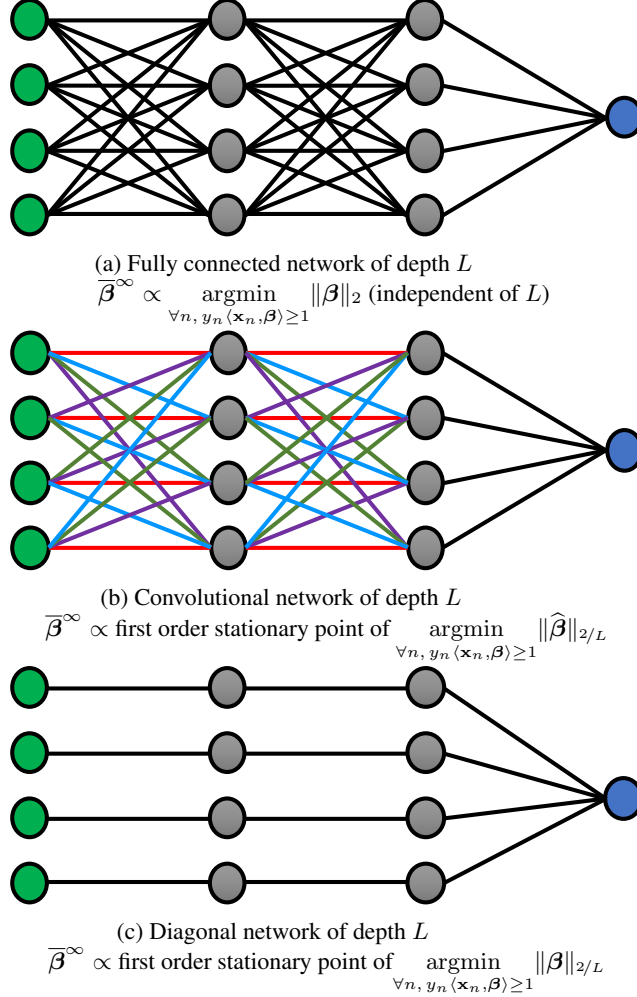

(a) Fully connected network of depth $L$

$$\overline{\boldsymbol{\beta}}^{\infty} \propto \underset{\forall n,\, y_n \langle \mathbf{x}_n, \boldsymbol{\beta} \rangle \geq 1}{\operatorname{argmin}} \|\boldsymbol{\beta}\|_2 \text{ (independent of } L)$$

(b) Convolutional network of depth $L$

$$\overline{\boldsymbol{\beta}}^{\infty} \propto \text{first order stationary point of } \underset{\forall n,\, y_n \langle \mathbf{x}_n, \boldsymbol{\beta} \rangle \geq 1}{\operatorname{argmin}} \|\widehat{\boldsymbol{\beta}}\|_{2/L}$$

(c) Diagonal network of depth $L$

$$\overline{\boldsymbol{\beta}}^{\infty} \propto \text{first order stationary point of } \underset{\forall n,\, y_n \langle \mathbf{x}_n, \boldsymbol{\beta} \rangle \geq 1}{\operatorname{argmin}} \|\boldsymbol{\beta}\|_{2/L}$$

Figure 1: Implicit bias of gradient descent for different linear network architectures.

**Theorem 1** (Linear fully connected networks). *For any depth $L$, almost all linearly separable datasets $\{\mathbf{x}_n, y_n\}_{n=1}^N$, almost all initializations $\mathbf{w}^{(0)}$, and any bounded sequence of step sizes $\{\eta_t\}_t$, consider the sequence gradient descent iterates $\mathbf{w}^{(t)}$ in eq. (7) for minimizing $\mathcal{L}_{\mathcal{P}_{full}}(\mathbf{w})$ in eq. (4) with exponential loss $\ell(\widehat{y}, y) = \exp(-\widehat{y}y)$ over $L$–layer fully connected linear networks.*

*If (a) the iterates $\mathbf{w}^{(t)}$ minimize the objective, i.e., $\mathcal{L}_{\mathcal{P}_{full}}(\mathbf{w}^{(t)}) \to 0$, (b) $\mathbf{w}^{(t)}$, and consequently $\boldsymbol{\beta}^{(t)} = \mathcal{P}_{full}(\mathbf{w}^{(t)})$, converge in direction to yield a separator with positive margin, and (c) gradients with respect to linear predictors $\nabla_{\boldsymbol{\beta}} \mathcal{L}(\boldsymbol{\beta}^{(t)})$ converge in direction, then the limit direction is given by,*

$$\overline{\boldsymbol{\beta}}^{\infty} = \lim_{t \to \infty} \frac{\mathcal{P}_{full}(\mathbf{w}^{(t)})}{\|\mathcal{P}_{full}(\mathbf{w}^{(t)})\|} = \frac{\boldsymbol{\beta}_{\ell_2}^*}{\|\boldsymbol{\beta}_{\ell_2}^*\|}, \text{where } \boldsymbol{\beta}_{\ell_2}^* := \underset{w}{\operatorname{argmin}} \|\boldsymbol{\beta}\|_2^2 \ s.t. \ \forall n, y_n \langle \mathbf{x}_n, \boldsymbol{\beta} \rangle \geq 1. \quad (8)$$

For fully connected networks with single output, Theorem 1 shows that there is no effect of depth on the implicit bias of gradient descent. Regardless of the depth of the network, the asymptotic classifier is always the hard margin support vector machine classifier, which is also the limit direction of gradient descent for linear logistic regression with the direct parameterization of $\boldsymbol{\beta} = \mathbf{w}$.

In contrast, next we show that for convolutional networks we get very different biases. Let us first look at a 2–layer linear convolutional network, *i.e.,* a network with single convolutional layer followed by a fully connected final layer.

Recall that $\widehat{\boldsymbol{\beta}} \in \mathbf{C}^D$ denote the Fourier coefficients of $\boldsymbol{\beta}$, i.e., $\widehat{\boldsymbol{\beta}}[d] = \frac{1}{\sqrt{D}} \sum_{p=0}^{D-1} \boldsymbol{\beta}[p] \exp\left(-\frac{2\pi \mathrm{i} p d}{D}\right)$, and that any non-zero $z \in \mathbf{C}$ is denoted in polar form as $z = |z| \mathrm{e}^{\mathrm{i}\phi_z}$ for $\phi_z \in [0, 2\pi)$. Linear predictors induced by gradient descent iterates $\mathbf{w}^{(t)}$ for convolutional networks are denoted by $\boldsymbol{\beta}^{(t)} = \mathcal{P}_{conv}(\mathbf{w}^{(t)})$. It is evident that if $\boldsymbol{\beta}^{(t)}$ converges in direction to $\overline{\boldsymbol{\beta}}^\infty$, then its Fourier transformation $\widehat{\boldsymbol{\beta}}^{(t)}$ converges in direction to $\overline{\widehat{\boldsymbol{\beta}}}^\infty$. In the following theorems, in addition to the earlier assumptions, we further assume a technical condition that the phase of the Fourier coefficients $\mathrm{e}^{\mathrm{i}\phi_{\widehat{\boldsymbol{\beta}}^{(t)}}}$ converge coordinate-wise. For coordinates $d$ with $\overline{\widehat{\boldsymbol{\beta}}}^\infty[d] \neq 0$ this follows from convergence in direction of $\mathbf{w}^{(t)}$, in which case $\mathrm{e}^{\mathrm{i}\phi_{\widehat{\boldsymbol{\beta}}^{(t)}[d]}} \to \mathrm{e}^{\mathrm{i}\phi_{\overline{\widehat{\boldsymbol{\beta}}}^\infty[d]}}$. We assume such a $\phi_{\overline{\widehat{\boldsymbol{\beta}}}^\infty[d]}$ also exists when $\overline{\widehat{\boldsymbol{\beta}}}^\infty[d] = 0$.

**Theorem 2** (Linear convolutional networks of depth two). *For almost all linearly separable datasets $\{\mathbf{x}_n, y_n\}_{n=1}^N$, almost all initializations $\mathbf{w}^{(0)}$, and any sequence of step sizes $\{\eta_t\}_t$ with $\eta_t$ smaller than the local Lipschitz at $\mathbf{w}^{(t)}$, consider the sequence gradient descent iterates $\mathbf{w}^{(t)}$ in eq. (7) for minimizing $\mathcal{L}_{\mathcal{P}_{conv}}(\mathbf{w})$ in eq. (4) with exponential loss over 2–layer linear convolutional networks.*

*If (a) the iterates $\mathbf{w}^{(t)}$ minimize the objective, i.e., $\mathcal{L}_{\mathcal{P}_{conv}}(\mathbf{w}^{(t)}) \to 0$, (b) $\mathbf{w}^{(t)}$ converge in direction to yield a separator $\overline{\boldsymbol{\beta}}^\infty$ with positive margin, (c) the phase of the Fourier coefficients $\widehat{\boldsymbol{\beta}}^{(t)}$ of the linear predictors $\boldsymbol{\beta}^{(t)}$ converge coordinate-wise, i.e., $\forall d$, $\mathrm{e}^{\mathrm{i}\phi_{\widehat{\boldsymbol{\beta}}^{(t)}[d]}} \to \mathrm{e}^{\mathrm{i}\phi_{\widehat{\boldsymbol{\beta}}^\infty[d]}}$, and (d) the gradients $\nabla_{\boldsymbol{\beta}} \mathcal{L}(\boldsymbol{\beta}^{(t)})$ converge in direction, then the limit direction $\overline{\boldsymbol{\beta}}^\infty$ is given by,*

$$\overline{\boldsymbol{\beta}}^\infty = \frac{\boldsymbol{\beta}^*_{\mathcal{F},1}}{\|\boldsymbol{\beta}^*_{\mathcal{F},1}\|}, \text{ where } \boldsymbol{\beta}^*_{\mathcal{F},1} := \underset{\boldsymbol{\beta}}{\mathrm{argmin}} \|\widehat{\boldsymbol{\beta}}\|_1 \text{ s.t. } \forall n, \ y_n \langle \boldsymbol{\beta}, \mathbf{x}_n \rangle \geq 1. \tag{9}$$

We already see how introducing a single convolutional layer changes the implicit bias of gradient descent—even without any explicit regularization, gradient descent on the parameters of convolutional network architecture returns solutions that are biased to have sparsity in the frequency domain.

Furthermore, unlike fully connected networks, for convolutional networks we also see that the implicit bias changes with the depth of the network as shown by the following theorem.

**Theorem 2a** (Linear Convolutional Networks of any Depth). *For any depth L, under the conditions of Theorem 2, the limit direction $\overline{\boldsymbol{\beta}}^\infty = \lim_{t \to \infty} \frac{\mathcal{P}_{conv}(\mathbf{w}^{(t)})}{\|\mathcal{P}_{conv}(\mathbf{w}^{(t)})\|}$ is a scaling of a first order stationary point of the following optimization problem,*

$$\min_{\boldsymbol{\beta}} \|\widehat{\boldsymbol{\beta}}\|_{2/L} \text{ s.t. } \forall n, \ y_n \langle \boldsymbol{\beta}, \mathbf{x}_n \rangle \geq 1, \tag{10}$$

*where the $\ell_p$ penalty given by $\|z\|_p = \left(\sum_{i=1}^D |z[i]|^p\right)^{1/p}$ (also called the bridge penalty) is a norm for $p = 1$ and a quasi-norm for $p < 1$.*

When $L > 2$, and thus $p = 2/L < 1$, problem (10) is non-convex and intractable Ge et al. [2011]. Hence, we cannot expect to ensure convergence to a global minimum. Instead we show convergence to a first order stationary point of (10) in the sense of sub-stationary points of Rockafellar [1979] for optimization problems with non-smooth and non-convex objectives. These are solutions where the *local directional derivative* along the directions in the tangent cone of the constraints are all zero.

The first order stationary points, or sub-stationary points, of (10) are the set of feasible predictors $\boldsymbol{\beta}$ such that $\exists \{\alpha_n \geq 0\}_{n=1}^N$ satisfying the following: $\forall n, \ y_n \langle \mathbf{x}_n, \boldsymbol{\beta} \rangle > 1 \implies \alpha_n = 0$, and

$$\sum_n \alpha_n y_n \widehat{\mathbf{x}}_n \in \partial^\circ \|\widehat{\boldsymbol{\beta}}\|_p, \tag{11}$$

where $\widehat{\mathbf{x}}_n$ is the Fourier transformation of $\mathbf{x}_n$, and $\partial^\circ$ denotes the local sub-differential (or Clarke's sub-differential) operator defined as $\partial^\circ f(\boldsymbol{\beta}) = \mathrm{conv}\{\mathbf{v} : \exists (\mathbf{z}_k)_k \text{ s.t. } \mathbf{z}_k \to \boldsymbol{\beta} \text{ and } \nabla f(\mathbf{z}_k) \to \mathbf{v}\}$.

For $p = 1$ and $\widehat{\boldsymbol{\beta}}$ represented in polar form as $\widehat{\boldsymbol{\beta}} = |\widehat{\boldsymbol{\beta}}| \mathrm{e}^{\mathrm{i}\phi_{\widehat{\boldsymbol{\beta}}}} \in \mathbb{C}^D$, $\|\widehat{\boldsymbol{\beta}}\|_p$ is convex and the local sub-differential is indeed the global sub-differential given by,

$$\partial^\circ \|\widehat{\boldsymbol{\beta}}\|_1 = \{\widehat{\mathbf{z}} : \forall d, \ |\widehat{\mathbf{z}}[d]| \leq 1 \text{ and } \widehat{\boldsymbol{\beta}}[d] \neq 0 \implies \widehat{\mathbf{z}}[d] = \mathrm{e}^{\mathrm{i}\phi_{\widehat{\boldsymbol{\beta}}}[d]}\}. \tag{12}$$

For $p < 1$, the local sub-differential of $\|\widehat{\boldsymbol{\beta}}\|_p$ is given by,

$$\forall p < 1, \quad \partial^\circ \|\widehat{\boldsymbol{\beta}}\|_p = \{\widehat{\mathbf{z}} : \widehat{\boldsymbol{\beta}}[d] \neq 0 \implies \widehat{\mathbf{z}}[d] = p\, e^{\mathrm{i}\phi_{\widehat{\beta}}[d]}\, |\widehat{\boldsymbol{\beta}}[d]|^{p-1}\}. \tag{13}$$

Figures 1a–1b summarize the implications of the main results in the paper. The proof of this Theorem, exploits the following representation of $\mathcal{P}_{conv}(\boldsymbol{\beta})$ in the Fourier domain.

**Lemma 3.** *For full-dimensional convolutions, $\boldsymbol{\beta} = \mathcal{P}_{conv}(\mathbf{w})$ is equivalent to*

$$\widehat{\boldsymbol{\beta}} = diag(\widehat{\mathbf{w}}_1)\dots diag(\widehat{\mathbf{w}}_{L-1})\widehat{\mathbf{w}}_L,$$

*where for $l = 1, 2, \dots, L$, $\widehat{\mathbf{w}}_1 \in \mathbf{C}^D$ are the Fourier coefficients of the parameters $\mathbf{w}_l \in \mathbb{R}^D$.*

From above lemma (proved in Appendix $C$), we can see a connection of convolutional networks to a special network where the linear transformation between layers is restricted to diagonal entries (see depiction in Figure 1c), we refer to such networks as *linear diagonal network*.

The proof of Theorem 1 and Theorem 2-2a are provided in Appendix $B$ and $C$, respectively.

## 4 Understanding Gradient Descent in the Parameter Space

We can decompose the characterization of implicit bias of gradient descent on a parameterization $\mathcal{P}(\mathbf{w})$ into two parts: (a) what is the implicit bias of gradient descent in the space of parameters $\mathbf{w}$?, and (b) what does this imply in term of the linear predictor $\boldsymbol{\beta} = \mathcal{P}(\mathbf{w})$, *i.e.,* how does the bias in parameter space translate to the linear predictor learned from the model class?

We look at the first question for a broad class of linear models, where the linear predictor is given by a homogeneous polynomial mapping of the parameters: $\boldsymbol{\beta} = \mathcal{P}(\mathbf{w})$, where $\mathbf{w} \in \mathbb{R}^P$ are the parameters of the model and $\mathcal{P} : \mathbb{R}^P \to \mathbb{R}^D$ satisfies definition below. This class covers the linear convolutional, fully connected networks, and diagonal networks discussed in Section 3.

**Definition** (Homogeneous Polynomial). *A multivariate polynomial function $\mathcal{P} : \mathbb{R}^P \to \mathbb{R}^D$ is said to be homogeneous, if for some finite integer $\nu < \infty$, $\forall \alpha \in \mathbb{R}, \mathbf{v} \in \mathbb{R}^P$, $\mathcal{P}(\alpha\mathbf{v}) = \alpha^\nu \mathcal{P}(\mathbf{v})$.*

**Theorem 4** (Homogeneous Polynomial Parameterization). *For any homogeneous polynomial map $\mathcal{P} : \mathbb{R}^P \to \mathbb{R}^D$ from parameters $\mathbf{w} \in \mathbb{R}^D$ to linear predictors, almost all datasets $\{\mathbf{x}_n, y_n\}_{n=1}^N$ separable by $\mathcal{B} := \{\mathcal{P}(\mathbf{w}) : \mathbf{w} \in \mathbb{R}^P\}$, almost all initializations $\mathbf{w}^{(0)}$, and any bounded sequence of step sizes $\{\eta_t\}_t$, consider the sequence of gradient descent updates $\mathbf{w}^{(t)}$ from eq. (7) for minimizing the empirical risk objective $\mathcal{L}_\mathcal{P}(\mathbf{w})$ in (4) with exponential loss $\ell(u, y) = \exp(-uy)$.*

*If (a) the iterates $\mathbf{w}^{(t)}$ asymptotically minimize the objective, i.e., $\mathcal{L}_\mathcal{P}(\mathbf{w}^{(t)}) = \mathcal{L}(\mathcal{P}(\mathbf{w}^{(t)})) \to 0$, (b) $\mathbf{w}^{(t)}$, and consequently $\boldsymbol{\beta}^{(t)} = \mathcal{P}(\mathbf{w}^{(t)})$, converge in direction to yield a separator with positive margin, and (c) the gradients w.r.t. to the linear predictors, $\nabla_{\boldsymbol{\beta}}\mathcal{L}(\boldsymbol{\beta}^{(t)})$ converge in direction, then the limit direction of the parameters $\overline{\mathbf{w}}^\infty = \lim_{t\to\infty} \frac{\mathbf{w}^{(t)}}{\|\mathbf{w}^{(t)}\|_2}$ is a positive scaling of a first order stationary point of the following optimization problem,*

$$\min_{\mathbf{w}\in\mathbb{R}^P} \|\mathbf{w}\|_2^2 \quad s.t. \quad \forall n,\, y_n\langle \mathbf{x}_n, \mathcal{P}(\mathbf{w})\rangle \geq 1. \tag{14}$$

Theorem 4 is proved in Appendix $A$. The proof of Theorem 4 involves showing that the asymptotic direction of gradient descent iterates satisfies the KKT conditions for first order stationary points of (14). This crucially relies on two properties. First, the sequence of gradients $\nabla_{\boldsymbol{\beta}}\mathcal{L}(\boldsymbol{\beta}^{(t)})$ converge in direction to a positive span of support vectors of $\overline{\boldsymbol{\beta}}^\infty = \lim_{t\to\infty} \frac{\boldsymbol{\beta}^{(t)}}{\|\boldsymbol{\beta}^{(t)}\|}$ (Lemma 8 in Gunasekar et al. [2018]), and this result relies on the loss function $\ell$ being exponential tailed. Secondly, if $\mathcal{P}$ is not homogeneous, then the optimization problems $\min_\mathbf{w} \|\mathbf{w}\|_2^2$ s.t. $\forall n, \langle \mathbf{x}_n, y_n\rangle \geq \gamma$ for different values of unnormalized margin $\gamma$ are not equivalent and lead to different separators. Thus, for general non-homogeneous $\mathcal{P}$, the unnormalized margin of one does not have a significance and the necessary conditions for the first order stationarity of (14) are not satisfied.

Finally, we also note that in many cases (including linear convolutional networks) the optimization problem (14) is non-convex and intractable (see *e.g.,* Ge et al. [2011]). So we cannot expect $\overline{\mathbf{w}}^\infty$ to be always be a global minimizer of eq. (14). We however suspect that it is possible to obtain a stronger result that $\overline{\mathbf{w}}^\infty$ reaches a higher order stationary point or even a local minimum of the explicitly regularized estimator in eq. (14).

**Implications of the implicit bias in predictor space** While eq. (14) characterizes the bias of gradient descent in the parameter space, what we really care about is the effective bias introduced in the space of functions learned by the network. In our case, this class of functions is the set of linear predictors $\{\boldsymbol{\beta} \in \mathbb{R}^D\}$. The $\ell_2$ norm penalized solution in eq. (14), is equivalently given by,

$$\boldsymbol{\beta}^*_{\mathcal{R}_{\mathcal{P}}} = \underset{\boldsymbol{\beta}}{\operatorname{argmin}} \, \mathcal{R}_{\mathcal{P}}(\boldsymbol{\beta}) \text{ s.t. } \forall n, \, y_n \langle \boldsymbol{\beta}, \mathbf{x}_n \rangle \geq 1, \text{ where } \mathcal{R}_{\mathcal{P}}(\boldsymbol{\beta}) = \inf_{\mathbf{w}:\mathcal{P}(\mathbf{w})=\boldsymbol{\beta}} \|\mathbf{w}\|_2^2. \qquad (15)$$

The problems in eq. (14) and eq. (15) have the same global minimizers, *i.e.,* $\mathbf{w}^*$ is global minimizer of eq. (14) if and only if $\boldsymbol{\beta}^* = \mathcal{P}(\mathbf{w}^*)$ minimizes eq. (15). However, such an equivalence does not extend to the stationary points of the two problems. Specifically, it is possible that a stationary point of eq. (14) is merely a feasible point for eq. (15) with no special significance. So instead of using Theorem 4, for the specific networks in Section 3, we directly show (in Appendix) that gradient descent updates converge in direction to a first order stationary point of the problem in eq. (15).

## 5  Understanding Gradient Descent in Predictor Space

In the previous section, we saw that the implicit bias of gradient descent on a parameterization $\mathcal{P}(\mathbf{w})$ can be described in terms of the optimization problem (14) and the implied penalty function $\mathcal{R}_{\mathcal{P}}(\boldsymbol{\beta}) = \min_{\mathbf{w}:\mathcal{P}(\mathbf{w})=\boldsymbol{\beta}} \|\mathbf{w}\|_2^2$. We now turn to studying this implied penalty $\mathcal{R}_{\mathcal{P}}(\boldsymbol{\beta})$ and obtaining explicit forms for it, which will reveal the precise form of the implicit bias in terms of the learned linear predictor. The proofs of the lemmas in this section are provided in the Appendix $D$.

**Lemma 5.** *For fully connected networks of any depth $L > 0$,*

$$\mathcal{R}_{\mathcal{P}_{full}}(\boldsymbol{\beta}) = \min_{\mathbf{w}:\mathcal{P}_{full}(\mathbf{w})=\boldsymbol{\beta}} \|\mathbf{w}\|_2^2 = L\|\boldsymbol{\beta}\|_2^{2/L} = \textit{monotone}(\|\boldsymbol{\beta}\|_2).$$

We see that $\boldsymbol{\beta}^*_{\mathcal{R}_{\mathcal{P}_{full}}} = \operatorname{argmin}_{\boldsymbol{\beta}} \mathcal{R}_{\mathcal{P}_{full}}(\boldsymbol{\beta}) \text{ s.t. } \forall n, y_n \langle \mathbf{x}_n, \boldsymbol{\beta} \rangle \geq 1$ in eq. (15) for fully connected networks is independent of the depth of the network $L$. In Theorem 1, we indeed show that gradient descent for this class of networks converges in the direction of $\boldsymbol{\beta}^*_{\mathcal{R}_{\mathcal{P}_{full}}}$.

Next, we motivate the characterization of $\mathcal{R}_{\mathcal{P}}(\boldsymbol{\beta})$ for linear convolutional networks by first looking at the special *linear diagonal network* depicted in Figure 1c. The depth–$L$ diagonal network is parameterized by $\mathbf{w} = [\mathbf{w}_l \in \mathbb{R}^D]_{l=1}^L$ and the mapping to a linear predictor is given by $\mathcal{P}_{diag}(\mathbf{w}) = \operatorname{diag}(\mathbf{w}_1)\operatorname{diag}(\mathbf{w}_2)\dots\operatorname{diag}(\mathbf{w}_{L-1})\mathbf{w}_L$.

**Lemma 6.** *For a depth–$L$ diagonal network with parameters $\mathbf{w} = [\mathbf{w}_l \in \mathbb{R}^D]_{l-1}^L$, we have*

$$\mathcal{R}_{\mathcal{P}_{diag}}(\boldsymbol{\beta}) = \min_{\mathbf{w}:\mathcal{P}_{diag}(\mathbf{w})=\boldsymbol{\beta}} \|\mathbf{w}\|_2^2 = L\|\boldsymbol{\beta}\|_{2/L}^{2/L} = \textit{monotone}(\|\boldsymbol{\beta}\|_{2/L}).$$

Finally, for full width linear convolutional networks parameterized by $\mathbf{w} = [\mathbf{w}_l \in \mathbb{R}^D]_{l=1}^L$, recall the following representation of $\boldsymbol{\beta} = \mathcal{P}_{conv}(\mathbf{w})$ in Fourier from Lemma 3.

$$\widehat{\boldsymbol{\beta}} = \operatorname{diag}(\widehat{\mathbf{w}}_1)\dots\operatorname{diag}(\widehat{\mathbf{w}}_{L-1})\widehat{\mathbf{w}}_L,$$

where $\widehat{\boldsymbol{\beta}}, \widehat{\mathbf{w}}_l \in \mathbf{C}^D$ are Fourier basis representation of $\boldsymbol{\beta}, \mathbf{w}_l \in \mathbb{R}^D$, respectively. Extending the result of diagonal networks for the complex vector spaces, we get the following characterization of $\mathcal{R}_{\mathcal{P}_{conv}}(\boldsymbol{\beta})$ for linear convolutional networks.

**Lemma 7.** *For a depth–$L$ convolutional network with parameters $\mathbf{w} = [\mathbf{w}_l \in \mathbb{R}^D]_{l-1}^L$, we have*

$$\mathcal{R}_{\mathcal{P}_{conv}}(\boldsymbol{\beta}) = \min_{\mathbf{w}:\mathcal{P}_{conv}(\mathbf{w})=\boldsymbol{\beta}} \|\mathbf{w}\|_2^2 = L\|\widehat{\boldsymbol{\beta}}\|_{2/L}^{2/L} = \textit{monotone}(\|\widehat{\boldsymbol{\beta}}\|_{2/L}).$$

## 6  Discussion

In this paper, we characterized the implicit bias of gradient descent on linear convolutional networks. We showed that even in the case of linear activations and a full width convolution, wherein the convolutional network defines the exact same model class as fully connected networks, merely changing to a convolutional parameterization introduces radically different, and very interesting, bias

when training with gradient descent. Namely, training a convolutional representation with gradient descent implicitly biases towards sparsity in the frequency domain representation of linear predictor.

For convenience and simplicity of presentation, we studied one dimensional circular convolutions. Our results can be directly extended to higher dimensional input signals and convolutions, including the two-dimensional convolutions common in image processing and computer vision. We also expect similar results for convolutions with zero padding instead of circular convolutions, although this requires more care with analysis of the edge effects.

A more significant way in which our setup differs from usual convolutional networks is that we use full width convolutions, while in practice it is common to use convolutions with bounded width, much smaller then the input dimensionality. This setting is within the scope of Theorem 4, as the linear transformation is still homogeneous. However, understanding the implied bias in the predictor space, i.e. understanding $\mathcal{R}_\mathcal{P}(\boldsymbol{\beta})$ requires additional work. It will be very interesting to see if restricting the width of the convolutional network gives rise to further interesting behaviors.

Another important direction for future study is understanding the implicit bias for networks with multiple outputs. For both fully connected and convolutional networks, we looked at networks with a single output. With $C > 1$ outputs, the network implements a linear transformation $\mathbf{x} \mapsto \boldsymbol{\beta}\mathbf{x}$ where $\boldsymbol{\beta} \in \mathbb{R}^{C \times D}$ is now a matrix. Results for matrix sensing in Gunasekar et al. [2018] imply that for two layer fully connected networks with multiple outputs, the implicit bias is to a maximum margin solution with respect to the nuclear norm $\|\boldsymbol{\beta}\|_\star$. This is already different from the implicit bias of a one-layer "network" (i.e. optimizing $\boldsymbol{\beta}$ directly), which would be in terms of the Frobenius norm $\|\boldsymbol{\beta}\|_F$ (from the result of Soudry et al. [2017]). We suspect that with multiple outputs, as more layers are added, even fully connected networks exhibit a shrinking sparsity penalty on the singular values of the effective linear matrix predictor $\boldsymbol{\beta} \in \mathbb{R}^{C \times D}$. Precisely characterizing these biases requires further study.

When using convolutions as part of a larger network, with multiple parallel filters, max pooling, and non-linear activations, the situation is of course more complex, and we do not expect to get the exact same bias. However, we do expect the bias to be at the very least related to the sparsity-in-frequency-domain bias that we uncover here, and we hope our work can serve as a basis for further such study. There are of course many other implicit and explicit sources of inductive bias—here we show that merely parameterizing transformations via convolutions and using gradient descent for training already induces sparsity in the frequency domain.

On a technical level, we provided a generic characterization for the bias of gradient descent on linear models parameterized as $\boldsymbol{\beta} = \mathcal{P}(\mathbf{w})$ for a homogeneous polynomial $\mathcal{P}$. The $\ell_2$ bias (in parameter space) we obtained is not surprising, but also should not be taken for granted – *e.g.,* the result does not hold in general for non-homogeneous $\mathcal{P}$, and even with homogeneous polynomials, the characterization is not as crisp when other loss functions are used, *e.g.,* with a squared loss and matrix factorization (a homogeneous degree two polynomial representation), the implicit bias is much more fragile Gunasekar et al. [2017], Li et al. [2017]. Moreover, Theorem 4 only ensures convergence to first order stationary point in the parameter space, which is not sufficient for convergence to stationary points of the implied bias in the model space (eq. (15)). It is of interest for future work to strengthen this result to show either convergence to higher order stationary points or local minima in parameter space, or to directly show the convergence to stationary points of (15).

It would also be of interest to strengthen other technical aspects of our results: extend the results to loss functions with tight exponential tails (including logistic loss) and handle all datasets including the set of measure zero degenerate datasets—these should be possible following the techniques of Soudry et al. [2017], Telgarsky [2013], Ji and Telgarsky [2018]. We can also calculate exact rates of convergence to the asymptotic separator along the lines of Soudry et al. [2017], Nacson et al. [2018], Ji and Telgarsky [2018] showing how fast the inductive bias from optimization kicks in and why it might be beneficial to continue optimizing even after the loss value $\mathcal{L}(\boldsymbol{\beta}^{(t)})$ itself is negligible. Finally, for logistic regression, Ji and Telgarsky [2018] extend the results of asymptotic convergence of gradient descent classifier to the cases where the data is not strictly linearly separable. This is an important relaxation of our assumption on strict linear separability. More generally, for non-separable data, we would like a more fine grained analysis connecting the iterates $\boldsymbol{\beta}^{(t)}$ along the optimization path to the estimates along regularization path, $\widehat{\boldsymbol{\beta}}(c) = \operatorname{argmin}_{\mathcal{R}_\mathcal{P}(\boldsymbol{\beta}) \leq c} \mathcal{L}(\boldsymbol{\beta})$, where an explicit regularization is added to the optimization objective.

## Footnotes

[1]We follow the convention used in neural networks literature that refer to the operation in (2) as convolution, while in the signal processing terminology, (2) is known as the discrete circular cross-correlation operator

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
