[Supplementary Material 1]

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

# Appendix

The proofs of the theorems in the paper are organized as follows: In Appendix A we first give the proof for Theorem 4, which includes linear fully connected and full width convolutional networks as special cases. This gives us some general results that can be special-cased to prove the stronger results for these networks in Section 3. In Appendix B, we prove Theorem 1 on the implicit bias of fully connected linear networks. In Appendix C, we prove Theorem 2–2a on the implicit bias of linear convolutional networks. Finally, in Appendix D we prove the lemmas in Section 5 on computing the form of implicit bias of linear networks learned using gradient descent.

Unless specified otherwise, $\|.\|$ denotes the Euclidean norm. We additionally use the notation $\mathbf{v} \propto \mathbf{v}'$ to denote equality up to strictly positive scalar multipliers, *i.e.*, when $\mathbf{v} = \gamma \mathbf{v}'$ for some $\gamma > 0$.

The following is a paraphrasing of Lemma 8 in Gunasekar et al. [2018] and is used in multiple proofs.

**Lemma 8.** *[Lemma 8 in Gunasekar et al. [2018]] For almost all linearly separable dataset $\{\mathbf{x}_n, y_n\}_n$, consider any sequence $\boldsymbol{\beta}^{(t)}$ that minimizes the empirical objective in eq. (5), i.e., $\mathcal{L}(\boldsymbol{\beta}^{(t)}) \to 0$. If (a) $\overline{\boldsymbol{\beta}}^{\infty} := \lim_{t \to \infty} \frac{\boldsymbol{\beta}^{(t)}}{\|\boldsymbol{\beta}^{(t)}\|}$ exists and has a positive margin, and (b) $\overline{\mathbf{z}}^{\infty} := \lim_{t \to \infty} \frac{-\nabla_{\boldsymbol{\beta}}\mathcal{L}(\boldsymbol{\beta}^{(t)})}{\|\nabla_{\boldsymbol{\beta}}\mathcal{L}(\boldsymbol{\beta}^{(t)})\|}$ exists, then $\exists \{\alpha_n \geq 0\}_{n \in S}$ s.t. $\overline{\mathbf{z}}^{\infty} = \sum_{n \in S} \alpha_n y_n \mathbf{x}_n$, where $S = \{n : y_n \langle \overline{\boldsymbol{\beta}}^{\infty}, \mathbf{x}_n \rangle = \min_n y_n \langle \overline{\boldsymbol{\beta}}^{\infty}, \mathbf{x}_n \rangle\}$ are the indices of the data points with smallest margin to the limit direction $\overline{\boldsymbol{\beta}}^{\infty}$.*

## A  Homogeneous Polynomial Parameterization: Proof of Theorem 4

**Theorem 4** (Homogeneous Polynomial Parameterization). *For any homogeneous polynomial map $\mathcal{P} : \mathbb{R}^P \to \mathbb{R}^D$ from parameters $\mathbf{w} \in \mathbb{R}^D$ to linear predictors, almost all datasets $\{\mathbf{x}_n, y_n\}_{n=1}^N$ separable by $\mathcal{B} := \{\mathcal{P}(\mathbf{w}) : \mathbf{w} \in \mathbb{R}^P\}$, almost all initializations $\mathbf{w}^{(0)}$, and any bounded sequence of step sizes $\{\eta_t\}_t$, consider the sequence of gradient descent updates $\mathbf{w}^{(t)}$ from eq. (7) for minimizing the empirical risk objective $\mathcal{L}_{\mathcal{P}}(\mathbf{w})$ in (4) with exponential loss $\ell(u, y) = \exp(-uy)$.*

*If (a) the iterates $\mathbf{w}^{(t)}$ asymptotically minimize the objective, i.e., $\mathcal{L}_{\mathcal{P}}(\mathbf{w}^{(t)}) = \mathcal{L}(\mathcal{P}(\mathbf{w}^{(t)})) \to 0$, (b) $\mathbf{w}^{(t)}$, and consequently $\boldsymbol{\beta}^{(t)} = \mathcal{P}(\mathbf{w}^{(t)})$, converge in direction to yield a separator with positive margin, and (c) the gradients w.r.t. to the linear predictors, $\nabla_{\boldsymbol{\beta}}\mathcal{L}(\boldsymbol{\beta}^{(t)})$ converge in direction, then the limit direction of the parameters $\overline{\mathbf{w}}^{\infty} = \lim_{t \to \infty} \frac{\mathbf{w}^{(t)}}{\|\mathbf{w}^{(t)}\|_2}$ is a positive scaling of a first order stationary point of the following optimization problem,*

$$\min_{\mathbf{w} \in \mathbb{R}^P} \|\mathbf{w}\|_2^2 \quad s.t. \quad \forall n, \, y_n \langle \mathbf{x}_n, \mathcal{P}(\mathbf{w}) \rangle \geq 1. \tag{14}$$

*Proof.* $\mathbf{w}^{(t)}$ are the sequence gradient descent iterates from eq. (7) for minimizing $\mathcal{L}_{\mathcal{P}}(\mathbf{w})$ in eq (4) with exponential loss over the model class of $\mathcal{B} = \{\mathcal{P}(\mathbf{w}) : \mathbf{w} \in \mathbb{R}^P\}$, where $\mathcal{P}$ is a homogeneous polynomial function. We first introduce some notation.

1. From the assumption in theorem, we have that $\overline{\mathbf{w}}^{\infty} = \lim_{t \to \infty} \frac{\mathbf{w}^{(t)}}{\|\mathbf{w}^{(t)}\|}$. Denoting $g(t) = \|\mathbf{w}^{(t)}\|$, we have that for some $\boldsymbol{\delta}_{\mathbf{w}}^{(t)} \to 0$, the following representation of $\mathbf{w}^{(t)}$ holds.

$$\mathbf{w}^{(t)} = \overline{\mathbf{w}}^{\infty} g(t) + \boldsymbol{\delta}_{\mathbf{w}}^{(t)} g(t). \tag{16}$$

2. Let $\boldsymbol{\beta}^{(t)} = \mathcal{P}(\mathbf{w}^{(t)})$ denote the sequence of linear predictors for this network induced by the gradient descent iterates. We can see that $\boldsymbol{\beta}^{(t)}$ converges in direction too using the following arguments: homogeneity of $\mathcal{P}$ implies that $\mathcal{P}(\mathbf{w}^{(t)}/\|\mathbf{w}^{(t)}\|) = \mathcal{P}(\mathbf{w}^{(t)})/\|\mathbf{w}^{(t)}\|^{\nu}$ for some $\nu$. Hence, $\frac{\boldsymbol{\beta}^{(t)}}{\|\boldsymbol{\beta}^{(t)}\|} = \frac{\mathcal{P}(\mathbf{w}^{(t)}/\|\mathbf{w}^{(t)}\|)}{\|\mathcal{P}(\mathbf{w}^{(t)}/\|\mathbf{w}^{(t)}\|)\|} \xrightarrow{t \to \infty} \frac{\mathcal{P}(\overline{\mathbf{w}}^{\infty})}{\|\mathcal{P}(\overline{\mathbf{w}}^{\infty})\|} := \overline{\boldsymbol{\beta}}^{\infty}$.

3. $\mathbf{z}^{(t)} = -\nabla_{\boldsymbol{\beta}}\mathcal{L}(\boldsymbol{\beta}^{(t)}) = \sum_n \exp\left(-\langle \boldsymbol{\beta}^{(t)}, y_n \mathbf{x}_n \rangle\right) y_n \mathbf{x}_n$. Since we assume that $\mathbf{z}^{(t)}$ converges in direction, let $\overline{\mathbf{z}}^{\infty} = \lim_{t \to \infty} \frac{\mathbf{z}^{(t)}}{\|\mathbf{z}^{(t)}\|}$. Denoting $p(t) = \|\mathbf{z}^{(t)}\|$, for some $\boldsymbol{\delta}_{\mathbf{z}}^{(t)} \to 0$, we can write $\mathbf{z}^{(t)}$ as,

$$\mathbf{z}^{(t)} = \overline{\mathbf{z}}^{\infty} p(t) + \boldsymbol{\delta}_{\mathbf{z}}^{(t)} p(t), \tag{17}$$

4. Let $\nabla_{\mathbf{w}}\mathcal{P}\left(\mathbf{w}^{(t)}\right) \in \mathbb{R}^{P \times D}$ denote the Jacobian of $\mathcal{P}(\mathbf{w})$, *i.e.*, $\nabla_{\mathbf{w}}\mathcal{P}\left(\mathbf{w}^{(t)}\right)[p,d] = \frac{\partial(\mathcal{P}(\mathbf{w}^{(t)})[d])}{\partial \mathbf{w}[p]}$. If $\mathcal{P} : \mathbb{R}^P \to \mathbb{R}^D$ is a homogeneous polynomial of degree $\nu > 0$, then $\nabla_{\mathbf{w}}\mathcal{P} : \mathbb{R}^P \to \mathbb{R}^{P \times D}$ is a homogeneous polynomial of degree $\nu - 1$. Using eq. (16), we have

$$\nabla_{\mathbf{w}}\mathcal{P}(\overline{\mathbf{w}}^\infty) = \lim_{t \to \infty} \nabla_{\mathbf{w}}\mathcal{P}\left(\frac{\mathbf{w}^{(t)}}{g(t)}\right) = \lim_{t \to \infty} \frac{\nabla_{\mathbf{w}}\mathcal{P}(\mathbf{w}^{(t)})}{g(t)^{\nu-1}}$$

Thus, $\exists \boldsymbol{\delta}_1^{(t)} \to 0$, such that

$$\nabla_{\mathbf{w}}\mathcal{P}\left(\mathbf{w}^{(t)}\right) = \nabla_{\mathbf{w}}\mathcal{P}\left(\overline{\mathbf{w}}^\infty\right) g(t)^{\nu-1} + \boldsymbol{\delta}_1^{(t)} g(t)^{\nu-1}. \tag{18}$$

5. Finally, from the definition of $\nabla_{\mathbf{w}}\mathcal{P}(\mathbf{w})$, we have $\nabla_{\mathbf{w}}\mathcal{L}_\mathcal{P}(\mathbf{w}^{(t)}) = \nabla_{\mathbf{w}}\mathcal{P}\left(\mathbf{w}^{(t)}\right)\nabla_{\boldsymbol{\beta}}\mathcal{L}(\boldsymbol{\beta}^{(t)})$, and hence from eq. (7),

$$\Delta\mathbf{w}^{(t)} := \mathbf{w}^{(t+1)} - \mathbf{w}^{(t)} = \eta_t \nabla_{\mathbf{w}}\mathcal{P}\left(\mathbf{w}^{(t)}\right)\mathbf{z}^{(t)} \tag{19}$$

Using the assumptions in the theorem along with our argument above for convergence of $\boldsymbol{\beta}^{(t)}$ in direction, we satisfy the conditions of Lemma 8, which will be crucially used in our proof.

**KKT conditions for first order stationary points**   We want show that there exists a positive scaling of $\overline{\mathbf{w}}^\infty$, denoted as $\widetilde{\mathbf{w}}^\infty = \gamma\overline{\mathbf{w}}^\infty$ for some $\gamma > 0$, such that $\widetilde{\mathbf{w}}^\infty$ is a first order stationary point of the explicitly regularized problem in eq. (14). Towards this we show that $\widetilde{\mathbf{w}}^\infty$ satisfy the following first order KKT conditions of eq. (14)

$$\forall n, \ y_n \langle \mathbf{x}_n, \mathcal{P}(\mathbf{w}) \rangle \geq 1,$$
$$\exists \{\alpha_n\}_{n=1}^N \text{ s.t. } \forall n, \alpha_n \geq 0 \text{ and } \alpha_n = 0, \forall n \notin S := \{n \in [N] : y_n \langle \mathbf{x}_n, \mathcal{P}(\mathbf{w}) \rangle = 1\}, \tag{20}$$
$$\mathbf{w} = \nabla_{\mathbf{w}}\mathcal{P}(\mathbf{w}) \left[\sum_n \alpha_n y_n \mathbf{x}_n\right].$$

**Primal feasibility.**   We showed earlier that if $\mathbf{w}^{(t)}$ converges in direction, then $\boldsymbol{\beta}^{(t)} = \mathcal{P}(\mathbf{w}^{(t)})$ converges in direction to $\overline{\boldsymbol{\beta}}^\infty = \lim_{t \to \infty} \frac{\boldsymbol{\beta}^{(t)}}{\|\boldsymbol{\beta}^{(t)}\|} \propto \mathcal{P}(\overline{\mathbf{w}}^\infty)$. Further, from the assumptions in the theorem, we have that $\overline{\boldsymbol{\beta}}^\infty$ satisfies $\forall n, y_n \langle \mathbf{x}_n, \overline{\boldsymbol{\beta}}^\infty \rangle > 0$, which also implies $\min_n y_n \langle \mathbf{x}_n, \mathcal{P}(\overline{\mathbf{w}}^\infty) \rangle > 0$ since $\overline{\boldsymbol{\beta}}^\infty \propto \mathcal{P}(\overline{\mathbf{w}}^\infty)$. Now, if $\mathcal{P}$ is homogeneous of of degree $\nu$, then for $\gamma = (\min_n y_n \langle \mathbf{x}_n, \mathcal{P}(\overline{\mathbf{w}}^\infty) \rangle)^{-1/\nu}$, $\widetilde{\mathbf{w}}^\infty = \gamma\overline{\mathbf{w}}^\infty$ satisfies $\min_n y_n \langle \mathbf{x}_n, \mathcal{P}(\widetilde{\mathbf{w}}^\infty) \rangle = 1$.

**Showing other KKT conditions for $\widetilde{\mathbf{w}}^\infty$.**   The crux of the proof of Theorem 4 involves showing the existence of $\{\alpha_n \geq 0\}_n$ such that the stationarity and complementary slackness conditions in eq. (20) are satisfied. This crucially relies on a key lemma (Lemma 8) showing that the gradient in the space of linear predictors $\nabla_{\boldsymbol{\beta}}\mathcal{L}(\boldsymbol{\beta}^{(t)})$ are dominated by positive linear combinations of support vectors of the asymptotic predictor $\overline{\boldsymbol{\beta}}^\infty$.

Let $S_\infty = \{n : y_n \langle \mathcal{P}(\widetilde{\mathbf{w}}^\infty), \mathbf{x}_n \rangle = 1\}$ denote the indices of support vectors for $\mathcal{P}(\widetilde{\mathbf{w}}^\infty)$, which are also the support vectors of $\overline{\boldsymbol{\beta}}^\infty$, since by homogeneity of $\mathcal{P}$, $\overline{\boldsymbol{\beta}}^\infty \propto \mathcal{P}(\overline{\mathbf{w}}^\infty) \propto \mathcal{P}(\widetilde{\mathbf{w}}^\infty)$. Thus, from Lemma 8, we have $\overline{\mathbf{z}}^\infty = \lim_{t \to \infty} \frac{\mathbf{z}^{(t)}}{\|\mathbf{z}^{(t)}\|} = \sum_{n \in S_\infty} \alpha_n y_n \mathbf{x}_n$ for some $\{\alpha_n\}_{n \in S_\infty}$ such that $\alpha_n \geq 0$. We propose a positive scaling of this $\{\alpha_n\}_{n=1}^N$ as our candidate dual certificate, which satisfies both dual feasibility and complementary slackness.

To prove the theorem, the remaining step is to show that $\widetilde{\mathbf{w}}^\infty \propto \nabla_{\mathbf{w}}\mathcal{P}(\widetilde{\mathbf{w}}^\infty)\overline{\mathbf{z}}^\infty$. Since $\widetilde{\mathbf{w}}^\infty = \gamma\overline{\mathbf{w}}^\infty$ and $\mathcal{P}$ is homogeneous, this condition is equivalent to showing that $\overline{\mathbf{w}}^\infty \propto \nabla_{\mathbf{w}}\mathcal{P}(\overline{\mathbf{w}}^\infty)\overline{\mathbf{z}}^\infty$.

**Showing that $\overline{\mathbf{w}}^\infty \propto \nabla_{\mathbf{w}}\mathcal{P}(\overline{\mathbf{w}}^\infty)\overline{\mathbf{z}}^\infty$.**   Substituting for $\mathbf{z}^{(t)}$ and $\nabla_{\mathbf{w}}\mathcal{P}(\mathbf{w}^{(t)})$ from eqs. (17) and (18), respectively, in the gradient descent updates (eq. (19)), we have the following:

$$\mathbf{w}^{(t+1)} - \mathbf{w}^{(t)} = \eta_t \nabla_{\mathbf{w}}\mathcal{P}\left(\mathbf{w}^{(t)}\right)\mathbf{z}^{(t)}$$
$$= \eta_t \left(\nabla_{\mathbf{w}}\mathcal{P}\left(\overline{\mathbf{w}}^\infty\right) g(t)^{\nu-1} + \boldsymbol{\delta}_1^{(t)} g(t)^{\nu-1}\right)\left(\overline{\mathbf{z}}^\infty p(t) + \boldsymbol{\delta}_{\mathbf{z}}^{(t)} p(t)\right) \tag{21}$$
$$\overset{(a)}{=} \left(\eta_t p(t) g(t)^{\nu-1}\right)\left[\nabla_{\mathbf{w}}\mathcal{P}\left(\overline{\mathbf{w}}^\infty\right)\overline{\mathbf{z}}^\infty + \boldsymbol{\delta}^{(t)}\right],$$

where in $(a)$ $\boldsymbol{\delta}^{(t)} = \nabla_{\mathbf{w}} \mathcal{P}\left(\overline{\mathbf{w}}^{\infty}\right) \boldsymbol{\delta}_{\mathbf{z}}^{(t)} + \boldsymbol{\delta}_1^{(t)} \boldsymbol{\delta}_{\mathbf{z}}^{(t)} + \boldsymbol{\delta}_1^{(t)} \overline{\mathbf{z}}^{\infty} \to 0$.

Summing over $t$, we have

$$\mathbf{w}^{(t)} - \mathbf{w}^{(0)} = \nabla_{\mathbf{w}} \mathcal{P}\left(\overline{\mathbf{w}}^{\infty}\right) \overline{\mathbf{z}}^{\infty} \sum_{u<t} \eta_u p(u) g(u)^{\nu-1} + \sum_{u<t} \boldsymbol{\delta}^{(u)} \eta_u p(u) g(u)^{\nu-1}, \qquad (22)$$

We want to argue that the first term, *i.e.,* $\nabla_{\mathbf{w}} \mathcal{P}\left(\overline{\mathbf{w}}^{\infty}\right) \overline{\mathbf{z}}^{\infty}$, is the dominant term. Towards this we state and prove the following intermediate claim

**Claim 1.** $\|\nabla_{\mathbf{w}} \mathcal{P}\left(\overline{\mathbf{w}}^{\infty}\right) \overline{\mathbf{z}}^{\infty}\| > 0$ *and* $\sum_{u<t} \eta_u p(u) g(u)^{\nu-1} \to \infty$.

*Proof.* First, it is straight forward to check that for any scalar valued homogeneous polynomial $f :$ $\mathbb{R}^P \to \mathbb{R}$ of degree $\nu$, we have $\langle \mathbf{w}, \nabla_{\mathbf{w}} f(\mathbf{w}) \rangle = \nu f(\mathbf{w})$, where for $p = 1, 2 \ldots, P$, $\nabla_{\mathbf{w}} f(\mathbf{w})[p] = \frac{\mathrm{d} f(\mathbf{w})}{\mathrm{d} \mathbf{w}[p]}$ (this is also known as the Euler's homogeneous function theorem). Extending this to our vector valued homogeneous function $\mathcal{P} : \mathbb{R}^P \to \mathbb{R}^D$, we have that for all $\mathbf{w}$, the Jacobian $\nabla_{\mathbf{w}} \mathcal{P}(\mathbf{w}) \in \mathbb{R}^{P \times D}$ satisfies $\nabla_{\mathbf{w}} \mathcal{P}(\mathbf{w})^{\top} \mathbf{w} = \nu \mathcal{P}(\mathbf{w})$.

Moreover, we have that for the limit direction $\overline{\mathbf{w}}^{\infty}$, the margin of the corresponding classifier is strictly positive, *i.e.,* $\min_n y_n \langle \mathcal{P}(\overline{\mathbf{w}}^{\infty}), \mathbf{x}_n \rangle > 0$. Now from Lemma 8, using that $\overline{\mathbf{z}}^{\infty} = \sum_{n \in S_{\infty}} \alpha_n y_n \mathbf{x}_n$ for $\alpha_n \geq 0$ (and not all zero since $\overline{\mathbf{z}}^{\infty}$ is unit norm), we immediately get the following

$$\overline{\mathbf{w}}^{\infty \top} \nabla_{\mathbf{w}} \mathcal{P}(\overline{\mathbf{w}}^{\infty}) \overline{\mathbf{z}}^{\infty} = \nu \mathcal{P}(\overline{\mathbf{w}}^{\infty})^{\top} \overline{\mathbf{z}}^{\infty} = \nu \sum_n \alpha_n y_n \langle \mathbf{x}_n, \mathcal{P}(\mathbf{w}^{\infty}) \rangle > 0 \implies \nabla_{\mathbf{w}} \mathcal{P}(\overline{\mathbf{w}}^{\infty}) \overline{\mathbf{z}}^{\infty} \neq 0.$$

To prove the second part, we note the following

- since $\boldsymbol{\delta}^{(t)} \to 0$ in eq. (22), $\exists t_0$ such that $\forall t > t_0$, $\|\boldsymbol{\delta}^{(t)}\| \leq 1$, and since all the incremental updates to gradient descent are finite, we have that $\sup_t \|\boldsymbol{\delta}^{(t)}\| < \infty$,
- since $p(t) = \|\mathbf{z}^{(t)}\|$ and $g(t) = \|\mathbf{w}^{(t)}\|$ are positive, we have that $b_t = \sum_{u<t} \eta_u p(u) g(u)^{\nu-1}$ is monotonic increasing.

Thus, if $\limsup_{t \to \infty} b_t = \infty$ then $\lim_{t \to \infty} b_t = \infty$. On contrary, if $\limsup_{t \to \infty} b_t = C < \infty$, then from eq. (22), for large $t$ we get, $\|\mathbf{w}^{(t)}\| \leq \|\mathbf{w}^{(0)}\| + \|\nabla \mathcal{P}(\overline{\mathbf{w}}^{\infty}) \overline{\mathbf{z}}^{\infty}\| C + \left(\sup_t \|\boldsymbol{\delta}^{(t)}\|\right) C < \infty$ which contradicts $\|\mathbf{w}^{(t)}\| \to \infty$. $\qquad \square$

From above claim, the sequence $b_t = \sum_{u<t} \eta_u p(u) g(u)^{\nu-1}$ is monotonic increasing and diverging. Thus, for $a_t = \sum_{u<t} \boldsymbol{\delta}^{(u)} \eta_u p(u) g(u)^{\nu-1}$, using Stolz-Cesaro theorem (Theorem 11), we have

$$\lim_{t \to \infty} \frac{a_t}{b_t} = \lim_{t \to \infty} \frac{\sum_{u<t} \boldsymbol{\delta}^{(u)} \eta_u p(u) g(u)^{\nu-1}}{\sum_{u<t} \eta_u p(u) g(u)^{\nu-1}} = \lim_{t \to \infty} \frac{a_{t+1} - a_t}{b_{t+1} - b_t} = \lim_{t \to \infty} \boldsymbol{\delta}^{(t)} = 0.$$

$$\implies \text{for } \boldsymbol{\delta}_2^{(t)} \to 0, \text{ we have } \sum_{u<t} \boldsymbol{\delta}^{(u)} \eta_u p(u) g(u)^{\nu-1} = \boldsymbol{\delta}_2^{(t)} \sum_{u<t} \eta_u p(u) g(u)^{\nu-1}. \qquad (23)$$

Substituting eq. (23) in eq. (22), we have

$$\mathbf{w}^{(t)} \stackrel{(a)}{=} \left[\nabla_{\mathbf{w}} \mathcal{P}\left(\overline{\mathbf{w}}^{\infty}\right) \overline{\mathbf{z}}^{\infty} + \boldsymbol{\delta}_3^{(t)}\right] \left[\sum_{u<t} \eta_u p(u) g(u)^{\nu-1}\right] \qquad (24)$$

$$\implies \frac{\mathbf{w}^{(t)}}{\|\mathbf{w}^{(t)}\|} = \frac{\nabla_{\mathbf{w}} \mathcal{P}\left(\overline{\mathbf{w}}^{\infty}\right) \overline{\mathbf{z}}^{\infty} + \boldsymbol{\delta}_3^{(t)}}{\|\nabla_{\mathbf{w}} \mathcal{P}\left(\overline{\mathbf{w}}^{\infty}\right) \overline{\mathbf{z}}^{\infty} + \boldsymbol{\delta}_3^{(t)}\|} \stackrel{(b)}{\to} \frac{\nabla_{\mathbf{w}} \mathcal{P}\left(\overline{\mathbf{w}}^{\infty}\right) \overline{\mathbf{z}}^{\infty}}{\|\nabla_{\mathbf{w}} \mathcal{P}\left(\overline{\mathbf{w}}^{\infty}\right) \overline{\mathbf{z}}^{\infty}\|} \qquad (25)$$

$$\implies \overline{\mathbf{w}}^{\infty} = \lim_{t \to \infty} \frac{\mathbf{w}^{(t)}}{\|\mathbf{w}^{(t)}\|} = \frac{\nabla_{\mathbf{w}} \mathcal{P}\left(\overline{\mathbf{w}}^{\infty}\right) \overline{\mathbf{z}}^{\infty}}{\|\nabla_{\mathbf{w}} \mathcal{P}\left(\overline{\mathbf{w}}^{\infty}\right) \overline{\mathbf{z}}^{\infty}\|} \propto \nabla_{\mathbf{w}} \mathcal{P}\left(\overline{\mathbf{w}}^{\infty}\right) \overline{\mathbf{z}}^{\infty}, \qquad (26)$$

where in $(a)$ we absorbed the diminishing terms into $\boldsymbol{\delta}_3^{(t)} = \boldsymbol{\delta}_2^{(t)} + \mathbf{w}^{(0)} / \sum_{u<t} \eta_u p(u) g(u)^{\nu-1} \to 0$, $(b)$ follows since we proved in the claim above that $\nabla_{\mathbf{w}} \mathcal{P}\left(\overline{\mathbf{w}}^{\infty}\right) \overline{\mathbf{z}}^{\infty} \neq 0$ and hence dominates $\boldsymbol{\delta}_3^{(t)}$. We have shown that $\overline{\mathbf{w}}^{\infty} = \overline{\gamma} \nabla_{\mathbf{w}} \mathcal{P}\left(\overline{\mathbf{w}}^{\infty}\right) \overline{\mathbf{z}}^{\infty}$ for a positive scalar $\overline{\gamma}$, which completes the proof. $\quad \square$

# B    Linear Fully Connected Networks: Proof of Theorem 1

**Theorem 1** (Linear fully connected networks). *For any depth L, almost all linearly separable datasets $\{\mathbf{x}_n, y_n\}_{n=1}^N$, almost all initializations $\mathbf{w}^{(0)}$, and any bounded sequence of step sizes $\{\eta_t\}_t$, consider the sequence gradient descent iterates $\mathbf{w}^{(t)}$ in eq. (7) for minimizing $\mathcal{L}_{\mathcal{P}_{full}}(\mathbf{w})$ in eq. (4) with exponential loss $\ell(\widehat{y}, y) = \exp(-\widehat{y}y)$ over L–layer fully connected linear networks.*

*If (a) the iterates $\mathbf{w}^{(t)}$ minimize the objective, i.e., $\mathcal{L}_{\mathcal{P}_{full}}(\mathbf{w}^{(t)}) \to 0$, (b) $\mathbf{w}^{(t)}$, and consequently $\boldsymbol{\beta}^{(t)} = \mathcal{P}_{full}(\mathbf{w}^{(t)})$, converge in direction to yield a separator with positive margin, and (c) gradients with respect to linear predictors $\nabla_{\boldsymbol{\beta}} \mathcal{L}(\boldsymbol{\beta}^{(t)})$ converge in direction, then the limit direction is given by,*

$$\overline{\boldsymbol{\beta}}^{\infty} = \lim_{t\to\infty} \frac{\mathcal{P}_{full}(\mathbf{w}^{(t)})}{\|\mathcal{P}_{full}(\mathbf{w}^{(t)})\|} = \frac{\boldsymbol{\beta}_{\ell_2}^*}{\|\boldsymbol{\beta}_{\ell_2}^*\|}, \text{ where } \boldsymbol{\beta}_{\ell_2}^* := \underset{w}{\operatorname{argmin}} \|\boldsymbol{\beta}\|_2^2 \text{ s.t. } \forall n, y_n\langle \mathbf{x}_n, \boldsymbol{\beta}\rangle \geq 1. \quad (8)$$

*Proof.* Recall that for fully connected networks of any depth $L > 0$ with parameters $\mathbf{w} = [\mathbf{w}_l \in \mathbb{R}^{D_{l-1} \times D_l}]_{l-1}^L$, the equivalent linear predictor given by $\mathcal{P}_{full}(\mathbf{w}) = \mathbf{w}_1 \mathbf{w}_2 \ldots \mathbf{w}_L$ is a homogeneous polynomial of degree $L$.

Let $\mathbf{w}^{(t)} = [\mathbf{w}_l^{(t)} \in \mathbb{R}^{D_{l-1} \times D_l}]_{l=1}^L$ denote the iterates of individual matrices $\mathbf{w}_l$ along the gradient descent path, and $\boldsymbol{\beta}^{(t)} = \mathcal{P}_{full}(\mathbf{w}^{(t)})$ denote the corresponding sequence of linear predictors.

We first introduce the following notation.

1. Let $\overline{\mathbf{w}}^{\infty} = \lim_{t\to\infty} \frac{\mathbf{w}^{(t)}}{\|\mathbf{w}^{(t)}\|}$ denote the limit direction of the parameters, with component matrices in each layer denoted as $\overline{\mathbf{w}}^{\infty} = [\overline{\mathbf{w}}_l^{\infty}]$. Specializing (16) for fully connected networks, we have:

$$\mathbf{w}_l^{(t)} = \overline{\mathbf{w}}_l^{\infty} g(t) + \boldsymbol{\delta}_{\mathbf{w}_l}^{(t)} g(t), \quad (27)$$

   where $g(t) = \|\mathbf{w}^{(t)}\|$ and $\boldsymbol{\delta}_{\mathbf{w}_l}^{(t)} \to 0$.

2. For $0 < l_1 < l_2 \leq L$, denote $\mathbf{w}_{l_1:l_2}^{(t)} = \mathbf{w}_{l_1}^{(t)} \mathbf{w}_{l_1+1}^{(t)} \ldots \mathbf{w}_{l_2}^{(t)}$ and $\overline{\mathbf{w}}_{l_1:l_2}^{\infty} = \overline{\mathbf{w}}_{l_1}^{\infty} \overline{\mathbf{w}}_{l_1+1}^{\infty} \ldots \overline{\mathbf{w}}_{l_2}^{\infty}$. Using eq. (27), we can check by induction on $l_2 - l_1$ that $\lim_{t\to\infty} \frac{\mathbf{w}_{l_1:l_2}^{(t)}}{g(t)^{l_2-l_1+1}} = \overline{\mathbf{w}}_{l_1:l_2}^{\infty}$, and hence $\exists \boldsymbol{\delta}_{\mathbf{w}_{l_1:l_2}}^{(t)} \to 0$ such that the following holds,

$$\mathbf{w}_{l_1:l_2}^{(t)} = \overline{\mathbf{w}}_{l_1:l_2}^{\infty} g(t)^{l_2-l_1+1} + \boldsymbol{\delta}_{\mathbf{w}_{l_1:l_2}}^{(t)} g(t)^{l_2-l_1+1}. \quad (28)$$

3. Let $\mathbf{z}^{(t)} = -\nabla_{\boldsymbol{\beta}} \mathcal{L}(\boldsymbol{\beta}^{(t)})$. Again repeating eq. (17) for fully connected networks, we have for some $\boldsymbol{\delta}_{\mathbf{z}}^{(t)} \to 0$ and $p(t) = \|\mathbf{z}^{(t)}\|$,

$$\mathbf{z}^{(t)} = \overline{\mathbf{z}}^{\infty} p(t) + \boldsymbol{\delta}_{\mathbf{z}}^{(t)} p(t). \quad (29)$$

4. From Lemma 8, we have that $\exists \{\alpha_n\}_{n\in S_{\infty}}$ such that $\overline{\mathbf{z}}^{\infty} = \sum_{n\in S_{\infty}} \alpha_n y_n \mathbf{x}_n$, where $S_{\infty}$ are support vectors of $\overline{\boldsymbol{\beta}}^{\infty} = \lim_{t\to\infty} \frac{\boldsymbol{\beta}^{(t)}}{\|\boldsymbol{\beta}^{(t)}\|} \propto \mathcal{P}_{full}(\overline{\mathbf{w}}^{\infty})$.

The proof of Theorem 1 is fairly straight forward from using Lemma 8 and the intermediate results in the proof of Theorem 4.

**Showing KKT conditions for $\widetilde{\boldsymbol{\beta}}^{\infty} \propto \mathcal{P}_{full}(\overline{\mathbf{w}}^{\infty})$.** Using our notation described above, we have $\overline{\mathbf{w}}_{1:L}^{\infty} = \mathcal{P}_{full}(\overline{\mathbf{w}}^{\infty})$. In the following arguments we show that a positive scaling $\widetilde{\boldsymbol{\beta}}^{\infty} = \gamma \overline{\mathbf{w}}_{1:L}^{\infty}$ satisfies the following KKT conditions for the optimality of $\ell_2$ maximum margin problem in eq. (8):

$$\exists \{\alpha_n\}_{n=1}^N \quad \text{s.t.} \quad \forall n, y_n\langle \mathbf{x}_n, \boldsymbol{\beta}\rangle \geq 1, \boldsymbol{\beta} = \sum_n \alpha_n y_n \mathbf{x}_n,$$
$$\forall n, \alpha_n \geq 0 \text{ and } \alpha_n = 0, \forall i \notin S := \{i \in [N] : y_n\langle \mathbf{x}_n, \boldsymbol{\beta}\rangle = 1\}. \quad (30)$$

As we saw in proof of Theorem 4, since $\overline{\mathbf{w}}_{1:L}^{\infty} = \mathcal{P}_{full}(\overline{\mathbf{w}}^{\infty})$ has strictly positive margin, using homogeneity of $\mathcal{P}_{full}$, we can scale $\overline{\mathbf{w}}_{1:L}^{\infty}$ to get $\widetilde{\boldsymbol{\beta}}^{\infty} = \gamma \overline{\mathbf{w}}_{1:L}^{\infty}$ with unit margin, *i.e.*,

$\forall n,\ y_n \langle \mathbf{x}_n, \widetilde{\boldsymbol{\beta}}^\infty \rangle \geq 1$. For dual variables, we again use a positive scaling of $\alpha_n$ from Lemma 8, such that $\overline{\mathbf{z}}^\infty = \sum_{n \in S_\infty} \alpha_n y_n \mathbf{x}_n$. In order to prove the theorem, we need to show that $\widetilde{\boldsymbol{\beta}}^\infty \propto \overline{\mathbf{z}}^\infty$ or equivalently $\overline{\mathbf{w}}_{1:L}^\infty \propto \overline{\mathbf{z}}^\infty$.

Recall that in the proof of Theorem 4, we showed a version of stationarity in the parameter space in eq. (26), repeated below.

$$\overline{\mathbf{w}}^\infty \propto \nabla \mathbf{w} \mathcal{P}(\overline{\mathbf{w}}^\infty) \overline{\mathbf{z}}^\infty. \tag{31}$$

This case in particular includes $\mathcal{P}_{full}$ which is homogeneous with $\nu = L$. We special case the result fully connected network. In particular, for the parameters of the first layer $\mathbf{w}_1$, we have $\mathcal{P}(\mathbf{w}) = \mathbf{w}_1 \mathbf{w}_{2:L}$, where $\mathbf{w}_1 \in \mathbb{R}^{d \times d_1}$ and $\mathbf{w}_{2:L} \in \mathbb{R}^{d_1 \times 1}$. This implies, for any $\mathbf{z}$, $\nabla_{\mathbf{w}_1} \mathcal{P}(\mathbf{w}) \mathbf{z} = \mathbf{z} \mathbf{w}_{2:L}^\top$. Using this along with eq. (31), we get the following expression for some positive scalar $\overline{\gamma}$

$$\overline{\mathbf{w}}_1^\infty = \overline{\gamma} \nabla_{\mathbf{w}_1} \mathcal{P}(\overline{\mathbf{w}}^\infty) \overline{\mathbf{z}}^\infty = \overline{\gamma} \overline{\mathbf{z}}^\infty \overline{\mathbf{w}}_{2:L}^{\infty \top} \implies \overline{\mathbf{w}}_{1:L}^\infty = \overline{\mathbf{w}}_1^\infty \overline{\mathbf{w}}_{2:L}^\infty = \overline{\gamma} \overline{\mathbf{z}}^\infty \cdot \|\overline{\mathbf{w}}_{2:L}^\infty\|^2 \propto \overline{\mathbf{z}}^\infty. \tag{32}$$

Since $\overline{\mathbf{w}}_{1:L}^\infty \propto \widetilde{\boldsymbol{\beta}}^\infty$, we have shown that $\widetilde{\boldsymbol{\beta}}^\infty \propto \overline{\mathbf{z}}^\infty$, which completes our proof of Theorem 1. □

## C   Linear Convolutional Networks: Proof of Theorem 2–2a

Recall that $L$–layer linear convolutional networks have parameters $\mathbf{w} = [\mathbf{w}_l \in \mathbb{R}^D]_{l-1}^L$. We first recall some complex numbers terminology and properties

1. Complex vectors $\widehat{\mathbf{z}} \in \mathbf{C}^D$ are represented in polar form as $\widehat{\mathbf{z}} = |\widehat{\mathbf{z}}| e^{i \phi_{\widehat{\mathbf{z}}}}$, where $|\widehat{\mathbf{z}}| \in \mathbb{R}_+^D$ and $\phi_{\widehat{\mathbf{z}}} \in [0, 2\pi)^D$ are the vectors with magnitudes and phases, respectively, of components $\widehat{\mathbf{z}}$.
2. For $\widehat{\mathbf{z}} = |\widehat{\mathbf{z}}| e^{i \phi_{\widehat{\mathbf{z}}}} \in \mathbf{C}^D$, the complex conjugate vector is denoted by $\widehat{\mathbf{z}}^* = |\widehat{\mathbf{z}}| e^{-i \phi_{\widehat{\mathbf{z}}}}$.
3. The complex inner product for $\widehat{\mathbf{x}}, \widehat{\boldsymbol{\beta}} \in \mathbf{C}^D$ is given by $\langle \widehat{\mathbf{x}}, \widehat{\boldsymbol{\beta}} \rangle = \sum_d \widehat{\mathbf{x}}[d] \widehat{\boldsymbol{\beta}}^*[d] = \widehat{\mathbf{x}}^\top \widehat{\boldsymbol{\beta}}^*$.
4. Let $\mathcal{F} \in \mathbb{C}^{D \times D}$ denote the discrete Fourier transform matrix with $\mathcal{F}[d, p] = \frac{1}{\sqrt{D}} \omega_D^{dp}$ where recall that $\omega_D = e^{-\frac{2\pi i}{D}}$ is the $D^{\text{th}}$ complex root of unity. Thus, for any $\mathbf{z} \in \mathbb{R}^D$, the representation in Fourier basis is given by $\widehat{\mathbf{z}} = \mathcal{F} \mathbf{z}$. $\mathcal{F}$ and its complex conjugate matrix $\mathcal{F}^*$ also satisfy: $\mathcal{F} \mathcal{F}^* = \mathcal{F}^* \mathcal{F} = I$, $\mathcal{F} = \mathcal{F}^\top$ and $\mathcal{F}^* = \mathcal{F}^{*\top}$.

Before getting into full proofs of Theorem 2a–2, we also prove the two lemmas (Lemma 3 and Lemma 9) that establish equivalence of dynamics of gradient descent on full dimensional convolutional networks to those on linear diagonal networks (Figure 1c), albeit with complex valued parameters. This makes the analysis of the of convolutional networks simpler and more intuitive.

We begin by proving Lemma 3 which shows the equivalence of representation between convolutional networks and diagonal networks.

**Lemma 3.** *For full-dimensional convolutions,* $\boldsymbol{\beta} = \mathcal{P}_{conv}(\mathbf{w})$ *is equivalent to*

$$\widehat{\boldsymbol{\beta}} = diag(\widehat{\mathbf{w}}_1) \dots diag(\widehat{\mathbf{w}}_{L-1}) \widehat{\mathbf{w}}_L,$$

*where for* $l = 1, 2, \dots, L,$ $\widehat{\mathbf{w}}_1 \in \mathbf{C}^D$ *are the Fourier coefficients of the parameters* $\mathbf{w}_l \in \mathbb{R}^D$.

*Proof.* First, we state the following properties which follow immediately from definitions:

1. For $\mathbf{x}, \boldsymbol{\beta} \in \mathbb{R}^D$,

$$\langle \mathbf{x}, \boldsymbol{\beta} \rangle = \mathbf{x}^\top \boldsymbol{\beta} = \mathbf{x}^\top \mathcal{F} \mathcal{F}^* \boldsymbol{\beta} = \widehat{\mathbf{x}}^\top \widehat{\boldsymbol{\beta}}^* = \langle \widehat{\mathbf{x}}, \widehat{\boldsymbol{\beta}} \rangle, \tag{33}$$

where recall that the complex inner product is given by $\langle \widehat{\mathbf{x}}, \widehat{\boldsymbol{\beta}} \rangle = \widehat{\mathbf{x}}^\top \widehat{\boldsymbol{\beta}}^*$.

2. We next show the following property

$$\mathcal{F}(\mathbf{h} \star \mathbf{w}) = (\mathcal{F} \mathbf{h}) \odot (\mathcal{F}^* \mathbf{w}) = \widehat{\mathbf{h}} \odot \widehat{\mathbf{w}}^*, \tag{34}$$

where $\odot$ denotes the Hadamard product (elementwise product), *i.e.,* $(a \odot b)[d] = a[d]b[d]$.

The above equation follows from simple manipulations of definitions: recall that $(\mathcal{F} \mathbf{z})[d] = \frac{1}{\sqrt{D}} \sum_{p=0}^{D-1} \mathbf{z}[p] \omega_D^{pd}$ and $\mathbf{h} \star \mathbf{w}$ defined in eq. (2) as $(\mathbf{h} \star \mathbf{w})[d] = \frac{1}{\sqrt{D}} \sum_{k=0}^{D-1} \mathbf{w}[k] \mathbf{h}[(d+k) \bmod D]$.

$$\widehat{\mathbf{h}} \odot \widehat{\mathbf{w}}^*[d] = \widehat{\mathbf{h}}[d]\widehat{\mathbf{w}}^*[d] = \frac{1}{D} \sum_{k=0}^{D-1} \sum_{k'=0}^{D-1} \mathbf{w}[k]\mathbf{h}[k']\omega_D^{(k'-k)d} \overset{(a)}{=} \frac{1}{D} \sum_{k=0}^{D-1} \sum_{k'=0}^{D-1} \mathbf{w}[k]\mathbf{h}[k']\omega_D^{((k'-k) \bmod \mathrm{D})d}$$

$$\overset{(b)}{=} \frac{1}{\sqrt{D}} \sum_{p=0}^{D-1} \left[ \frac{1}{\sqrt{D}} \sum_{k=0}^{D-1} \mathbf{w}[k]\mathbf{h}[(p+k) \bmod D] \right] \omega_D^{pd} = (\mathcal{F}(\mathbf{h} \star \mathbf{w}))[d], \qquad (35)$$

where $(a)$ follows as $\omega_D^D = 1$ and in $(b)$ we used the change of variables $p = (k' - k) \bmod D$ (recall our use of modulo operator as $a \bmod D = a - D\lfloor \frac{a}{D} \rfloor$).

Recall from eq. (3) the output of an $L$-layer convolutional network is given by

$$\widehat{y}(\mathbf{x}) = ((((\mathbf{x} \star \mathbf{w}_1) \star \mathbf{w}_2) \ldots) \star \mathbf{w}_{L-1})^\top \mathbf{w}_L = \langle \mathbf{x}, \boldsymbol{\beta} \rangle.$$

Denote $\mathbf{h}_{L-1}(\mathbf{x}) = (((\mathbf{x} \star \mathbf{w}_1) \star \mathbf{w}_2) \ldots) \star \mathbf{w}_{L-1}$. By iteratively using eq. (34), we have

$$\mathcal{F}\mathbf{h}_{L-1}(\mathbf{x}) = \mathcal{F}\mathbf{x} \odot \mathcal{F}^*\mathbf{w}_1 \odot \mathcal{F}^*\mathbf{w}_2 \ldots \odot \mathcal{F}^*\mathbf{w}_{L-1}. \qquad (36)$$

Thus, on one hand using the above equation we have,

$$\widehat{y}(\mathbf{x}) = \mathbf{h}_{L-1}(x)^\top \mathbf{w}_L = \mathbf{h}_{L-1}(\mathbf{x}))^\top \mathcal{F}\mathcal{F}^*\mathbf{w}_L = (\mathcal{F}\mathbf{h}_{L-1}(\mathbf{x}))^\top (\mathcal{F}^*\mathbf{w}_L)$$
$$\overset{(a)}{=} (\mathcal{F}(\mathbf{x}))^\top (\mathcal{F}^*\mathbf{w}_1 \odot \mathcal{F}^*\mathbf{w}_2 \ldots \odot \mathcal{F}^*\mathbf{w}_L) \overset{(b)}{=} \langle \widehat{\mathbf{x}}, \mathcal{F}\mathbf{w}_1 \odot \mathcal{F}\mathbf{w}_2 \ldots \odot \mathcal{F}\mathbf{w}_L \rangle, \qquad (37)$$

where $(a)$ follows from substituting for $\mathcal{F}\mathbf{h}_{L-1}(\mathbf{x})$ from eq. (36) and noting that for any $\{\mathbf{z}_l \in \mathbb{R}^D\}$, $(\mathbf{z}_1 \odot \mathbf{z}_2 \odot \ldots \mathbf{z}_{L-1})^\top \mathbf{z}_L = \mathbf{z}_1^\top (\mathbf{z}_2 \odot \mathbf{z}_3 \odot \ldots \mathbf{z}_L)$, and $(b)$ uses the definition of complex inner product $\langle \widehat{\mathbf{x}}, \widehat{\boldsymbol{\beta}} \rangle = \widehat{\mathbf{x}}^\top \widehat{\boldsymbol{\beta}}^*$.

Now further using eq. (33) in above equation, we have

$$\langle \mathbf{x}, \mathcal{P}_{conv}(\mathbf{w}) \rangle = \langle \widehat{\mathbf{x}}, \mathcal{F}\mathcal{P}_{conv}(\mathbf{w}) \rangle = \widehat{y}(\mathbf{x})$$
$$\implies \langle \widehat{\mathbf{x}}, \mathcal{F}\mathcal{P}_{conv}(\mathbf{w}) \rangle = \langle \widehat{\mathbf{x}}, \mathcal{F}\mathbf{w}_1 \odot \mathcal{F}\mathbf{w}_2 \ldots \odot \mathcal{F}\mathbf{w}_L \rangle. \qquad (38)$$

Thus, for $\boldsymbol{\beta} = \mathcal{P}_{conv}(\mathbf{w})$, we have shown that $\widehat{\boldsymbol{\beta}} = \mathcal{F}\mathcal{P}_{conv}(\mathbf{w}) = \widehat{\mathbf{w}}_1 \odot \widehat{\mathbf{w}}_2 \ldots \odot \widehat{\mathbf{w}}_L = \mathrm{diag}(\widehat{\mathbf{w}}_1)\mathrm{diag}(\widehat{\mathbf{w}}_2) \ldots \mathrm{diag}(\widehat{\mathbf{w}}_{L-1})\widehat{\mathbf{w}}_L$. □

For $\widehat{\mathbf{w}} = [\widehat{\mathbf{w}}_l \in \mathbf{C}^D]_{l=1}^L$, let $\mathcal{P}_{diag}(\widehat{\mathbf{w}}) = \mathrm{diag}(\widehat{\mathbf{w}}_1)\mathrm{diag}(\widehat{\mathbf{w}}_2) \ldots \mathrm{diag}(\widehat{\mathbf{w}}_{L-1})\widehat{\mathbf{w}}_L = \widehat{\mathbf{w}}_1 \odot \widehat{\mathbf{w}}_2 \ldots \odot \widehat{\mathbf{w}}_L$ denote the equivalent parameterization of convolutional network in Fourier domain.

The above lemma shows that optimizing $\mathcal{L}_{\mathcal{P}_{conv}}(\mathbf{w})$ in eq. (4) is equivalent to the following minimization problem in terms of representation,

$$\min_{\widehat{\mathbf{w}}} \widehat{\mathcal{L}}_{\mathcal{P}_{diag}}(\widehat{\mathbf{w}}) := \sum_{n=1}^N \ell(\langle \widehat{\mathbf{x}}_n, \mathcal{P}_{diag}(\widehat{\mathbf{w}}) \rangle, y_n) \qquad (39)$$

The following lemma further shown that not only the representations of $\mathcal{P}_{conv}(\mathbf{w})$ and $\mathcal{P}_{diag}(\widehat{\mathbf{w}})$ are equivalent, but there corresponding gradient descent updates for problems in eq. (4) and eq. (39) are also equivalent up to Fourier transformations.

**Lemma 9.** *Consider the gradient descent iterates* $\mathbf{w}^{(t)} = [\mathbf{w}_l^{(t)}]_{l=1}^L$ *from eq. (7) for minimizing* $\mathcal{L}_{\mathcal{P}_{conv}}$ *in eq. (4) over full dimensional linear convolutional networks. For all $l$, the incremental update directions,* $\Delta\mathbf{w}_l^{(t)} := \mathbf{w}_l^{(t+1)} - \mathbf{w}_l^{(t)} = -\eta_t \nabla_{\mathbf{w}_l} \mathcal{L}_{\mathcal{P}_{conv}}(\mathbf{w}^{(t)})$ *satisfy the following,*

$$\mathcal{F}\Delta\mathbf{w}_l^{(t)} = \widehat{\mathbf{w}}_l^{(t+1)} - \widehat{\mathbf{w}}_l^{(t)} = -\eta_t \nabla_{\widehat{\mathbf{w}}_l} \widehat{\mathcal{L}}_{\mathcal{P}_{diag}}(\widehat{\mathbf{w}}^{(t)}), \qquad (40)$$

*where* $\widehat{\mathbf{w}}^{(t)} = \left[\widehat{\mathbf{w}}_l^{(t)}\right]_{l=1}^L$ *are the Fourier transformations of* $\mathbf{w}^{(t)} = [\mathbf{w}_l^{(t)}]_{l=1}^L$, *respectively.*

The above lemma shows that Fourier transformation of the gradient descent iterates $\mathbf{w}^{(t)} = [\mathbf{w}_l^{(t)}]_{l=1}^L$ for $\mathcal{L}_{\mathcal{P}_{conv}}$ in eq. (4) are equivalently obtained by gradient descent on the complex parameters $\widehat{\mathbf{w}}$ for minimizing $\widehat{\mathcal{L}}_{\mathcal{P}_{diag}}$ in eq. (39)

*Proof.* We use the notation $\underset{l' \neq l}{\odot} \widehat{\mathbf{w}}_{l'} = \widehat{\mathbf{w}}_1 \odot \widehat{\mathbf{w}}_2 \ldots \widehat{\mathbf{w}}_{l-1} \odot \widehat{\mathbf{w}}_{l+1} \ldots \odot \widehat{\mathbf{w}}_L$ to denote Hadamard product across all parameters $\widehat{\mathbf{w}}_{l'}$ with $l' \neq l$.

For any $\mathbf{w} = [\mathbf{w}_l]_{l=1}^L$, using eq. (38), we have the following for all $l$,

$$\langle \mathbf{x}, \mathcal{P}_{conv}(\mathbf{w}) \rangle = \langle \widehat{\mathbf{x}}, \mathcal{P}_{diag}(\widehat{\mathbf{w}}) \rangle = \widehat{\mathbf{x}}^\top (\widehat{\mathbf{w}}_1^* \odot \widehat{\mathbf{w}}_2^* \odot \ldots \widehat{\mathbf{w}}_L^*) = \widehat{\mathbf{w}}_l^{*\top} \left[ \left( \underset{l' \neq l}{\odot} \widehat{\mathbf{w}}_{l'}^* \right) \odot \widehat{\mathbf{x}} \right]. \quad (41)$$

Using the above equation we have,

$$\ell(\langle \mathbf{x}, \mathcal{P}_{conv}(\mathbf{w}) \rangle, y_n) = \ell \left( \mathbf{w}_l^\top \, \mathcal{F}^* \left[ \left( \underset{l' \neq l}{\odot} \widehat{\mathbf{w}}_{l'}^* \right) \odot \widehat{\mathbf{x}} \right], y_n \right)$$

$$\implies \nabla_{\mathbf{w}_l} \ell(\langle \mathbf{x}, \mathcal{P}_{conv}(\mathbf{w}) \rangle, y_n) \overset{(a)}{=} \ell'(\langle \mathbf{x}, \mathcal{P}_{conv}(\mathbf{w}) \rangle, y_n) \mathcal{F}^* \left[ \left( \underset{l' \neq l}{\odot} \widehat{\mathbf{w}}_l^* \right) \odot \widehat{\mathbf{x}} \right] \quad (42)$$

$$= \mathcal{F}^* \left[ \ell'(\langle \widehat{\mathbf{x}}, \mathcal{P}_{diag}(\widehat{\mathbf{w}}) \rangle, y_n) \left( \underset{l' \neq l}{\odot} \widehat{\mathbf{w}}_l^* \right) \odot \widehat{\mathbf{x}} \right] = \mathcal{F}^* \nabla_{\widehat{\mathbf{w}}_l} \ell(\langle \widehat{\mathbf{x}}, \mathcal{P}_{diag}(\widehat{\mathbf{w}}) \rangle, y_n).$$

where in $(a)$ we use $\ell'(\widehat{y}, y) = \frac{\partial \ell(\widehat{y}, y)}{\partial \widehat{y}}$ and the remaining equalities simply follow from manipulation of derivatives. From above equation, we have the following:

$$\mathcal{F} \Delta \mathbf{w}_l^{(t)} = -\eta_t \mathcal{F} \nabla_{\mathbf{w}_l} \mathcal{L}_{\mathcal{P}_{conv}}(\mathbf{w}^{(t)}) = -\eta_t \mathcal{F} \sum_{n=1}^N \nabla_{\mathbf{w}_l} \ell(\langle \mathbf{x}_n, \mathcal{P}_{conv}(\mathbf{w}^{(t)}) \rangle, y_n)$$

$$= -\eta_t \mathcal{F} \mathcal{F}^* \sum_{n=1}^N \nabla_{\widehat{\mathbf{w}}_l} \ell(\langle \widehat{\mathbf{x}}_n, \mathcal{P}_{diag}(\widehat{\mathbf{w}}^{(t)}) \rangle, y_n) = -\eta_t \nabla_{\widehat{\mathbf{w}}_l} \widehat{\mathcal{L}}_{\mathcal{P}_{diag}}(\widehat{\mathbf{w}}^{(t)}). \quad \square$$

## C.1 Proof of Theorem 2–2a

**Theorem 2** (Linear convolutional networks of depth two). *For almost all linearly separable datasets* $\{\mathbf{x}_n, y_n\}_{n=1}^N$, *almost all initializations* $\mathbf{w}^{(0)}$, *and any sequence of step sizes* $\{\eta_t\}_t$ *with* $\eta_t$ *smaller than the local Lipschitz at* $\mathbf{w}^{(t)}$, *consider the sequence gradient descent iterates* $\mathbf{w}^{(t)}$ *in eq. (7) for minimizing* $\mathcal{L}_{\mathcal{P}_{conv}}(\mathbf{w})$ *in eq. (4) with exponential loss over 2–layer linear convolutional networks.*

*If (a) the iterates* $\mathbf{w}^{(t)}$ *minimize the objective, i.e.,* $\mathcal{L}_{\mathcal{P}_{conv}}(\mathbf{w}^{(t)}) \to 0$, *(b)* $\mathbf{w}^{(t)}$ *converge in direction to yield a separator* $\overline{\boldsymbol{\beta}}^\infty$ *with positive margin, (c) the phase of the Fourier coefficients* $\widehat{\boldsymbol{\beta}}^{(t)}$ *of the linear predictors* $\boldsymbol{\beta}^{(t)}$ *converge coordinate-wise, i.e.,* $\forall d, e^{i\phi_{\widehat{\boldsymbol{\beta}}^{(t)}[d]}} \to e^{i\phi_{\widehat{\boldsymbol{\beta}}^\infty[d]}}$, *and (d) the gradients* $\nabla_{\boldsymbol{\beta}} \mathcal{L}(\boldsymbol{\beta}^{(t)})$ *converge in direction, then the limit direction* $\overline{\boldsymbol{\beta}}^\infty$ *is given by,*

$$\overline{\boldsymbol{\beta}}^\infty = \frac{\boldsymbol{\beta}_{\mathcal{F},1}^*}{\|\boldsymbol{\beta}_{\mathcal{F},1}^*\|}, \text{ where } \boldsymbol{\beta}_{\mathcal{F},1}^* := \underset{\boldsymbol{\beta}}{\operatorname{argmin}} \|\widehat{\boldsymbol{\beta}}\|_1 \text{ s.t. } \forall n, y_n \langle \boldsymbol{\beta}, \mathbf{x}_n \rangle \geq 1. \quad (9)$$

**Theorem 2a** (Linear Convolutional Networks of any Depth). *For any depth L, under the conditions of Theorem 2, the limit direction* $\overline{\boldsymbol{\beta}}^\infty = \lim_{t \to \infty} \frac{\mathcal{P}_{conv}(\mathbf{w}^{(t)})}{\|\mathcal{P}_{conv}(\mathbf{w}^{(t)})\|}$ *is a scaling of a first order stationary point of the following optimization problem,*

$$\min_{\boldsymbol{\beta}} \|\widehat{\boldsymbol{\beta}}\|_{2/L} \text{ s.t. } \forall n, y_n \langle \boldsymbol{\beta}, \mathbf{x}_n \rangle \geq 1, \quad (10)$$

*where the* $\ell_p$ *penalty given by* $\|z\|_p = \left( \sum_{i=1}^D |z[i]|^p \right)^{1/p}$ *(also called the bridge penalty) is a norm for* $p = 1$ *and a quasi-norm for* $p < 1$.

For the gradient descent iterates $\mathbf{w}^{(t)} = [\mathbf{w}_l^{(t)}]_{l=1}^L$ from eq. (7) denote the sequence of corresponding linear predictors as $\boldsymbol{\beta}^{(t)} = \mathcal{P}_{conv}(\mathbf{w}^{(t)})$. Let $\widehat{\boldsymbol{\beta}}^{(t)} = \mathcal{F}\boldsymbol{\beta}^{(t)}$ and $\widehat{\mathbf{w}}_l^{(t)} = \mathcal{F}\mathbf{w}_l^{(t)}$ denote the Fourier transforms of $\boldsymbol{\beta}^{(t)}$ and $\mathbf{w}_l^{(t)}$, respectively, and let $\widehat{\mathbf{w}}^{(t)} = \left[ \widehat{\mathbf{w}}_l^{(t)} \right]_{l=1}^L$.

Summarizing the results so far, we have $\widehat{\boldsymbol{\beta}}^{(t)} = \widehat{\mathbf{w}}_1^{(t)} \odot \widehat{\mathbf{w}}_2^{(t)} \ldots \odot \widehat{\mathbf{w}}_L^{(t)}$ (from Lemma 3) and $\Delta \widehat{\mathbf{w}}_l^{(t)} := \widehat{\mathbf{w}}_l^{(t+1)} - \widehat{\mathbf{w}}_l^{(t)} = -\eta_t \nabla_{\widehat{\mathbf{w}}_l} \widehat{\mathcal{L}}_{\mathcal{P}_{diag}}(\widehat{\mathbf{w}}^{(t)})$ (from Lemma 9).

We use the following observations/notations

1. Let $\overline{\mathbf{w}}^\infty = \lim_{t\to\infty} \frac{\mathbf{w}^{(t)}}{\|\mathbf{w}^{(t)}\|}$. Denote the Fourier transform of $\overline{\mathbf{w}}^\infty = [\overline{\mathbf{w}}_l^\infty]$ as $\widehat{\overline{\mathbf{w}}}^\infty = [\widehat{\overline{\mathbf{w}}}_l^\infty]$.
   Taking Fourier transforms of eq. (16) which are also applicable here, we have:

$$\widehat{\mathbf{w}}_l^{(t)} = \widehat{\overline{\mathbf{w}}}_l^\infty g(t) + \widehat{\boldsymbol{\delta}}_{\mathbf{w}_l}^{(t)} g(t), \tag{43}$$

   where $g(t) = \|\mathbf{w}^{(t)}\| = \|\widehat{\mathbf{w}}^{(t)}\|$ and $\widehat{\boldsymbol{\delta}}_{\mathbf{w}_l}^{(t)} \to 0$.

2. Denote the negative gradients with respect to $\boldsymbol{\beta}^{(t)}$ as $\mathbf{z}^{(t)} = -\nabla_\beta \mathcal{L}(\boldsymbol{\beta}^{(t)})$ and let $\widehat{\mathbf{z}}^{(t)} = \mathcal{F}\mathbf{z}^{(t)}$. From the assumption of Theorem 2-2a, $\lim_{t\to\infty} \frac{\mathbf{z}^{(t)}}{\|\mathbf{z}^{(t)}\|}$ exists. Let $\overline{\mathbf{z}}^\infty = \lim_{t\to\infty} \frac{\mathbf{z}^{(t)}}{\|\mathbf{z}^{(t)}\|}$.
   Denote $\widehat{\overline{\mathbf{z}}}^\infty = \mathcal{F}\overline{\mathbf{z}}^\infty$. We get the following by taking Fourier transform of eq. (17)

$$\widehat{\mathbf{z}}^{(t)} = \widehat{\overline{\mathbf{z}}}^\infty p(t) + \widehat{\boldsymbol{\delta}}_{\mathbf{z}}^{(t)} p(t), \tag{44}$$

   where $p(t) = \|\mathbf{z}^{(t)}\| = \|\widehat{\mathbf{z}}^{(t)}\|$ and $\widehat{\boldsymbol{\delta}}_{\mathbf{z}}^{(t)} \to 0$.

3. From Lemma 8, we have that $\exists\{\alpha_n\}_{n\in S_\infty}$ such that $\lim_{t\to\infty} \frac{\mathbf{z}^{(t)}}{\|\mathbf{z}^{(t)}\|} = \sum_{n\in S_\infty} \alpha_n y_n \mathbf{x}_n$. Thus,

$$\widehat{\overline{\mathbf{z}}}^\infty = \sum_{n\in S_\infty} \alpha_n y_n \widehat{\mathbf{x}}_n. \tag{45}$$

**KKT conditions for optimality**   We want to show that a positive scaling of $\overline{\boldsymbol{\beta}}^\infty \propto \mathcal{P}_{conv}(\overline{\mathbf{w}}^\infty)$, denoted by $\widetilde{\boldsymbol{\beta}}^\infty = \gamma \mathcal{P}_{conv}(\overline{\mathbf{w}}^\infty)$ is a first order stationary point of eq. (10), repeated below,

$$\min_{\boldsymbol{\beta}} \|\widehat{\boldsymbol{\beta}}\|_{2/L} \text{ s.t. } \forall n,\, y_n \langle \boldsymbol{\beta}, \mathbf{x}_n \rangle \geq 1.$$

Recall the KKT conditions discussed in Section 3. The first order stationary points, or sub-stationary points, of (10) are the set of feasible predictors $\boldsymbol{\beta}$ such that $\exists\{\alpha_n \geq 0\}_{n=1}^N$ satisfying the following:
$\forall n,\, y_n \langle \mathbf{x}_n, \boldsymbol{\beta}\rangle > 1 \implies \alpha_n = 0$, and

$$\sum_n \alpha_n y_n \widehat{\mathbf{x}}_n \in \partial^\circ \|\widehat{\boldsymbol{\beta}}\|_p, \tag{46}$$

where $\partial^\circ$ denotes the local sub-differential (or Clarke's sub-differential) operator defined as $\partial^\circ f(\boldsymbol{\beta}) = \text{conv}\{\mathbf{v} : \exists(\mathbf{z}_k)_k \text{ s.t. } \mathbf{z}_k \to \boldsymbol{\beta} \text{ and } \nabla f(\mathbf{z}_k) \to \mathbf{v}\}$.

For $p = 1$ and $\widehat{\boldsymbol{\beta}}$ represented in polar form as $\widehat{\boldsymbol{\beta}} = |\widehat{\boldsymbol{\beta}}| e^{i\phi_{\hat{\beta}}} \in \mathbb{C}^D$, $\|\widehat{\boldsymbol{\beta}}\|_p$ is convex and the local sub-differential is indeed the global sub-differential given by,

$$\partial^\circ \|\widehat{\boldsymbol{\beta}}\|_1 = \{\mathbf{z} : \forall d,\, |\mathbf{z}[d]| \leq 1 \text{ and } \widehat{\boldsymbol{\beta}}[d] \neq 0 \implies \mathbf{z}[d] = e^{i\phi_{\hat{\beta}}[d]}\}. \tag{47}$$

For $p < 1$, the local sub-differential of $\|\widehat{\boldsymbol{\beta}}\|_p$ is given by,

$$\forall p < 1, \quad \partial^\circ \|\widehat{\boldsymbol{\beta}}\|_p = \{\mathbf{z} : \widehat{\boldsymbol{\beta}}[d] \neq 0 \implies \mathbf{z}[d] = p\, e^{i\phi_{\hat{\beta}}[d]} |\widehat{\boldsymbol{\beta}}[d]|^{p-1}\}. \tag{48}$$

**Showing KKT conditions for $\widetilde{\boldsymbol{\beta}}^\infty \propto \mathcal{P}_{conv}(\overline{\mathbf{w}}^\infty)$.**   As we showed proof of Theorem 4, since $\mathcal{P}_{conv}(\overline{\mathbf{w}}^\infty)$ has strictly positive margin, using homogeneity of $\mathcal{P}_{conv}$, we can scale $\mathcal{P}_{conv}(\overline{\mathbf{w}}^\infty)$ to get $\widetilde{\boldsymbol{\beta}}^\infty = \gamma \mathcal{P}_{conv}(\overline{\mathbf{w}}^\infty)$ with unit margin, i.e., $\forall n,\, y_n \langle \mathbf{x}_n, \widetilde{\boldsymbol{\beta}}^\infty \rangle \geq 1$. For dual variables, we again use a positive scaling of $\alpha_n$ from Lemma 8, such that $\overline{\mathbf{z}}^\infty = \sum_{n\in S_\infty} \alpha_n y_n \mathbf{x}_n$.

In order to prove the theorem, we need to show that for some positive scalar $\overline{\gamma}$, $\overline{\gamma}\widehat{\overline{\mathbf{z}}}^\infty \in \partial^\circ \|\widehat{\boldsymbol{\beta}}\|_{2/L}$, i.e., satisfies the conditions in eq. (47) and (48), for $L = 2$ and $L > 2$, respectively.

We start from the stationarity condition in the parameter space in eq. (26) of Theorem 4. For some positive scalar $\overline{\gamma}$, we have

$$\overline{\mathbf{w}}^\infty = \overline{\gamma} \nabla_\mathbf{w} \mathcal{P}_{conv}(\overline{\mathbf{w}}^\infty) \overline{\mathbf{z}}^\infty. \tag{49}$$

We will now special case the above equation for fully width convolutional networks.

From Lemma 3, we have that for all $\mathbf{w} = [\mathbf{w}_l \in \mathbb{R}^D]$, we have $\mathcal{P}_{conv}(\mathbf{w}) = \mathcal{F}^* \mathcal{P}_{diag}(\mathcal{F}\mathbf{w})$ where $\mathcal{F}$ and $\mathcal{F}^*$ denote discrete Fourier matrix and its inverse in appropriate dimensions. Let $\{e_d\}_{d=1}^D$

denote the standard basis in $\mathbb{R}^D$. We first note that for all $l = 1, 2, \ldots, L$ and for all $d = 1, 2, \ldots, D$, the following holds

$$\mathcal{P}_{conv}(\mathbf{w})[d] = e_d^\top \mathcal{F}^* \mathcal{P}_{diag}(\mathcal{F}\mathbf{w}) = e_d^\top \mathcal{F}^* \left( \odot_{l'=1}^{L-1} \widehat{\mathbf{w}}_{l'} \right) \tag{50}$$

$$= e_d^\top \mathcal{F}^* \left( \prod_{l' \neq l} \text{diag}(\widehat{\mathbf{w}}_{l'}) \right) \mathcal{F}\mathbf{w}_l = \langle \mathbf{w}_l, \mathcal{F}^* \left( \prod_{l' \neq l} \text{diag}(\widehat{\mathbf{w}}_{l'}^*) \right) \mathcal{F}e_d \rangle. \tag{51}$$

$$\implies \nabla_{\mathbf{w}_l} \mathcal{P}_{conv}(\mathbf{w})[:, d] = \mathcal{F}^* \left( \prod_{l' \neq l} \text{diag}(\widehat{\mathbf{w}}_{l'}^*) \right) \mathcal{F}e_d. \tag{52}$$

This implies, for $l = 1, 2, \ldots, L$ and any $\mathbf{z} \in \mathbb{R}^D$, we have

$$\nabla_{\mathbf{w}_l} \mathcal{P}_{conv}(\mathbf{w})\mathbf{z} = \sum_d \nabla_{\mathbf{w}_l} \mathcal{P}_{conv}(\mathbf{w})[:, d]\mathbf{z}[d] = \mathcal{F}^* \left( \prod_{l' \neq l} \text{diag}(\widehat{\mathbf{w}}_{l'}^*) \right) \mathcal{F}\mathbf{z}. \tag{53}$$

Substituting the above equation in eq. (49), we have,

$$\widehat{\overline{\mathbf{w}}}_l^\infty = \mathcal{F}\overline{\mathbf{w}}_l^\infty = \overline{\gamma}\mathcal{F}\nabla_{\mathbf{w}_l}\mathcal{P}_{conv}(\overline{\mathbf{w}}^\infty)\overline{\mathbf{z}}^\infty = \overline{\gamma}\left( \odot_{l' \neq l} \widehat{\overline{\mathbf{w}}}_{l'}^{\infty*} \right) \odot \widehat{\overline{\mathbf{z}}}^\infty, \tag{54}$$

where $\widehat{\overline{\mathbf{w}}}_{l'}^{\infty*}$ denotes the complex conjugate of $\widehat{\overline{\mathbf{w}}}_{l'}^\infty$.

Let $\widehat{\overline{\boldsymbol{\beta}}}^\infty = \mathcal{P}_{diag}(\widehat{\overline{\mathbf{w}}}^\infty)$. The above equation, further implies, for all $l$

$$|\widehat{\overline{\mathbf{w}}}_l^\infty|^2 = \widehat{\overline{\mathbf{w}}}_l^{\infty*} \odot \widehat{\overline{\mathbf{w}}}_l^\infty = \overline{\gamma}\widehat{\overline{\boldsymbol{\beta}}}^{\infty*} \odot \widehat{\overline{\mathbf{z}}}^\infty = \overline{\gamma}|\widehat{\overline{\boldsymbol{\beta}}}^\infty| \odot |\widehat{\overline{\mathbf{z}}}^\infty|e^{i(\phi_{\widehat{\overline{\mathbf{z}}}^\infty} - \phi_{\widehat{\overline{\boldsymbol{\beta}}}^\infty})} \tag{55}$$

In eq. (55), since the LHS is a real number, we have that for all $d$ such that $|\widehat{\overline{\boldsymbol{\beta}}}^\infty[d]| > 0$

$$e^{i\phi_{\widehat{\overline{\mathbf{z}}}^\infty}[d]} = e^{i\phi_{\widehat{\overline{\boldsymbol{\beta}}}^\infty}[d]}. \tag{56}$$

Also, by multiplying the LHS of eq. (55) across all $l$ and taking $L$th root over positive scalars, we have for $d = 0, 1, \ldots, D - 1$,

$$\left| \widehat{\overline{\boldsymbol{\beta}}}^\infty[d] \right|^{2/L} = \overline{\gamma} \left| \widehat{\overline{\boldsymbol{\beta}}}^\infty[d] \right| |\widehat{\overline{\mathbf{z}}}^\infty[d]|, \tag{57}$$

Finally, let $\gamma$ be a positive scaling of $\overline{\boldsymbol{\beta}}^\infty$ such that $\widetilde{\boldsymbol{\beta}}^\infty = \gamma\overline{\boldsymbol{\beta}}^\infty$ has unit margin. Let $\widehat{\widetilde{\boldsymbol{\beta}}}^\infty = \mathcal{F}\widetilde{\boldsymbol{\beta}}^\infty = \gamma\widehat{\overline{\boldsymbol{\beta}}}^\infty$. Since $\overline{\gamma}$ is arbitrary positive scalar, redefining as $\overline{\gamma} \leftarrow \frac{2}{L}\gamma^{2/L-1}\overline{\gamma}$, we have from eq. (56)-(57),

$$\forall d \text{ s.t. } \left| \widehat{\widetilde{\boldsymbol{\beta}}}^\infty[d] \right| \neq 0, \quad \overline{\gamma}\widehat{\overline{\mathbf{z}}}[d] = e^{i\phi_{\widehat{\widetilde{\boldsymbol{\beta}}}}[d]} \left| \widehat{\widetilde{\boldsymbol{\beta}}}^\infty[d] \right|^{2/L-1} \tag{58}$$

**C.1.1   Case of $L > 2$ or $p = {}^2/_L < 1$**

For $p = {}^2/_L < 1$, since $\widehat{\overline{\mathbf{z}}}^\infty = \sum_{n \in S_\infty} \alpha_n y_n \widehat{\mathbf{x}}_n$, eq. (58) is indeed the first order stationarity condition for eq. (10) as described in eq. (11) and (13).

**C.1.2   Case of $L = 2$ or $p = {}^2/_L = 1$**

For the case of $p = 1$, in addition to eq. (58), we need to show that $\overline{\gamma}|\widehat{\overline{\mathbf{z}}}^\infty| \leq 1$. From eq. (58), for $L = 2$ we have $\left| \widehat{\widetilde{\boldsymbol{\beta}}}^\infty[d] \right| \neq 0 \implies \overline{\gamma}|\widehat{\overline{\mathbf{z}}}^\infty[d]| = 1$.

We need to further show that $\forall d$ s.t. $\left| \widehat{\widetilde{\boldsymbol{\beta}}}^\infty[d] \right| \propto \left| \widehat{\overline{\boldsymbol{\beta}}}^\infty[d] \right| = 0, \overline{\gamma}|\widehat{\overline{\mathbf{z}}}^\infty[d]| \leq 1$.

**Showing** $\forall d, \left|\widehat{\overline{\boldsymbol{\beta}}}^{\infty}[d]\right| = 0 \implies \overline{\gamma}|\widehat{\overline{\mathbf{z}}}^{\infty}[d]| \leq 1$

Using Lemma 9 for for the special case of 2–layer linear convolutional network, for $\forall d$,

$$\Delta\widehat{\mathbf{w}}_1^{(t)}[d] = \eta_t \widehat{\mathbf{z}}^{(t)}[d]\,\widehat{\mathbf{w}}_2^{(t)*}[d],$$
$$\Delta\widehat{\mathbf{w}}_2^{(t)}[d] = \eta_t \widehat{\mathbf{z}}^{(t)}[d]\,\widehat{\mathbf{w}}_1^{(t)*}[d]. \tag{59}$$

Recall: for $l = 1, 2$, $\frac{\widehat{\mathbf{w}}_l^{(t)}}{g(t)} \to \widehat{\overline{\mathbf{w}}}_l^{\infty}$, $\frac{\widehat{\mathbf{z}}^{(t)}}{p(t)} \to \widehat{\overline{\mathbf{z}}}^{\infty}$, $\widehat{\boldsymbol{\beta}}^{(t)} = \widehat{\mathbf{w}}_1^{(t)} \odot \widehat{\mathbf{w}}_2^{(t)}$ and $\frac{\widehat{\boldsymbol{\beta}}^{(t)}}{g(t)^2} \to \widehat{\overline{\boldsymbol{\beta}}}^{\infty} = \widehat{\overline{\mathbf{w}}}_1^{\infty} \odot \widehat{\overline{\mathbf{w}}}_2^{\infty}$.

Further, from eq. (55), we have $\forall d, |\widehat{\overline{\mathbf{w}}}_1^{\infty}[d]|^2 = |\widehat{\overline{\mathbf{w}}}_2^{\infty}[d]|^2$, and hence

$$|\widehat{\overline{\mathbf{w}}}_1^{\infty}[d]| = |\widehat{\overline{\mathbf{w}}}_2^{\infty}[d]| = \sqrt{|\widehat{\overline{\boldsymbol{\beta}}}^{\infty}[d]|}. \tag{60}$$

From the convergence of complex numbers, we have the following:

1. $\forall d$ such that $|\widehat{\overline{\mathbf{z}}}^{\infty}[d]| \neq 0$, we have

$$\frac{|\widehat{\mathbf{z}}^{(t)}[d]|}{p(t)} \to |\widehat{\overline{\mathbf{z}}}^{\infty}[d]| \text{ and } \mathrm{e}^{\mathrm{i}\phi_{\widehat{\mathbf{z}}^{(t)}[d]}} \to \mathrm{e}^{\mathrm{i}\phi_{\widehat{\overline{\mathbf{z}}}^{\infty}[d]}}. \tag{61}$$

2. $\forall d$ such that $|\widehat{\overline{\boldsymbol{\beta}}}^{\infty}[d]| \neq 0$, we have $|\widehat{\overline{\mathbf{w}}}_1^{\infty}[d]|, |\widehat{\overline{\mathbf{w}}}_2^{\infty}[d]| \neq 0$, and the following holds

for $l = 1, 2,$  $\dfrac{|\widehat{\mathbf{w}}_l^{(t)}[d]|}{g(t)} \to |\widehat{\overline{\mathbf{w}}}_l^{\infty}[d]|$  and  $\mathrm{e}^{\mathrm{i}\phi_{\widehat{\mathbf{w}}_l^{(t)}[d]}} \to \mathrm{e}^{\mathrm{i}\phi_{\widehat{\overline{\mathbf{w}}}_l^{\infty}[d]}}$

$$\dfrac{|\widehat{\boldsymbol{\beta}}^{(t)}[d]|}{g(t)^2} \to |\widehat{\overline{\boldsymbol{\beta}}}^{\infty}[d]| \text{ and } \mathrm{e}^{\mathrm{i}\phi_{\widehat{\boldsymbol{\beta}}^{(t)}[d]}} \to \mathrm{e}^{\mathrm{i}\phi_{\widehat{\overline{\boldsymbol{\beta}}}^{\infty}[d]}} = \mathrm{e}^{\mathrm{i}\phi_{\widehat{\overline{\mathbf{w}}}_1^{\infty}[d]}} \cdot \mathrm{e}^{\mathrm{i}\phi_{\widehat{\overline{\mathbf{w}}}_2^{\infty}[d]}} \tag{62}$$

$$\overline{\gamma}|\widehat{\overline{\mathbf{z}}}^{\infty}[d]| = 1 \text{ and } \mathrm{e}^{\mathrm{i}\phi_{\widehat{\overline{\mathbf{z}}}^{\infty}[d]}} = \mathrm{e}^{\mathrm{i}\phi_{\widehat{\overline{\boldsymbol{\beta}}}^{\infty}[d]}},$$

where the last equation follows from eq. (56).

3. $\forall d$ such that $|\widehat{\overline{\boldsymbol{\beta}}}^{\infty}[d]| = 0$, from eq. (60), we have $|\widehat{\overline{\mathbf{w}}}_1^{\infty}[d]| = |\widehat{\overline{\mathbf{w}}}_2^{\infty}[d]| = 0$.

In the remainder of the proof, we only consider $d$ with $|\widehat{\overline{\mathbf{z}}}^{\infty}[d]| \neq 0$.

Consider $\mathbf{u}_d^{(t)}$ defined below,

$$\mathbf{u}_d^{(t)} := \widehat{\mathbf{w}}_1^{(t)}[d] \cdot \mathrm{e}^{-\mathrm{i}\phi_{\widehat{\overline{\mathbf{z}}}^{\infty}[d]}} + \widehat{\mathbf{w}}_2^{(t)*}[d]. \tag{63}$$

Since for $l = 1, 2$, $\mathbf{w}_l^{(t)}/g(t) \to \overline{\mathbf{w}}_l^{\infty}$, we have the following:

$$\lim_{t\to\infty} \frac{\mathbf{u}_d^{(t)}}{g(t)} = \widehat{\overline{\mathbf{w}}}_1^{\infty}[d] \cdot \mathrm{e}^{-\mathrm{i}\phi_{\widehat{\overline{\mathbf{z}}}^{\infty}[d]}} + \widehat{\overline{\mathbf{w}}}_2^{\infty*}[d] \overset{(a)}{=} \begin{cases} 0 & \text{if } |\widehat{\overline{\boldsymbol{\beta}}}^{\infty}[d]| = 0 \\ \mathrm{e}^{-\mathrm{i}\phi_{\widehat{\overline{\mathbf{w}}}_2^{\infty}[d]}}\left[|\widehat{\overline{\mathbf{w}}}_1^{\infty}[d]| + |\widehat{\overline{\mathbf{w}}}_2^{\infty}[d]|\right] & \text{if } |\widehat{\overline{\boldsymbol{\beta}}}^{\infty}[d]| > 0 \end{cases}$$

$$\overset{(b)}{=} \begin{cases} 0 & \text{if } |\widehat{\overline{\boldsymbol{\beta}}}^{\infty}[d]| = 0 \\ 2\mathrm{e}^{-\mathrm{i}\phi_{\widehat{\overline{\mathbf{w}}}_2^{\infty}[d]}}\sqrt{|\widehat{\overline{\boldsymbol{\beta}}}^{\infty}[d]|} & \text{if } |\widehat{\overline{\boldsymbol{\beta}}}^{\infty}[d]| > 0 \end{cases}, \tag{64}$$

where $(a)$ follows from using $\mathrm{e}^{\mathrm{i}\phi_{\widehat{\overline{\mathbf{z}}}^{\infty}[d]}} = \mathrm{e}^{\mathrm{i}\phi_{\widehat{\overline{\boldsymbol{\beta}}}^{\infty}[d]}} = \mathrm{e}^{\mathrm{i}\phi_{\widehat{\overline{\mathbf{w}}}_1^{\infty}[d]}} \cdot \mathrm{e}^{\mathrm{i}\phi_{\widehat{\overline{\mathbf{w}}}_2^{\infty}[d]}}$ whenever $\overline{\boldsymbol{\beta}}^{\infty}[d] \neq 0$ (from eq. (62)), and $(b)$ follows from eq. (60).

Step 1. *Dynamics of* $\mathbf{u}_d^{(t)}$: Now looking at the dynamics of $\mathbf{u}_d$, using eq. (59) we have that

$$\mathbf{u}_d^{(t+1)} = \mathbf{u}_d^{(t)} + \mathrm{e}^{-\mathrm{i}\phi_{\widehat{\overline{\mathbf{z}}}^{\infty}[d]}} \cdot \eta_t \widehat{\mathbf{z}}^{(t)}[d]\,\widehat{\mathbf{w}}_2^{(t)*}[d] + \eta_t \widehat{\mathbf{z}}^{(t)*}[d]\,\widehat{\mathbf{w}}_1^{(t)}[d]$$

$$= \mathbf{u}_d^{(t)} + \eta_t |\widehat{\mathbf{z}}^{(t)}[d]|\left[\mathrm{e}^{\mathrm{i}\left(\phi_{\widehat{\mathbf{z}}^{(t)}[d]} - \phi_{\widehat{\overline{\mathbf{z}}}^{\infty}[d]}\right)}\widehat{\mathbf{w}}_2^{(t)*}[d] + \mathrm{e}^{-\mathrm{i}\left(\phi_{\widehat{\mathbf{z}}^{(t)}[d]} - \phi_{\widehat{\overline{\mathbf{z}}}^{\infty}[d]}\right)} \cdot \widehat{\mathbf{w}}_1^{(t)}[d] \cdot \mathrm{e}^{-\mathrm{i}\phi_{\widehat{\overline{\mathbf{z}}}^{\infty}[d]}}\right]$$

Additionally, since $e^{i\phi_{\hat{\mathbf{z}}^{(t)}[d]}} \to e^{i\phi_{\hat{\bar{\mathbf{z}}}^\infty[d]}}$, we can write $e^{\pm i\left(\phi_{\hat{\mathbf{z}}^{(t)}[d]} - \phi_{\hat{\bar{\mathbf{z}}}^\infty[d]}\right)} = 1 + \boldsymbol{\delta}_{1,d}^{(t)} \pm i\boldsymbol{\delta}_{2,d}^{(t)}$ where $\boldsymbol{\delta}_{1,d}^{(t)}, \boldsymbol{\delta}_{2,d}^{(t)} \to 0$ are real scalars. Substituting in above equation and rearranging the terms, we have

$$
\begin{aligned}
\mathbf{u}_d^{(t+1)} &= \left[1 + \eta_t |\hat{\mathbf{z}}^{(t)}[d]|(1 + \boldsymbol{\delta}_{1,d}^{(t)})\right] \mathbf{u}_d^{(t)} + i\boldsymbol{\delta}_{2,d}^{(t)} \eta_t |\hat{\mathbf{z}}^{(t)}[d]| \left[\widehat{\mathbf{w}}_2^{(t)*}[d] - \widehat{\mathbf{w}}_1^{(t)}[d] \cdot e^{-i\phi_{\hat{\bar{\mathbf{z}}}^\infty[d]}}\right] \\
&\overset{(a)}{:=} \left[1 + \eta_t |\hat{\mathbf{z}}^{(t)}[d]|(1 + \boldsymbol{\delta}_{1,d}^{(t)})\right] \mathbf{u}_d^{(t)} + \eta_t |\hat{\mathbf{z}}^{(t)}[d]| \boldsymbol{\tau}_d^{(t)},
\end{aligned}
\tag{65}
$$

where in $(a)$ we define $\boldsymbol{\tau}_d^{(t)} = i\boldsymbol{\delta}_{2,d}^{(t)} \left[\widehat{\mathbf{w}}_2^{(t)*}[d] - \widehat{\mathbf{w}}_1^{(t)}[d] \cdot e^{-i\phi_{\hat{\bar{\mathbf{z}}}^\infty[d]}}\right]$.

The following intermediate lemma is proved in Appendix C.1.3.

**Lemma 10.** *Consider $\boldsymbol{\tau}_d^{(t)}$ in eq. (65). For all $d$ such that $\hat{\bar{\mathbf{z}}}^\infty[d] \neq 0$, $\mathbf{u}_d^{(t)} \to \infty$ and $\frac{\boldsymbol{\tau}_d^{(t)}}{\mathbf{u}_d^{(t)}} \to 0$.*

Using the above lemma, we have $\boldsymbol{\delta}_{3,d}^{(t)} \to 0$ such that $\boldsymbol{\tau}_d^{(t)} = \boldsymbol{\delta}_{3,d}^{(t)} \mathbf{u}_d(t)$. Additionally, since $\frac{|\hat{\mathbf{z}}^{(t)}[d]|}{p(t)} \to |\hat{\bar{\mathbf{z}}}^\infty[d]|$, there exists $\boldsymbol{\delta}_{4,d}^{(t)} \to 0$ such that $|\hat{\mathbf{z}}^{(t)}[d]| = |\hat{\bar{\mathbf{z}}}^\infty[d]|p(t) + \boldsymbol{\delta}_{4,d}^{(t)} p(t)$. Substituting these representations in eq. (65), we have the following dynamics for $\mathbf{u}_d(t)$,

$$
\begin{aligned}
\mathbf{u}_d^{(t+1)} &= \left[1 + \eta_t p(t) \left(|\hat{\bar{\mathbf{z}}}^\infty[d]| + \boldsymbol{\delta}_{4,d}^{(t)}\right)\left(1 + \boldsymbol{\delta}_{1,d}^{(t)} + \boldsymbol{\delta}_{3,d}^{(t)}\right)\right] \mathbf{u}_d^{(t)} \\
&\overset{(a)}{:=} \left[1 + \eta_t p(t) \left(|\hat{\bar{\mathbf{z}}}^\infty[d]| + \boldsymbol{\delta}_d^{(t)}\right)\right] \mathbf{u}_d^{(t)},
\end{aligned}
\tag{66}
$$

where in $(a)$ we have accumulated all diminishing terms into $\boldsymbol{\delta}_d^{(t)} = \boldsymbol{\delta}_{4,d}^{(t)}\left(1 + \boldsymbol{\delta}_{1,d}^{(t)} + \boldsymbol{\delta}_{3,d}^{(t)}\right) + |\hat{\bar{\mathbf{z}}}^\infty[d]|\left(\boldsymbol{\delta}_{1,d}^{(t)} + \boldsymbol{\delta}_{3,d}^{(t)}\right) \to 0$.

Step 2. *Remainder of the proof:* We now prove our theorem by looking the following quantity: For any $d, d'$ with $\hat{\bar{\mathbf{z}}}^\infty[d], \hat{\bar{\mathbf{z}}}^\infty[d'] \neq 0$, define $\kappa_{d,d'}^{(t)} = \left|\frac{\mathbf{u}_d^{(t)}}{\mathbf{u}_{d'}^{(t)}}\right|$.

We will show that whenever $|\hat{\bar{\mathbf{z}}}^\infty[d]| > |\hat{\bar{\mathbf{z}}}^\infty[d']|$, we get $\kappa_{d,d'}^{(t)} \to \infty$. Along with eq. (64), this would imply that $\lim_{t\to\infty} \kappa_{d,d'}^{(t)} = \sqrt{\frac{|\hat{\bar{\boldsymbol{\beta}}}^\infty[d]|}{|\hat{\bar{\boldsymbol{\beta}}}^\infty[d']|}} = \infty$. Hence, for any $d, d'$ with $\hat{\bar{\boldsymbol{\beta}}}^\infty[d] = 0$ and $\hat{\bar{\boldsymbol{\beta}}}^\infty[d'] \neq 0$, we have $\bar{\gamma}|\hat{\bar{\mathbf{z}}}^\infty[d]| \leq \bar{\gamma}|\hat{\bar{\mathbf{z}}}^\infty[d']|$. Moreover from eq.(57)), we know that $\bar{\gamma}|\hat{\bar{\mathbf{z}}}^\infty[d']| = 1$ for all $d'$ with $\hat{\bar{\boldsymbol{\beta}}}^\infty[d'] \neq 0$. This implies $\forall d, \bar{\gamma}|\hat{\bar{\mathbf{z}}}^\infty[d]| \leq 1$ and concludes the proof.

*Showing $|\hat{\bar{\mathbf{z}}}^\infty[d]| > |\hat{\bar{\mathbf{z}}}^\infty[d']| \implies \kappa_{d,d'}^{(t)} \to \infty$:*

For any $2\epsilon > 0$, let $|\hat{\bar{\mathbf{z}}}^\infty[d]| - |\hat{\bar{\mathbf{z}}}^\infty[d']| = 2\epsilon > 0$. We note that the since the loss $\mathcal{L}(\boldsymbol{\beta}^{(t)}) \to 0$, norm of the gradient $p(t) = \|\mathbf{z}^{(t)}\| = \|\hat{\mathbf{z}}^t\| \to 0$. Hence, for any finite step size sequence $\{\eta_t\}$, there exists $t_1$ such that $\forall t \geq t_1$ and $\forall d$, $\eta_t p(t) \left(|\hat{\bar{\mathbf{z}}}^\infty[d]| + |\boldsymbol{\delta}_d^{(t)}|\right) < 0.5$ and the following inequalities hold,

$$
\kappa_{d,d'}^{(t+1)} = \left|\frac{\mathbf{u}_d^{(t+1)}}{\mathbf{u}_{d'}^{(t+1)}}\right| = \left|\frac{\left(1 + \eta_t\left(|\hat{\bar{\mathbf{z}}}^\infty[d]| + \boldsymbol{\delta}_d^{(t)}\right)p(t)\right) \mathbf{u}_d^{(t)}}{\left(1 + \eta_t\left(|\hat{\bar{\mathbf{z}}}^\infty[d']| + \boldsymbol{\delta}_d^{(t)}\right)p(t)\right) \mathbf{u}_{d'}^{(t)}}\right| \tag{67}
$$

$$
\geq \frac{\left(1 + \eta_t\left(|\hat{\bar{\mathbf{z}}}^\infty[d]| - |\boldsymbol{\delta}_d^{(t)}|\right)p(t)\right)}{\left(1 + \eta_t\left(|\hat{\bar{\mathbf{z}}}^\infty[d']| + |\boldsymbol{\delta}_{d'}^{(t)}|\right)p(t)\right)} \kappa_{d,d'}^{(t)} \tag{68}
$$

$$
\overset{(a)}{\geq} \left(1 + \eta_t\left(|\hat{\bar{\mathbf{z}}}^\infty[d]| - |\boldsymbol{\delta}_d^{(t)}|\right)p(t)\right)\left(1 - \eta_t\left(|\hat{\bar{\mathbf{z}}}^\infty[d']| + |\boldsymbol{\delta}_{d'}^{(t)}|\right)p(t)\right) \kappa_{d,d'}^{(t)} \tag{69}
$$

$$
\overset{(c)}{\geq} \left(1 + \eta_t\left(2\epsilon + \boldsymbol{\delta}_{d,d'}^{(t)}\right)p(t)\right) \kappa_{d,d'}^{(t)}, \tag{70}
$$

where in $(a)$ follows from using $1/(1+x) \geq (1-x)$ for $x < 1$ since $\eta_t p(t)\left(|\widehat{\overline{\mathbf{z}}}^\infty[d]| + |\boldsymbol{\delta}_d^{(t)}|\right) < 0.5$ for all $t \geq t_1$, and in $(c)$, we absorbed all $o(p(t))$ terms as $\boldsymbol{\delta}_{d,d'}^{(t)} p(t)$ for $\boldsymbol{\delta}_{d,d'}^{(t)} \to 0$ and used $|\widehat{\overline{\mathbf{z}}}^\infty[d]| - |\widehat{\overline{\mathbf{z}}}^\infty[d']| = 2\epsilon > 0$.

Since $\boldsymbol{\delta}_{d,d'}^{(t)} \to 0$, for large enough $t_2$ and $t \geq t_2$, we have $|\boldsymbol{\delta}_{d,d'}^{(t)}| < \epsilon$. Thus, for all $t \geq \max\{t_1, t_2\}$,

$$\kappa_{d,d'}^{(t+1)} \geq (1 + \eta_t \epsilon p(t))\, \kappa_{d,d'}^{(t)}. \tag{71}$$

Further, from the conditions of the theorem, for almost all initializations, $|\widehat{\mathbf{w}}_l^{(0)}[d]| > 0$ for all $d$. For step sizes $\{\eta_t\}$ smaller than the local Lipschitz constant, for all finite $t' < \infty$, we also have $|\mathbf{w}_l^{(t')}[d]| > 0$. Moreover from Lemma 10, we have that $|\mathbf{u}_d^{(t)}|, |\mathbf{u}_{d'}^{(t)}| \to \infty$ and hence $\exists t_3$ such that $\forall t \geq t_3, |\mathbf{u}_d^{(t)}| > 0$, but for any finite $t' < \infty, |\mathbf{u}_{d'}^{(t')}| < \infty$. Thus, for $t_0 = \max\{t_1, t_2, t_3\}$, using the above observations, we have that $\kappa_{d,d'}^{(t_0)} = \left|\dfrac{\mathbf{u}_d^{(t_0)}}{\mathbf{u}_{d'}^{(t_0)}}\right| > 0$.

Now, using eq. (71), for all $t \geq t_0$,

$$\kappa_{d,d'}^{(t+1)} \geq (1 + \eta_t \epsilon p(t))\kappa_{d,d'}^{(t)} = \left(\prod_{u=t_0}^{t}(1 + \eta_u \epsilon p(u))\right)\kappa_{d,d'}^{(t_0)} \text{ and } \kappa_{d,d'}^{(t_0)} > 0. \tag{72}$$

Finally, we show the following claim:

**Claim 2.** *For any finite $t_0$, finite step-sizes $\{\eta_t\}$, and any $\epsilon > 0$, we have $\prod_{u=t_0}^{t}(1 + \eta_u \epsilon p(u)) \to \infty$.*

*Proof.* Let $\mu = \max_d |\widehat{\overline{\mathbf{z}}}^\infty[d]| + \max_{t>t_0} |\boldsymbol{\delta}_d^{(t)}| < \infty$. From eq. (66), we have that for all $d$,

$$|\mathbf{u}_d^{(t+1)}| \leq (1 + \mu\eta_t p(t))|\mathbf{u}_d^{(t)}| \leq |\mathbf{u}_d^{(t_0)}|\prod_{u=t_0}^{t}(1 + \mu\eta_u p(u)) \leq |\mathbf{u}_d^{(t_0)}|\exp\left(\sum_{u=t_0}^{t}\mu\eta_u p(u)\right).$$

Moreover, we have $\mathbf{u}_d^{(t)} \to \infty$ for at least one $d$, and for any finite step sizes and finite $t_0$, $|\mathbf{u}_d^{(t_0)}| < \infty$. This then implies that for some $\mu < \infty$, $\exp\left(\sum_{u=t_0}^{t}\mu\eta_u p(u)\right) \to \infty \implies \sum_{u=t_0}^{t}\eta_u p(u) \to \infty$. Thus, for any $\epsilon > 0$, we also have $\prod_{u=t_0}^{t}(1 + \epsilon\eta_u p(u)) \geq \epsilon\sum_{u=t_0}^{t}\eta_u p(u) \to \infty$. $\qquad\square$

From eq. (72) and above claim, we conclude that for all $d, d'$, if $|\widehat{\overline{\mathbf{z}}}^\infty[d]| > |\widehat{\overline{\mathbf{z}}}^\infty[d']|$, then $\kappa_{d,d'}^{(t)} \to \infty$.

This completes the proof of the theorem. $\qquad\square$

### C.1.3 Proof of Lemma 10

**Lemma 10.** *Consider $\boldsymbol{\tau}_d^{(t)}$ in eq. (65). For all $d$ such that $\widehat{\overline{\mathbf{z}}}^\infty[d] \neq 0$, $\mathbf{u}_d^{(t)} \to \infty$ and $\dfrac{\boldsymbol{\tau}_d^{(t)}}{\mathbf{u}_d^{(t)}} \to 0$.*

*Proof.* Recalling $\boldsymbol{\tau}_d^{(t)}$ from eq. (65) and $\mathbf{u}_d^{(t)}$ from eq. (63), we have the following:

$$\frac{\boldsymbol{\tau}_d^{(t)}}{\mathbf{u}_d^{(t)}} = \mathrm{i}\boldsymbol{\delta}_{2,d}^{(t)}\frac{\widehat{\mathbf{w}}_2^{(t)*}[d] - \widehat{\mathbf{w}}_1^{(t)}[d]\cdot \mathrm{e}^{-\mathrm{i}\phi_{\widehat{\overline{\mathbf{z}}}^\infty}[d]}}{\widehat{\mathbf{w}}_1^{(t)}[d]\cdot \mathrm{e}^{-\mathrm{i}\phi_{\widehat{\overline{\mathbf{z}}}^\infty}[d]} + \widehat{\mathbf{w}}_2^{(t)*}[d]} = \mathrm{i}\boldsymbol{\delta}_{2,d}^{(t)}\frac{1 - \frac{|\widehat{\mathbf{w}}_1^{(t)}[d]|}{|\widehat{\mathbf{w}}_2^{(t)}[d]|}\cdot \mathrm{e}^{-\mathrm{i}\phi_{\widehat{\overline{\mathbf{z}}}^\infty}[d]+\mathrm{i}\phi_{\widehat{\boldsymbol{\beta}}^{(t)}}[d]}}{1 + \frac{|\widehat{\mathbf{w}}_1^{(t)}[d]|}{|\widehat{\mathbf{w}}_2^{(t)}[d]|}\cdot \mathrm{e}^{-\mathrm{i}\phi_{\widehat{\overline{\mathbf{z}}}^\infty}[d]+\mathrm{i}\phi_{\widehat{\boldsymbol{\beta}}^{(t)}}[d]}} \tag{73}$$

For all $d$ if $\widehat{\overline{\boldsymbol{\beta}}}^\infty[d] = \widehat{\overline{\mathbf{w}}}_1^\infty[d]\cdot\widehat{\overline{\mathbf{w}}}_2^\infty[d] \neq 0$, the it is straightforward to see that $\dfrac{|\widehat{\mathbf{w}}_1^{(t)}[d]|}{|\widehat{\mathbf{w}}_2^{(t)}[d]|} = \dfrac{|\widehat{\mathbf{w}}_1^{(t)}[d]|/g(t)}{|\widehat{\mathbf{w}}_2^{(t)}[d]|/g(t)} \to \dfrac{|\widehat{\overline{\mathbf{w}}}_1^\infty[d]|}{|\widehat{\overline{\mathbf{w}}}_2^\infty[d]|} = 1$ (from eq. (60)), and also that $\mathrm{e}^{-\mathrm{i}\phi_{\widehat{\overline{\mathbf{z}}}^\infty}[d]+\mathrm{i}\phi_{\widehat{\boldsymbol{\beta}}^{(t)}}[d]} \to \mathrm{e}^{-\mathrm{i}\phi_{\widehat{\overline{\mathbf{z}}}^\infty}[d]+\mathrm{i}\phi_{\widehat{\overline{\boldsymbol{\beta}}}^\infty}[d]} = 1$ (from eq. (62)). This along with eq. (73) gives us $\dfrac{\boldsymbol{\tau}_d^{(t)}}{\mathbf{u}_d^{(t)}} \to 0$.

Moreover, since $|\widehat{\boldsymbol{\beta}}^{(t)}[d]| \to \infty$, we have $|\widehat{\mathbf{w}}_2^{(t)}[d]|$ or $|\widehat{\mathbf{w}}_2^{(t)}[d]| \to \infty$. Further, using $e^{-i\phi_{\widehat{\mathbf{z}}^\infty}[d]+i\phi_{\widehat{\boldsymbol{\beta}}^{(t)}[d]}} \to 1$, we have $|\mathbf{u}_d^{(t)}| = |\widehat{\mathbf{w}}_2^{(t)}[d]| + |\widehat{\mathbf{w}}_1^{(t)}[d]|e^{-i\phi_{\widehat{\mathbf{z}}^\infty}[d]+i\phi_{\widehat{\boldsymbol{\beta}}^{(t)}[d]}} \to \infty$.

We now only need to show that these results also hold for $d$ such that $\overline{\widehat{\boldsymbol{\beta}}}^\infty[d] = 0$. Recall from the assumptions of the theorem that even when $\overline{\widehat{\boldsymbol{\beta}}}^\infty[d] = 0$, $\exists \phi_{\widehat{\boldsymbol{\beta}}^\infty_{[d]}} \in [0, 2\pi)$ such that $e^{i\phi_{\widehat{\boldsymbol{\beta}}^{(t)}[d]}} \to e^{i\phi_{\widehat{\boldsymbol{\beta}}^\infty_{[d]}}}$. We now prove the lemma by showing the following steps for $d$ such that $\overline{\widehat{\boldsymbol{\beta}}}^\infty[d] = 0$. :

Step 1. Show $\dfrac{|\widehat{\mathbf{w}}_1^{(t)}[d]|}{|\widehat{\mathbf{w}}_2^{(t)}[d]|} \to 1$.

Step 2. Show $\mathrm{Re}(e^{-i\phi_{\widehat{\mathbf{z}}^\infty}[d]+i\phi_{\widehat{\boldsymbol{\beta}}^\infty}[d]}) = 2\cos\left(\phi_{\widehat{\mathbf{z}}^\infty_{[d]}} - \phi_{\widehat{\boldsymbol{\beta}}^\infty_{[d]}}\right) \geq 0$.

**Proof of lemma assuming Step 1 and Step 2 hold** The above steps would imply that in eq. (73),

- the denominator satisfies

$$
\left|1 + \frac{|\widehat{\mathbf{w}}_1^{(t)}[d]|}{|\widehat{\mathbf{w}}_2^{(t)}[d]|} \cdot e^{-i\phi_{\widehat{\mathbf{z}}^\infty}[d]+i\phi_{\widehat{\boldsymbol{\beta}}^{(t)}[d]}}\right| \to \left|1 + e^{-i\phi_{\widehat{\mathbf{z}}^\infty}[d]+i\phi_{\widehat{\boldsymbol{\beta}}^\infty}[d]}\right|
$$
$$
\geq \left|1 + \mathrm{Re}(e^{-i\phi_{\widehat{\mathbf{z}}^\infty}[d]+i\phi_{\widehat{\boldsymbol{\beta}}^\infty}[d]})\right| \geq 1.
$$

(74)

- the numerator satisfies

$$
\left|\boldsymbol{\delta}_{2,d}^{(t)}\left(1 - \frac{|\widehat{\mathbf{w}}_1^{(t)}[d]|}{|\widehat{\mathbf{w}}_2^{(t)}[d]|} \cdot e^{-i\phi_{\widehat{\mathbf{z}}^\infty}[d]+i\phi_{\widehat{\boldsymbol{\beta}}^{(t)}[d]}}\right)\right| \leq |\boldsymbol{\delta}_{2,d}^{(t)}|\left|1 + \frac{|\widehat{\mathbf{w}}_1^{(t)}[d]|}{|\widehat{\mathbf{w}}_2^{(t)}[d]|}\right| \to 0. \qquad (75)
$$

These eqs. along with eq. (73) in turn prove the lemma, *i.e.*, $\dfrac{\boldsymbol{\tau}_d^{(t)}}{\mathbf{u}_d^{(t)}} \to 0$ and $|\mathbf{u}_d^{(t)}| \to \infty$.

**Showing Step 1 and Step 2**

**Step** 1. *Show* $\dfrac{|\widehat{\mathbf{w}}_1^{(t)}[d]|}{|\widehat{\mathbf{w}}_2^{(t)}[d]|} \to 1$.

From the dynamics of $\widehat{\mathbf{w}}_l^{(t)}[d]$ from eq. (59), we have the following,

$$
|\widehat{\mathbf{w}}_1^{(t+1)}[d]|^2 = |\widehat{\mathbf{w}}_1^{(t)}[d]|^2 + \eta_t\widehat{\mathbf{z}}^{(t)}[d]\cdot\widehat{\boldsymbol{\beta}}^{(t)*}[d] + \eta_t\widehat{\mathbf{z}}^{(t)*}[d]\cdot\widehat{\boldsymbol{\beta}}^{(t)}[d] + \eta_t^2|\widehat{\mathbf{z}}^{(t)}[d]|^2|\widehat{\mathbf{w}}_2^{(t)}[d]|^2
$$
$$
|\widehat{\mathbf{w}}_2^{(t+1)}[d]|^2 = |\widehat{\mathbf{w}}_2^{(t)}[d]|^2 + \eta_t\widehat{\mathbf{z}}^{(t)}[d]\cdot\widehat{\boldsymbol{\beta}}^{(t)*}[d] + \eta_t\widehat{\mathbf{z}}^{(t)*}[d]\cdot\widehat{\boldsymbol{\beta}}^{(t)}[d] + \eta_t^2|\widehat{\mathbf{z}}^{(t)}[d]|^2|\widehat{\mathbf{w}}_1^{(t)}[d]|^2
$$

(76)

Note that since $|\widehat{\mathbf{z}}^{(t)}[d]|^2 \to 0$ and $\eta_t$ are finite, we have that $\exists t_1$ such that for all $t \geq t_1$, $\eta_t|\widehat{\mathbf{z}}^{(t)}[d]|^2 \leq 1$. From the above equation, we have the following for $t \geq t_1$,

$$
\left||\widehat{\mathbf{w}}_1^{(t+1)}[d]|^2 - |\widehat{\mathbf{w}}_2^{(t+1)}[d]|^2\right| = \left|\left(1 - \eta_t^2|\widehat{\mathbf{z}}^{(t)}[d]|^2\right)\left(|\widehat{\mathbf{w}}_1^{(t)}[d]|^2 - \widehat{\mathbf{w}}_2^{(t)}[d]|^2\right)\right|
$$
$$
\overset{(a)}{=} \left(\prod_{u=t_1}^{t}\left(1 - \eta_u^2|\widehat{\mathbf{z}}^{(u)}[d]|^2\right)\right)\left||\widehat{\mathbf{w}}_1^{(t_1)}[d]|^2 - \widehat{\mathbf{w}}_2^{(t_1)}[d]|^2\right| \qquad (77)
$$
$$
\leq \left||\widehat{\mathbf{w}}_1^{(t_1)}[d]|^2 - \widehat{\mathbf{w}}_2^{(t_1)}[d]|^2\right| < \infty,
$$

where $(a)$ follows from iterating over $t$ and using $|\widehat{\mathbf{z}}^{(t)}[d]|^2 \leq 1$ for $t \geq t_1$.

Since $|\widehat{\boldsymbol{\beta}}^{(t)}[d]| = |\widehat{\mathbf{w}}_1^{(t)}[d]| \cdot |\widehat{\mathbf{w}}_2^{(t)}[d]| \to \infty$, at least one of $|\widehat{\mathbf{w}}_1^{(t)}[d]|, |\widehat{\mathbf{w}}_2^{(t)}[d]|$ must diverge. Without loss of generality, let $|\widehat{\mathbf{w}}_2^{(t)}[d]| \to \infty$. Let $c(t) := |\widehat{\mathbf{w}}_1^{(t)}[d]|^2 - |\widehat{\mathbf{w}}_2^{(t)}[d]|^2$ with $|c(t)| < \infty$. We have

$$
\frac{|\widehat{\mathbf{w}}_1^{(t)}[d]|^2}{|\widehat{\mathbf{w}}_2^{(t)}[d]|^2} = 1 + \frac{c(t)}{|\widehat{\mathbf{w}}_2^{(t)}[d]|^2} \overset{(a)}{\to} 1, \qquad (78)
$$

where the convergence in $(a)$ follows since $|c(t)| < \infty$ (from eq. (76)) and $|\widehat{\mathbf{w}}_2^{(t)}[d]| \to \infty$.

**Step** 2. *Show* $Re(e^{-i\phi_{\widehat{\overline{\mathbf{z}}}^\infty[d]} + i\phi_{\widehat{\overline{\boldsymbol{\beta}}}^\infty[d]}}) = 2\cos\left(\phi_{\widehat{\overline{\boldsymbol{\beta}}}^\infty[d]} - \phi_{\widehat{\overline{\mathbf{z}}}^\infty[d]}\right) \geq 0.$

Note that from Step 1 above, we have that $\frac{|\widehat{\mathbf{w}}_1^{(t)}[d]|^2}{|\widehat{\mathbf{w}}_2^{(t)}[d]|^2} \to 1$, which implies $\frac{|\widehat{\mathbf{w}}_1^{(t)}[d]|^2 + |\widehat{\mathbf{w}}_2^{(t)}[d]|^2}{2|\widehat{\boldsymbol{\beta}}^{(t)}[d]|} =$
$\frac{|\widehat{\mathbf{w}}_1^{(t)}[d]|^2 + |\widehat{\mathbf{w}}_2^{(t)}[d]|^2}{2|\widehat{\mathbf{w}}_1^{(t)}[d]| \cdot |\widehat{\mathbf{w}}_2^{(t)}[d]|} \to 1$. Thus, there exists $\boldsymbol{\delta}_{1,d}^{(t)} \to 0$, such that

$$|\widehat{\mathbf{w}}_1^{(t)}[d]|^2 + |\widehat{\mathbf{w}}_2^{(t)}[d]|^2 = 2|\widehat{\boldsymbol{\beta}}^{(t)}[d]| \cdot (1 + \boldsymbol{\delta}_{1,d}^{(t)}). \tag{79}$$

Also, from eq. (44), there exists $\boldsymbol{\delta}_{2,d}^{(t)} \to 0$, such that

$$\widehat{\mathbf{z}}^{(t)}[d] = \widehat{\overline{\mathbf{z}}}^\infty[d]p(t) + \boldsymbol{\delta}_{2,d}^{(t)}p(t), \text{ with } p(t) = \|\widehat{\mathbf{z}}^{(t)}\| \to 0. \tag{80}$$

Using the above representations, along with eq. (59), we have the following,

$$\widehat{\boldsymbol{\beta}}^{(t+1)}[d] = \widehat{\boldsymbol{\beta}}^{(t)}[d] + \eta_t \widehat{\mathbf{z}}^{(t)}[d]\left[|\widehat{\mathbf{w}}_1^{(t)}[d]|^2 + |\widehat{\mathbf{w}}_2^{(t)}[d]|^2 + \eta_t \widehat{\mathbf{z}}^{(t)}[d] \cdot \widehat{\boldsymbol{\beta}}^{(t)*}[d]\right]$$

$$\overset{(a)}{=} \widehat{\boldsymbol{\beta}}^{(t)}[d] + 2\eta_t p(t)|\widehat{\boldsymbol{\beta}}^{(t)}[d]|\left(\widehat{\overline{\mathbf{z}}}^\infty[d] + \boldsymbol{\delta}_{2,d}^{(t)}\right)\left[1 + \boldsymbol{\delta}_{1,d}^{(t)} + 1/2\eta_t \widehat{\mathbf{z}}^{(t)}[d]e^{-i\phi_{\widehat{\boldsymbol{\beta}}^{(t)}[d]}}\right]$$

$$\overset{(b)}{:=} \widehat{\boldsymbol{\beta}}^{(t)}[d] + 2\eta_t p(t)|\widehat{\boldsymbol{\beta}}^{(t)}[d]|\left[\widehat{\overline{\mathbf{z}}}^\infty[d] + \boldsymbol{\delta}_{3,d}^{(t)}\right], \tag{81}$$

where $(a)$ follows from substituting eqs. (79)-(80), and $(b)$ follows from using $|\widehat{\mathbf{z}}^{(t)}[d]| \leq p(t) \to 0$ and defining $\boldsymbol{\delta}_{3,d}^{(t)} = \boldsymbol{\delta}_{2,d}^{(t)}\left[1 + \boldsymbol{\delta}_{1,d}^{(t)} + 1/2\eta_t \widehat{\mathbf{z}}^{(t)}[d]e^{-i\phi_{\widehat{\boldsymbol{\beta}}^{(t)}[d]}}\right] + \widehat{\overline{\mathbf{z}}}^\infty[d]\left[\boldsymbol{\delta}_{1,d}^{(t)} + 1/2\eta_t \widehat{\mathbf{z}}^{(t)}[d]e^{-i\phi_{\widehat{\boldsymbol{\beta}}^{(t)}[d]}}\right] \to 0.$

Denote $\Delta_d = \phi_{\widehat{\overline{\boldsymbol{\beta}}}^\infty[d]} - \phi_{\widehat{\overline{\mathbf{z}}}^\infty[d]}$. Additionally, from the assumption in the theorem, we have $e^{i\phi_{\widehat{\boldsymbol{\beta}}^{(t)}[d]}} \to e^{i\phi_{\widehat{\overline{\boldsymbol{\beta}}}^\infty[d]}}$, hence there exists $\boldsymbol{\delta}_{4,d}^{(t)} \to 0$ such that $e^{i\phi_{\widehat{\boldsymbol{\beta}}^{(t)}[d]} - i\phi_{\widehat{\overline{\mathbf{z}}}^\infty[d]}} = e^{i\Delta_d}(1 + \boldsymbol{\delta}_{4,d}^{(t)}).$

Now, from the above equation, for any $t_0$ and $t \geq t_0$, we derive the updates for $|\widehat{\boldsymbol{\beta}}^{(t)}[d]|$,

$$|\widehat{\boldsymbol{\beta}}^{(t+1)}[d]|^2 = |\widehat{\boldsymbol{\beta}}^{(t)}[d]|^2 \left(e^{i\phi_{\widehat{\boldsymbol{\beta}}^{(t)}[d]}} + 2\eta_t p(t)\left[\widehat{\overline{\mathbf{z}}}^\infty[d] + \boldsymbol{\delta}_{3,d}^{(t)}\right]\right)\left(e^{-i\phi_{\widehat{\boldsymbol{\beta}}^{(t)}[d]}} + 2\eta_t p(t)\left[\widehat{\overline{\mathbf{z}}}^{\infty*}[d] + \boldsymbol{\delta}_{3,d}^{(t)*}\right]\right)$$

$$\overset{(a)}{=} |\widehat{\boldsymbol{\beta}}^{(t)}[d]|^2\left[1 + 2\eta_t p(t)\left(|\widehat{\overline{\mathbf{z}}}^\infty[d]|\left(e^{i\Delta_d}(1 + \boldsymbol{\delta}_{4,d}^{(t)}) + e^{-i\Delta_d}(1 + \boldsymbol{\delta}_{4,d}^{(t)*})\right) + \boldsymbol{\delta}_{5,d}^{(t)}\right)\right]$$

$$\overset{(b)}{=} |\widehat{\boldsymbol{\beta}}^{(t)}[d]|^2\left[1 + 4\eta_t p(t)\left(|\widehat{\overline{\mathbf{z}}}^\infty[d]|\cos(\Delta_d) + \boldsymbol{\delta}_{6,d}^{(t)}\right)\right]$$

$$\overset{(c)}{=} |\widehat{\boldsymbol{\beta}}^{(t_0)}[d]|^2\left[\prod_{u=t_0}^{t}\left(1 + 4\eta_u p(u)\left(|\widehat{\overline{\mathbf{z}}}^\infty[d]|\cos(\Delta_d) + \boldsymbol{\delta}_{6,d}^{(u)}\right)\right)\right]$$

$$\overset{(d)}{\leq} |\widehat{\boldsymbol{\beta}}^{(t_0)}[d]|^2 \exp\left(\sum_{u=t_0}^{t} 4\eta_u p(u)\left(|\widehat{\overline{\mathbf{z}}}^\infty[d]|\cos(\Delta_d) + \boldsymbol{\delta}_{6,d}^{(u)}\right)\right), \tag{82}$$

where in $(a)$ we used $e^{i\phi_{\widehat{\boldsymbol{\beta}}^{(t)}[d]} - i\phi_{\widehat{\overline{\mathbf{z}}}^\infty[d]}} = e^{i\Delta_d}(1 + \boldsymbol{\delta}_{4,d}^{(t)})$ and collected all $o(p(t))$ terms into $\boldsymbol{\delta}_{5,d}^{(t)} = 1/2e^{i\phi_{\widehat{\boldsymbol{\beta}}^{(t)}[d]}}\boldsymbol{\delta}_{3,d}^{(t)*} + 2p(t)\widehat{\overline{\mathbf{z}}}^\infty[d]\left[\widehat{\overline{\mathbf{z}}}^{\infty*}[d] + \boldsymbol{\delta}_{3,d}^{(t)*}\right] + \boldsymbol{\delta}_{3,d}^{(t)}\left(e^{-i\phi_{\widehat{\boldsymbol{\beta}}^{(t)}[d]}} + 2p(t)\left[\widehat{\overline{\mathbf{z}}}^{\infty*}[d] + \boldsymbol{\delta}_{3,d}^{(t)*}\right]\right) \to 0$ (since $p(t), \boldsymbol{\delta}_{3,d}^{(t)} \to 0$); in $(b)$ we defined $\boldsymbol{\delta}_{6,d}^{(t)} = 1/2\boldsymbol{\delta}_{4,d}^{(t)*}e^{i\Delta_d} + 1/2\boldsymbol{\delta}_{3,d}^{(t)}e^{-i\Delta_d} + \boldsymbol{\delta}_{5,d}^{(t)} \to 0$; $(c)$ is obtained by iterating over $t$; and $(d)$ follows from using $(1 + x) \leq \exp(x)$.

If possible, let $\cos(\Delta_d) = -2\epsilon < 0$. Since $|\boldsymbol{\delta}_{6,d}^{(t)}| \to 0$, and for finite step sizes $\eta_t p(t) \to 0$, $\exists t_0$ such that for all $t \geq t_0$, $|\boldsymbol{\delta}_{6,d}^{(t)}| < \epsilon|\widehat{\overline{\mathbf{z}}}^\infty[d]|$ and $\exp\left(-4\epsilon|\widehat{\overline{\mathbf{z}}}^\infty[d]|\eta_t p(t)\right) \leq 1$. From eq. (82), we now have

$$|\widehat{\boldsymbol{\beta}}^{(t+1)}[d]|^2 \leq |\widehat{\boldsymbol{\beta}}^{(t_0)}[d]|^2 \exp\left(-4\epsilon|\widehat{\overline{\mathbf{z}}}^\infty[d]|\sum_{u=t_0}^{t}\eta_u p(u)\right) \leq |\widehat{\boldsymbol{\beta}}^{(t_0)}[d]|^2.$$

Finally, for any finite step sizes and finite $t_0$, we have $|\widehat{\beta}^{(t_0)}[d]|^2 < \infty$ and this creates a contradiction since the LHS in the above equation diverges, $|\widehat{\beta}^{(t+1)}[d]|^2 \to \infty$. Hence, in order for the updates in eq. (82) to lead to a divergent $|\widehat{\beta}^{(t+1)}[d]|$, we necessarily require that $\cos(e^{i\Delta_d}) = \text{Re}(e^{-i\phi_{\widehat{\mathbf{z}}^\infty[d]} + i\phi_{\widehat{\beta}^\infty[d]}}) = 2\cos\left(\phi_{\widehat{\mathbf{z}}^\infty[d]} - \phi_{\widehat{\beta}^\infty[d]}\right) \geq 0.$

This completes the proof of the lemma. $\qquad\square$

## D  Computing $\mathcal{R}_\mathcal{P}(\beta)$: Proofs of Lemmas in Section 5

In this appendix we prove the lemmas in Section 5 that compute the form of induced bias of linear networks in the space of predictors. Recall that for linear predictors parameterized as $\beta = \mathcal{P}(\mathbf{w})$, $\mathcal{R}_\mathcal{P}(\beta) = \min_{\mathbf{w}:\mathcal{P}(\mathbf{w})=\beta}\|\mathbf{w}\|_2^2$.

**Lemma 5.** *For fully connected networks of any depth $L > 0$,*

$$\mathcal{R}_{\mathcal{P}_{full}}(\beta) = \min_{\mathbf{w}:\mathcal{P}_{full}(\mathbf{w})=\beta}\|\mathbf{w}\|_2^2 = L\|\beta\|_2^{2/L} = \textit{monotone}(\|\beta\|_2).$$

*Proof.* Recall that for fully connected networks of any depth $L > 0$ with parameters $\mathbf{w} = [\mathbf{w}_l \in \mathbb{R}^{D_{l-1} \times D_l}]_{l-1}^L$, the equivalent linear predictor given by $\mathcal{P}_{full}(\mathbf{w}) = \mathbf{w}_1\mathbf{w}_2\ldots\mathbf{w}_L$.

We first show that $\mathcal{R}_{\mathcal{P}_{full}}(\beta) \geq L\|\beta\|_2^{2/L}$.
Let $\mathbf{w}^\star(\beta) = [\mathbf{w}_l^\star(\beta)]_{l=1}^L$ be the minimizer of $\min_{\mathbf{w}:\mathcal{P}_{full}(\mathbf{w})=\beta}\|\mathbf{w}\|_2^2$, so that $\beta = \mathcal{P}_{full}(\mathbf{w}^\star(\beta)) = \mathbf{w}_1^\star(\beta) \cdot \mathbf{w}_2^\star(\beta)\ldots\mathbf{w}_L^\star(\beta)$ and $\mathcal{R}_{\mathcal{P}_{full}}(\beta) = \|\mathbf{w}^\star(\beta)\|_2^2 = \sum_{l=1}^L\|\mathbf{w}_l^\star(\beta)\|_2^2$. We then have,

$$\|\beta\|_2^{2/L} = \|\mathbf{w}_1^\star(\beta) \cdot \mathbf{w}_2^\star(\beta)\ldots\mathbf{w}_L^\star(\beta)\|_2^{2/L} \leq \|\mathbf{w}_1^\star(\beta)\|_2^{2/L}\|\mathbf{w}_2^\star(\beta)\|_2^{2/L}\ldots\|\mathbf{w}_L^\star(\beta)\|_2^{2/L}$$

$$\overset{(a)}{\leq} \frac{1}{L}\sum_{l=1}^L\|\mathbf{w}_l^\star(\beta)\|_2^2 = \frac{1}{L}\mathcal{R}_{\mathcal{P}_{full}}(\beta), \tag{83}$$

where $(a)$ follows as arithmetic mean is greater than the geometric mean.

Next, we show that $\mathcal{R}_{\mathcal{P}_{full}}(\beta) \leq L\|\beta\|_2^{2/L}$.
Given any unit norm vectors $\mathbf{z}_l \in \mathbb{R}^{D_l}$ for $l = 1, 2, \ldots, L$, consider $\overline{\mathbf{w}} = [\overline{\mathbf{w}}_l]$, defined as

$$\overline{\mathbf{w}}_l = \begin{cases} \|\beta\|_2^{1/L}\frac{\beta}{\|\beta\|_2}\mathbf{z}_1^\top & \text{if } l = 1 \\ \|\beta\|_2^{1/L}\mathbf{z}_{l-1}\mathbf{z}_l^\top & \text{if } l = 2, 3, \ldots, L-1 \\ \|\beta\|_2^{1/L}\mathbf{z}_{L-1} & \text{if } l = L \end{cases}$$

This ensures that $\mathcal{P}_{full}(\overline{\mathbf{w}}) = \overline{\mathbf{w}}_1\overline{\mathbf{w}}_2\ldots\overline{\mathbf{w}}_L = \beta$ and $\|\overline{\mathbf{w}}\|_2^2 = L\|\beta\|_2^{2/L}$, and hence

$$\mathcal{R}(\beta) = \min_{\mathbf{w}:\mathcal{P}_{full}(\mathbf{w})=\beta}\|\mathbf{w}\|_2^2 \leq \|\overline{\mathbf{w}}\|_2^2 = L\|\beta\|_2^{2/L}. \tag{84}$$

Combining eq. (83) and eq. (84), we get $\mathcal{R}_{\mathcal{P}_{full}}(\beta) = L\|\beta\|_2^{2/L}$ $\qquad\square$

The proofs of the lemmas for computing $\mathcal{R}_\mathcal{P}(\mathbf{w})$ for diagonal and convolutional networks are similar to those of fully connected network.

**Lemma 6.** *For a depth–$L$ diagonal network with parameters $\mathbf{w} = [\mathbf{w}_l \in \mathbb{R}^D]_{l-1}^L$, we have*

$$\mathcal{R}_{\mathcal{P}_{diag}}(\beta) = \min_{\mathbf{w}:\mathcal{P}_{diag}(\mathbf{w})=\beta}\|\mathbf{w}\|_2^2 = L\|\beta\|_{2/L}^{2/L} = \textit{monotone}(\|\beta\|_{2/L}).$$

*Proof.* Recall that for an $L$–layer linear diagonal networks with parameters $\mathbf{w} = [\mathbf{w}_l \in \mathbb{R}^D]_{l-1}^L$, the equivalent linear predictor is given by $\mathcal{P}_{diag}(\mathbf{w}) = \text{diag}(\mathbf{w}_1)\text{diag}(\mathbf{w}_2)\ldots\text{diag}(\mathbf{w}_{L-1})\mathbf{w}_L$.

Let $\mathbf{w}^{\star}(\boldsymbol{\beta}) = [\mathbf{w}_l^{\star}(\boldsymbol{\beta})]_{l=1}^L$ be the minimizer of $\min_{\mathbf{w}:\mathcal{P}_{diag}(\mathbf{w})=\boldsymbol{\beta}} \|\mathbf{w}\|_2^2$, so that $\boldsymbol{\beta} = \mathcal{P}_{diag}(\mathbf{w}^{\star}(\boldsymbol{\beta}))$ and $\mathcal{R}_{\mathcal{P}_{diag}}(\boldsymbol{\beta}) = \|\mathbf{w}^{\star}(\boldsymbol{\beta})\|_2^2$. We then have,

$$\sum_{d=0}^{D-1} |\boldsymbol{\beta}[d]|^{2/L} = \sum_{d=0}^{D-1} \prod_{l=1}^{L} |\mathbf{w}_1^{\star}(\boldsymbol{\beta})[d]|^{2/L} \overset{(a)}{\leq} \frac{1}{L} \sum_{d=0}^{D-1} \sum_{l=1}^{L} |\mathbf{w}_1^{\star}(\boldsymbol{\beta})[d]|^2$$
$$= \frac{1}{L} \|\mathbf{w}^{\star}(\boldsymbol{\beta})\|_2^2 = \frac{1}{L} \mathcal{R}_{\mathcal{P}_{diag}}(\boldsymbol{\beta}), \tag{85}$$

where $(a)$ again follows as arithmetic mean is greater than the geometric mean.

Similar to the case of fully connected networks, we now choose $\overline{\mathbf{w}} = [\overline{\mathbf{w}}_l]$ that satisfies $\mathcal{P}_{diag}(\overline{\mathbf{w}}) = \boldsymbol{\beta}$ and $\|\overline{\mathbf{w}}\|_2^2 = L\|\boldsymbol{\beta}\|_{2/L}^{2/L}$. This would ensure that,

$$\mathcal{R}_{\mathcal{P}_{diag}}(\boldsymbol{\beta}) = \min_{\mathbf{w}:\mathcal{P}_{diag}(\mathbf{w})=\boldsymbol{\beta}} \|\mathbf{w}\|_2^2 \leq \|\overline{\mathbf{w}}\|_2^2 = L\|\boldsymbol{\beta}\|_{2/L}^{2/L}.$$

We can check that these properties are satisfied by choosing $\overline{\mathbf{w}}$ as follows: for $d = 0, 1, \ldots D - 1$, let $\overline{\mathbf{w}}_1[d] = \text{sign}(\boldsymbol{\beta}^{(d)}) |\boldsymbol{\beta}^{(d)}|^{1/L}$ and $\overline{\mathbf{w}}_l[d] = |\boldsymbol{\beta}^{(d)}|^{1/L}$ for $l = 2, 3, \ldots, L$.

Combining this argument with eq. 85 concludes the proof. $\qquad\square$

For convolutional networks, the argument is the exactly the same as that for diagonal network adapted for complex vectors.

**Lemma 7.** *For a depth–$L$ convolutional network with parameters $\mathbf{w} = [\mathbf{w}_l \in \mathbb{R}^D]_{l-1}^L$, we have*

$$\mathcal{R}_{\mathcal{P}_{conv}}(\boldsymbol{\beta}) = \min_{\mathbf{w}:\mathcal{P}_{conv}(\mathbf{w})=\boldsymbol{\beta}} \|\mathbf{w}\|_2^2 = L\|\widehat{\boldsymbol{\beta}}\|_{2/L}^{2/L} = monotone(\|\widehat{\boldsymbol{\beta}}\|_{2/L}).$$

*Proof.* Denote the Fourier basis coefficients of $\mathbf{w}_l \in \mathbb{R}^D$ and $\boldsymbol{\beta} = \mathcal{P}_{conv}(\mathbf{w}) \in \mathbb{R}^D$ in polar form as

$$\widehat{\mathbf{w}}_l = |\widehat{\mathbf{w}}_l| e^{\mathrm{i}\phi_{\widehat{\mathbf{w}}_l}} \in \mathbb{C}^D, \quad \widehat{\boldsymbol{\beta}} = |\widehat{\boldsymbol{\beta}}| e^{\mathrm{i}\phi_{\widehat{\boldsymbol{\beta}}}} \in \mathbb{C}^D,$$

where $|\widehat{\mathbf{w}}_l|, |\widehat{\boldsymbol{\beta}}| \in \mathbb{R}_+^D$ and $\phi_{\widehat{\mathbf{w}}_l}, \phi_{\widehat{\boldsymbol{\beta}}} \in [0, 2\pi)^D$ are the vectors with magnitudes and phases, respectively, of $\widehat{\mathbf{w}}_l, \widehat{\boldsymbol{\beta}}$.

From Lemma 3, the Fourier basis representation of $\boldsymbol{\beta} = \mathcal{P}_{conv}(\mathbf{w})$ is given by

$$\widehat{\boldsymbol{\beta}} = \text{diag}(\widehat{\mathbf{w}}_1)\text{diag}(\widehat{\mathbf{w}}_2)\ldots\text{diag}(\widehat{\mathbf{w}}_{L-1})\widehat{\mathbf{w}}_L = \mathcal{P}_{diag}(\widehat{\mathbf{w}}),$$

where we have overloaded the notation $\mathcal{P}_{diag}$ to denote the mapping of diagonal networks in complex vector fields, and $\widehat{\mathbf{w}} = [\widehat{\mathbf{w}}_l]_{l=1}^L$. We thus have for $d = 0, 1, \ldots, D - 1$,

$$|\widehat{\boldsymbol{\beta}}[d]| = \prod_{l=1}^{L} |\widehat{\mathbf{w}}_l[d]|, \quad \text{and} \quad \phi_{\widehat{\boldsymbol{\beta}}}[d] = \left(\sum_{l=1}^{L} \phi_{\widehat{\mathbf{w}}_l}[d]\right) \mod 2\pi.$$

From orthonormality of discrete Fourier transformation, we have for all $\mathbf{w}$, $\|\mathbf{w}\|_2^2 = \|\widehat{\mathbf{w}}\|_2^2$. Thus,

$$\mathcal{R}_{\mathcal{P}_{conv}}(\boldsymbol{\beta}) = \min_{\mathbf{w}:\mathcal{P}_{conv}(\mathbf{w})=\boldsymbol{\beta}} \|\mathbf{w}\|_2^2 = \min_{\widehat{\boldsymbol{\beta}}:\widehat{\boldsymbol{\beta}}=\mathcal{P}_{\text{diag}}(\widehat{\mathbf{w}})} \|\widehat{\mathbf{w}}\|_2^2. \tag{86}$$

We can now adapt the proof of diagonal networks here. Let $\widehat{\mathbf{w}}^{\star}(\boldsymbol{\beta}) = [\widehat{\mathbf{w}}_l^{\star}(\boldsymbol{\beta}) \in \mathbf{C}^D]_{l=1}^L$ be the minimizer of $\min_{\widehat{\mathbf{w}}:\widehat{\boldsymbol{\beta}}=\mathcal{P}_{\text{diag}}(\widehat{\mathbf{w}})} \|\widehat{\mathbf{w}}\|_2^2$, so that $\widehat{\boldsymbol{\beta}} = \mathcal{P}_{diag}(\widehat{\mathbf{w}}^{\star}(\boldsymbol{\beta}))$ and $\mathcal{R}_{\mathcal{P}_{conv}}(\boldsymbol{\beta}) = \|\widehat{\mathbf{w}}^{\star}(\boldsymbol{\beta})\|_2^2$, and

$$\sum_{d=0}^{D-1} |\widehat{\boldsymbol{\beta}}[d]|^{2/L} = \sum_{d} \prod_{l=1}^{L} |\widehat{\mathbf{w}}_1^{\star}(\boldsymbol{\beta})[d]|^{2/L} \leq \frac{1}{L} \sum_{d} \sum_{l=1}^{L} |\widehat{\mathbf{w}}_1^{\star}(\boldsymbol{\beta})[d]|^2$$
$$= \frac{\|\widehat{\mathbf{w}}^{\star}(\boldsymbol{\beta})\|_2^2}{L} = \frac{1}{L} \mathcal{R}_{\mathcal{P}_{conv}}(\boldsymbol{\beta}). \tag{87}$$

Similar to the diagonal networks, we can choose the parameters in the Fourier domain $\widehat{\overline{\mathbf{w}}} = [\widehat{\overline{\mathbf{w}}}_l \in \mathbb{C}^D]$ to ensure that $\mathcal{P}_{diag}(\widehat{\overline{\mathbf{w}}}) = \widehat{\boldsymbol{\beta}}$ and $\|\widehat{\overline{\mathbf{w}}}\|_2^2 = L\|\widehat{\boldsymbol{\beta}}\|_{2/L}^{2/L}$ as follows: for $d = 0, 1, \ldots D - 1$, let

$$\widehat{\overline{\mathbf{w}}}_1[d] = \phi_{\widehat{\boldsymbol{\beta}}}[d] \, |\widehat{\boldsymbol{\beta}}[d]|^{1/L} \text{ and } \widehat{\overline{\mathbf{w}}}_l[d] = |\widehat{\boldsymbol{\beta}}[d]|^{1/L}, \forall l > 1.$$

This gives us

$$\mathcal{R}_{\mathcal{P}_{conv}}(\boldsymbol{\beta}) = \min_{\mathbf{w}:\mathcal{P}_{diag}(\widehat{\mathbf{w}})=\widehat{\boldsymbol{\beta}}} \|\widehat{\mathbf{w}}\|_2^2 \leq \|\widehat{\overline{\mathbf{w}}}\|_2^2 \leq L\|\widehat{\boldsymbol{\beta}}\|_{2/L}^{2/L}.$$

Combining this with eq. 87 concludes the proof. □

# E   Background Results

**Theorem 11** (Stolz–Cesaro theorem, proof in Theorem 1.22 of Muresan [2009]). *Assume that $\{a_k\}_{k=1}^{\infty}$ and $\{b_k\}_{k=1}^{\infty}$ are two sequences of real numbers such that $\{b_k\}_{k=1}^{\infty}$ is strictly monotonic and diverging (i.e., monotone increasing with $b_k \to \infty$, or monotone decreasing with $b_k \to -\infty$). Additionally, if $\lim_{k\to\infty} \frac{a_{k+1}-a_k}{b_{k+1}-b_k} = L$ exists, then $\lim_{k\to\infty} \frac{a_k}{b_k}$ exists and is equal to L.*

[Supplementary Material 2 · Supplementary-appendix-only.pdf]

# Appendix

The proofs of the theorems in the paper are organized as follows: In Appendix A we first give the proof for Theorem 4, which includes linear fully connected and full width convolutional networks as special cases. This gives us some general results that can be special-cased to prove the stronger results for these networks in Section 3. In Appendix B, we prove Theorem 1 on the implicit bias of fully connected linear networks. In Appendix C, we prove Theorem 2–2a on the implicit bias of linear convolutional networks. Finally, in Appendix D we prove the lemmas in Section 5 on computing the form of implicit bias of linear networks learned using gradient descent.

Unless specified otherwise, $\|.\|$ denotes the Euclidean norm. We additionally use the notation $\mathbf{v} \propto \mathbf{v}'$ to denote equality up to strictly positive scalar multipliers, *i.e.*, when $\mathbf{v} = \gamma \mathbf{v}'$ for some $\gamma > 0$.

The following is a paraphrasing of Lemma 8 in Gunasekar et al. [2018] and is used in multiple proofs.

**Lemma 8.** *[Lemma 8 in Gunasekar et al. [2018]] For almost all linearly separable dataset $\{\mathbf{x}_n, y_n\}_n$, consider any sequence $\boldsymbol{\beta}^{(t)}$ that minimizes the empirical objective in eq. (5), i.e., $\mathcal{L}(\boldsymbol{\beta}^{(t)}) \to 0$. If (a) $\overline{\boldsymbol{\beta}}^\infty := \lim_{t \to \infty} \frac{\boldsymbol{\beta}^{(t)}}{\|\boldsymbol{\beta}^{(t)}\|}$ exists and has a positive margin, and (b) $\overline{\mathbf{z}}^\infty := \lim_{t \to \infty} \frac{-\nabla_{\boldsymbol{\beta}} \mathcal{L}(\boldsymbol{\beta}^{(t)})}{\|\nabla_{\boldsymbol{\beta}} \mathcal{L}(\boldsymbol{\beta}^{(t)})\|}$ exists, then $\exists \{\alpha_n \geq 0\}_{n \in S}$ s.t. $\overline{\mathbf{z}}^\infty = \sum_{n \in S} \alpha_n y_n \mathbf{x}_n$, where $S = \{n : y_n \langle \overline{\boldsymbol{\beta}}^\infty, \mathbf{x}_n \rangle = \min_n y_n \langle \overline{\boldsymbol{\beta}}^\infty, \mathbf{x}_n \rangle\}$ are the indices of the data points with smallest margin to the limit direction $\overline{\boldsymbol{\beta}}^\infty$.*

## A   Homogeneous Polynomial Parameterization: Proof of Theorem 4

**Theorem 4** (Homogeneous Polynomial Parameterization). *For any homogeneous polynomial map $\mathcal{P} : \mathbb{R}^P \to \mathbb{R}^D$ from parameters $\mathbf{w} \in \mathbb{R}^D$ to linear predictors, almost all datasets $\{\mathbf{x}_n, y_n\}_{n=1}^N$ separable by $\mathcal{B} := \{\mathcal{P}(\mathbf{w}) : \mathbf{w} \in \mathbb{R}^P\}$, almost all initializations $\mathbf{w}^{(0)}$, and any bounded sequence of step sizes $\{\eta_t\}_t$, consider the sequence of gradient descent updates $\mathbf{w}^{(t)}$ from eq. (7) for minimizing the empirical risk objective $\mathcal{L}_{\mathcal{P}}(\mathbf{w})$ in (4) with exponential loss $\ell(u, y) = \exp(-uy)$.*

*If (a) the iterates $\mathbf{w}^{(t)}$ asymptotically minimize the objective, i.e., $\mathcal{L}_{\mathcal{P}}(\mathbf{w}^{(t)}) = \mathcal{L}(\mathcal{P}(\mathbf{w}^{(t)})) \to 0$, (b) $\mathbf{w}^{(t)}$, and consequently $\boldsymbol{\beta}^{(t)} = \mathcal{P}(\mathbf{w}^{(t)})$, converge in direction to yield a separator with positive margin, and (c) the gradients w.r.t. to the linear predictors, $\nabla_{\boldsymbol{\beta}} \mathcal{L}(\boldsymbol{\beta}^{(t)})$ converge in direction, then the limit direction of the parameters $\overline{\mathbf{w}}^\infty = \lim_{t \to \infty} \frac{\mathbf{w}^{(t)}}{\|\mathbf{w}^{(t)}\|_2}$ is a positive scaling of a first order stationary point of the following optimization problem,*

$$\min_{\mathbf{w} \in \mathbb{R}^P} \|\mathbf{w}\|_2^2 \quad s.t. \quad \forall n, \, y_n \langle \mathbf{x}_n, \mathcal{P}(\mathbf{w}) \rangle \geq 1. \tag{14}$$

*Proof.* $\mathbf{w}^{(t)}$ are the sequence gradient descent iterates from eq. (7) for minimizing $\mathcal{L}_{\mathcal{P}}(\mathbf{w})$ in eq (4) with exponential loss over the model class of $\mathcal{B} = \{\mathcal{P}(\mathbf{w}) : \mathbf{w} \in \mathbb{R}^P\}$, where $\mathcal{P}$ is a homogeneous polynomial function. We first introduce some notation.

1. From the assumption in theorem, we have that $\overline{\mathbf{w}}^\infty = \lim_{t \to \infty} \frac{\mathbf{w}^{(t)}}{\|\mathbf{w}^{(t)}\|}$. Denoting $g(t) = \|\mathbf{w}^{(t)}\|$, we have that for some $\boldsymbol{\delta}_{\mathbf{w}}^{(t)} \to 0$, the following representation of $\mathbf{w}^{(t)}$ holds.

$$\mathbf{w}^{(t)} = \overline{\mathbf{w}}^\infty g(t) + \boldsymbol{\delta}_{\mathbf{w}}^{(t)} g(t). \tag{16}$$

2. Let $\boldsymbol{\beta}^{(t)} = \mathcal{P}(\mathbf{w}^{(t)})$ denote the sequence of linear predictors for this network induced by the gradient descent iterates. We can see that $\boldsymbol{\beta}^{(t)}$ converges in direction too using the following arguments: homogeneity of $\mathcal{P}$ implies that $\mathcal{P}(\mathbf{w}^{(t)}/\|\mathbf{w}^{(t)}\|) = \mathcal{P}(\mathbf{w}^{(t)})/\|\mathbf{w}^{(t)}\|^\nu$ for some $\nu$. Hence, $\frac{\boldsymbol{\beta}^{(t)}}{\|\boldsymbol{\beta}^{(t)}\|} = \frac{\mathcal{P}(\mathbf{w}^{(t)}/\|\mathbf{w}^{(t)}\|)}{\|\mathcal{P}(\mathbf{w}^{(t)}/\|\mathbf{w}^{(t)}\|)\|} \xrightarrow{t \to \infty} \frac{\mathcal{P}(\overline{\mathbf{w}}^\infty)}{\|\mathcal{P}(\overline{\mathbf{w}}^\infty)\|} := \overline{\boldsymbol{\beta}}^\infty$.

3. $\mathbf{z}^{(t)} = -\nabla_{\boldsymbol{\beta}} \mathcal{L}(\boldsymbol{\beta}^{(t)}) = \sum_n \exp\left(-\langle \boldsymbol{\beta}^{(t)}, y_n \mathbf{x}_n \rangle\right) y_n \mathbf{x}_n$. Since we assume that $\mathbf{z}^{(t)}$ converges in direction, let $\overline{\mathbf{z}}^\infty = \lim_{t \to \infty} \frac{\mathbf{z}^{(t)}}{\|\mathbf{z}^{(t)}\|}$. Denoting $p(t) = \|\mathbf{z}^{(t)}\|$, for some $\boldsymbol{\delta}_{\mathbf{z}}^{(t)} \to 0$, we can write $\mathbf{z}^{(t)}$ as,

$$\mathbf{z}^{(t)} = \overline{\mathbf{z}}^\infty p(t) + \boldsymbol{\delta}_{\mathbf{z}}^{(t)} p(t), \tag{17}$$

4. Let $\nabla_{\mathbf{w}}\mathcal{P}\left(\mathbf{w}^{(t)}\right) \in \mathbb{R}^{P \times D}$ denote the Jacobian of $\mathcal{P}(\mathbf{w})$, *i.e.*, $\nabla_{\mathbf{w}}\mathcal{P}\left(\mathbf{w}^{(t)}\right)[p,d] = \frac{\partial(\mathcal{P}(\mathbf{w}^{(t)})[d])}{\partial \mathbf{w}[p]}$.
If $\mathcal{P} : \mathbb{R}^P \to \mathbb{R}^D$ is a homogeneous polynomial of degree $\nu > 0$, then $\nabla_{\mathbf{w}}\mathcal{P} : \mathbb{R}^P \to \mathbb{R}^{P \times D}$ is a homogeneous polynomial of degree $\nu - 1$. Using eq. (16), we have

$$\nabla_{\mathbf{w}}\mathcal{P}(\overline{\mathbf{w}}^\infty) = \lim_{t \to \infty} \nabla_{\mathbf{w}}\mathcal{P}\left(\frac{\mathbf{w}^{(t)}}{g(t)}\right) = \lim_{t \to \infty} \frac{\nabla_{\mathbf{w}}\mathcal{P}(\mathbf{w}^{(t)})}{g(t)^{\nu-1}}$$

Thus, $\exists \boldsymbol{\delta}_1^{(t)} \to 0$, such that

$$\nabla_{\mathbf{w}}\mathcal{P}\left(\mathbf{w}^{(t)}\right) = \nabla_{\mathbf{w}}\mathcal{P}\left(\overline{\mathbf{w}}^\infty\right) g(t)^{\nu-1} + \boldsymbol{\delta}_1^{(t)} g(t)^{\nu-1}. \tag{18}$$

5. Finally, from the definition of $\nabla_{\mathbf{w}}\mathcal{P}(\mathbf{w})$, we have $\nabla_{\mathbf{w}}\mathcal{L}_{\mathcal{P}}(\mathbf{w}^{(t)}) = \nabla_{\mathbf{w}}\mathcal{P}\left(\mathbf{w}^{(t)}\right)\nabla_{\boldsymbol{\beta}}\mathcal{L}(\boldsymbol{\beta}^{(t)})$, and hence from eq. (7),

$$\Delta\mathbf{w}^{(t)} := \mathbf{w}^{(t+1)} - \mathbf{w}^{(t)} = \eta_t \nabla_{\mathbf{w}}\mathcal{P}\left(\mathbf{w}^{(t)}\right)\mathbf{z}^{(t)} \tag{19}$$

Using the assumptions in the theorem along with our argument above for convergence of $\boldsymbol{\beta}^{(t)}$ in direction, we satisfy the conditions of Lemma 8, which will be crucially used in our proof.

**KKT conditions for first order stationary points**  We want show that there exists a positive scaling of $\overline{\mathbf{w}}^\infty$, denoted as $\widetilde{\mathbf{w}}^\infty = \gamma \overline{\mathbf{w}}^\infty$ for some $\gamma > 0$, such that $\widetilde{\mathbf{w}}^\infty$ is a first order stationary point of the explicitly regularized problem in eq. (14). Towards this we show that $\widetilde{\mathbf{w}}^\infty$ satisfy the following first order KKT conditions of eq. (14)

$$\forall n, \; y_n \langle \mathbf{x}_n, \mathcal{P}(\mathbf{w}) \rangle \geq 1,$$
$$\exists \{\alpha_n\}_{n=1}^N \text{ s.t. } \forall n, \alpha_n \geq 0 \text{ and } \alpha_n = 0, \forall n \notin S := \{n \in [N] : y_n \langle \mathbf{x}_n, \mathcal{P}(\mathbf{w}) \rangle = 1\},$$
$$\mathbf{w} = \nabla_{\mathbf{w}}\mathcal{P}(\mathbf{w})\left[\sum_n \alpha_n y_n \mathbf{x}_n\right]. \tag{20}$$

**Primal feasibility.** We showed earlier that if $\mathbf{w}^{(t)}$ converges in direction, then $\boldsymbol{\beta}^{(t)} = \mathcal{P}(\mathbf{w}^{(t)})$ converges in direction to $\overline{\boldsymbol{\beta}}^\infty = \lim_{t \to \infty} \frac{\boldsymbol{\beta}^{(t)}}{\|\boldsymbol{\beta}^{(t)}\|} \propto \mathcal{P}(\overline{\mathbf{w}}^\infty)$. Further, from the assumptions in the theorem, we have that $\overline{\boldsymbol{\beta}}^\infty$ satisfies $\forall n, y_n \langle \mathbf{x}_n, \overline{\boldsymbol{\beta}}^\infty \rangle > 0$, which also implies $\min_n y_n \langle \mathbf{x}_n, \mathcal{P}(\overline{\mathbf{w}}^\infty) \rangle > 0$ since $\overline{\boldsymbol{\beta}}^\infty \propto \mathcal{P}(\overline{\mathbf{w}}^\infty)$. Now, if $\mathcal{P}$ is homogeneous of of degree $\nu$, then for $\gamma = (\min_n y_n \langle \mathbf{x}_n, \mathcal{P}(\overline{\mathbf{w}}^\infty) \rangle)^{-1/\nu}$, $\widetilde{\mathbf{w}}^\infty = \gamma \overline{\mathbf{w}}^\infty$ satisfies $\min_n y_n \langle \mathbf{x}_n, \mathcal{P}(\widetilde{\mathbf{w}}^\infty) \rangle = 1$.

**Showing other KKT conditions for $\widetilde{\mathbf{w}}^\infty$.**  The crux of the proof of Theorem 4 involves showing the existence of $\{\alpha_n \geq 0\}_n$ such that the stationarity and complementary slackness conditions in eq. (20) are satisfied. This crucially relies on a key lemma (Lemma 8) showing that the gradient in the space of linear predictors $\nabla_{\boldsymbol{\beta}}\mathcal{L}(\boldsymbol{\beta}^{(t)})$ are dominated by positive linear combinations of support vectors of the asymptotic predictor $\overline{\boldsymbol{\beta}}^\infty$.

Let $S_\infty = \{n : y_n \langle \mathcal{P}(\widetilde{\mathbf{w}}^\infty), \mathbf{x}_n \rangle = 1\}$ denote the indices of support vectors for $\mathcal{P}(\widetilde{\mathbf{w}}^\infty)$, which are also the support vectors of $\overline{\boldsymbol{\beta}}^\infty$, since by homogeneity of $\mathcal{P}$, $\overline{\boldsymbol{\beta}}^\infty \propto \mathcal{P}(\overline{\mathbf{w}}^\infty) \propto \mathcal{P}(\widetilde{\mathbf{w}}^\infty)$. Thus, from Lemma 8, we have $\overline{\mathbf{z}}^\infty = \lim_{t \to \infty} \frac{\mathbf{z}^{(t)}}{\|\mathbf{z}^{(t)}\|} = \sum_{n \in S_\infty} \alpha_n y_n \mathbf{x}_n$ for some $\{\alpha_n\}_{n \in S_\infty}$ such that $\alpha_n \geq 0$. We propose a positive scaling of this $\{\alpha_n\}_{n=1}^N$ as our candidate dual certificate, which satisfies both dual feasibility and complementary slackness.

To prove the theorem, the remaining step is to show that $\widetilde{\mathbf{w}}^\infty \propto \nabla_{\mathbf{w}}\mathcal{P}(\widetilde{\mathbf{w}}^\infty)\overline{\mathbf{z}}^\infty$. Since $\widetilde{\mathbf{w}}^\infty = \gamma \overline{\mathbf{w}}^\infty$ and $\mathcal{P}$ is homogeneous, this condition is equivalent to showing that $\overline{\mathbf{w}}^\infty \propto \nabla_{\mathbf{w}}\mathcal{P}(\overline{\mathbf{w}}^\infty)\overline{\mathbf{z}}^\infty$.

**Showing that $\overline{\mathbf{w}}^\infty \propto \nabla_{\mathbf{w}}\mathcal{P}(\overline{\mathbf{w}}^\infty)\overline{\mathbf{z}}^\infty$.**  Substituting for $\mathbf{z}^{(t)}$ and $\nabla_{\mathbf{w}}\mathcal{P}(\mathbf{w}^{(t)})$ from eqs. (17) and (18), respectively, in the gradient descent updates (eq. (19)), we have the following:

$$\mathbf{w}^{(t+1)} - \mathbf{w}^{(t)} = \eta_t \nabla_{\mathbf{w}}\mathcal{P}\left(\mathbf{w}^{(t)}\right)\mathbf{z}^{(t)}$$
$$= \eta_t \left(\nabla_{\mathbf{w}}\mathcal{P}\left(\overline{\mathbf{w}}^\infty\right) g(t)^{\nu-1} + \boldsymbol{\delta}_1^{(t)} g(t)^{\nu-1}\right)\left(\overline{\mathbf{z}}^\infty p(t) + \boldsymbol{\delta}_{\mathbf{z}}^{(t)} p(t)\right) \tag{21}$$
$$\overset{(a)}{=} \left(\eta_t p(t) g(t)^{\nu-1}\right)\left[\nabla_{\mathbf{w}}\mathcal{P}\left(\overline{\mathbf{w}}^\infty\right)\overline{\mathbf{z}}^\infty + \boldsymbol{\delta}^{(t)}\right],$$

where in $(a)$ $\boldsymbol{\delta}^{(t)} = \nabla_{\mathbf{w}}\mathcal{P}\left(\overline{\mathbf{w}}^\infty\right)\boldsymbol{\delta}_{\mathbf{z}}^{(t)} + \boldsymbol{\delta}_1^{(t)}\boldsymbol{\delta}_{\mathbf{z}}^{(t)} + \boldsymbol{\delta}_1^{(t)}\overline{\mathbf{z}}^\infty \to 0$.

Summing over $t$, we have

$$\mathbf{w}^{(t)} - \mathbf{w}^{(0)} = \nabla_{\mathbf{w}}\mathcal{P}\left(\overline{\mathbf{w}}^\infty\right)\overline{\mathbf{z}}^\infty \sum_{u<t} \eta_u p(u)g(u)^{\nu-1} + \sum_{u<t}\boldsymbol{\delta}^{(u)}\eta_u p(u)g(u)^{\nu-1}, \qquad (22)$$

We want to argue that the first term, *i.e.,* $\nabla_{\mathbf{w}}\mathcal{P}\left(\overline{\mathbf{w}}^\infty\right)\overline{\mathbf{z}}^\infty$, is the dominant term. Towards this we state and prove the following intermediate claim

**Claim 1.** $\|\nabla_{\mathbf{w}}\mathcal{P}\left(\overline{\mathbf{w}}^\infty\right)\overline{\mathbf{z}}^\infty\| > 0$ *and* $\sum_{u<t}\eta_u p(u)g(u)^{\nu-1} \to \infty$.

*Proof.* First, it is straight forward to check that for any scalar valued homogeneous polynomial $f : \mathbb{R}^P \to \mathbb{R}$ of degree $\nu$, we have $\langle\mathbf{w}, \nabla_{\mathbf{w}}f(\mathbf{w})\rangle = \nu f(\mathbf{w})$, where for $p = 1, 2\ldots, P$, $\nabla_{\mathbf{w}}f(\mathbf{w})[p] = \frac{\mathrm{d}f(\mathbf{w})}{\mathrm{d}\mathbf{w}[p]}$ (this is also known as the Euler's homogeneous function theorem). Extending this to our vector valued homogeneous function $\mathcal{P} : \mathbb{R}^P \to \mathbb{R}^D$, we have that for all $\mathbf{w}$, the Jacobian $\nabla_{\mathbf{w}}\mathcal{P}(\mathbf{w}) \in \mathbb{R}^{P\times D}$ satisfies $\nabla_{\mathbf{w}}\mathcal{P}(\mathbf{w})^\top\mathbf{w} = \nu\mathcal{P}(\mathbf{w})$.

Moreover, we have that for the limit direction $\overline{\mathbf{w}}^\infty$, the margin of the corresponding classifier is strictly positive, *i.e.,* $\min_n y_n\langle\mathcal{P}(\overline{\mathbf{w}}^\infty), \mathbf{x}_n\rangle > 0$. Now from Lemma 8, using that $\overline{\mathbf{z}}^\infty = \sum_{n\in S_\infty}\alpha_n y_n\mathbf{x}_n$ for $\alpha_n \geq 0$ (and not all zero since $\overline{\mathbf{z}}^\infty$ is unit norm), we immediately get the following

$$\overline{\mathbf{w}}^{\infty\top}\nabla_{\mathbf{w}}\mathcal{P}(\overline{\mathbf{w}}^\infty)\overline{\mathbf{z}}^\infty = \nu\mathcal{P}(\overline{\mathbf{w}}^\infty)^\top\overline{\mathbf{z}}^\infty = \nu\sum_n\alpha_n y_n\langle\mathbf{x}_n, \mathcal{P}(\mathbf{w}^\infty)\rangle > 0 \implies \nabla_{\mathbf{w}}\mathcal{P}(\overline{\mathbf{w}}^\infty)\overline{\mathbf{z}}^\infty \neq 0.$$

To prove the second part, we note the following

- since $\boldsymbol{\delta}^{(t)} \to 0$ in eq. (22), $\exists t_0$ such that $\forall t > t_0$, $\|\boldsymbol{\delta}^{(t)}\| \leq 1$, and since all the incremental updates to gradient descent are finite, we have that $\sup_t\|\boldsymbol{\delta}^{(t)}\| < \infty$,
- since $p(t) = \|\mathbf{z}^{(t)}\|$ and $g(t) = \|\mathbf{w}^{(t)}\|$ are positive, we have that $b_t = \sum_{u<t}\eta_u p(u)g(u)^{\nu-1}$ is monotonic increasing.

Thus, if $\limsup_{t\to\infty} b_t = \infty$ then $\lim_{t\to\infty} b_t = \infty$. On contrary, if $\limsup_{t\to\infty} b_t = C < \infty$, then from eq. (22), for large $t$ we get, $\|\mathbf{w}^{(t)}\| \leq \|\mathbf{w}^{(0)}\| + \|\nabla\mathcal{P}(\overline{\mathbf{w}}^\infty)\overline{\mathbf{z}}^\infty\|C + \left(\sup_t\|\boldsymbol{\delta}^{(t)}\|\right)C < \infty$ which contradicts $\|\mathbf{w}^{(t)}\| \to \infty$. $\qquad\square$

From above claim, the sequence $b_t = \sum_{u<t}\eta_u p(u)g(u)^{\nu-1}$ is monotonic increasing and diverging. Thus, for $a_t = \sum_{u<t}\boldsymbol{\delta}^{(u)}\eta_u p(u)g(u)^{\nu-1}$, using Stolz-Cesaro theorem (Theorem 11), we have

$$\lim_{t\to\infty}\frac{a_t}{b_t} = \lim_{t\to\infty}\frac{\sum_{u<t}\boldsymbol{\delta}^{(u)}\eta_u p(u)g(u)^{\nu-1}}{\sum_{u<t}\eta_u p(u)g(u)^{\nu-1}} = \lim_{t\to\infty}\frac{a_{t+1} - a_t}{b_{t+1} - b_t} = \lim_{t\to\infty}\boldsymbol{\delta}^{(t)} = 0.$$

$$\implies \text{for } \boldsymbol{\delta}_2^{(t)} \to 0, \text{ we have } \sum_{u<t}\boldsymbol{\delta}^{(u)}\eta_u p(u)g(u)^{\nu-1} = \boldsymbol{\delta}_2^{(t)}\sum_{u<t}\eta_u p(u)g(u)^{\nu-1}. \qquad (23)$$

Substituting eq. (23) in eq. (22), we have

$$\mathbf{w}^{(t)} \overset{(a)}{=} \left[\nabla_{\mathbf{w}}\mathcal{P}\left(\overline{\mathbf{w}}^\infty\right)\overline{\mathbf{z}}^\infty + \boldsymbol{\delta}_3^{(t)}\right]\left[\sum_{u<t}\eta_u p(u)g(u)^{\nu-1}\right] \qquad (24)$$

$$\implies \frac{\mathbf{w}^{(t)}}{\|\mathbf{w}^{(t)}\|} = \frac{\nabla_{\mathbf{w}}\mathcal{P}\left(\overline{\mathbf{w}}^\infty\right)\overline{\mathbf{z}}^\infty + \boldsymbol{\delta}_3^{(t)}}{\|\nabla_{\mathbf{w}}\mathcal{P}\left(\overline{\mathbf{w}}^\infty\right)\overline{\mathbf{z}}^\infty + \boldsymbol{\delta}_3^{(t)}\|} \overset{(b)}{\to} \frac{\nabla_{\mathbf{w}}\mathcal{P}\left(\overline{\mathbf{w}}^\infty\right)\overline{\mathbf{z}}^\infty}{\|\nabla_{\mathbf{w}}\mathcal{P}\left(\overline{\mathbf{w}}^\infty\right)\overline{\mathbf{z}}^\infty\|} \qquad (25)$$

$$\implies \overline{\mathbf{w}}^\infty = \lim_{t\to\infty}\frac{\mathbf{w}^{(t)}}{\|\mathbf{w}^{(t)}\|} = \frac{\nabla_{\mathbf{w}}\mathcal{P}\left(\overline{\mathbf{w}}^\infty\right)\overline{\mathbf{z}}^\infty}{\|\nabla_{\mathbf{w}}\mathcal{P}\left(\overline{\mathbf{w}}^\infty\right)\overline{\mathbf{z}}^\infty\|} \propto \nabla_{\mathbf{w}}\mathcal{P}\left(\overline{\mathbf{w}}^\infty\right)\overline{\mathbf{z}}^\infty, \qquad (26)$$

where in $(a)$ we absorbed the diminishing terms into $\boldsymbol{\delta}_3^{(t)} = \boldsymbol{\delta}_2^{(t)} + \mathbf{w}^{(0)}/\sum_{u<t}\eta_u p(u)g(u)^{\nu-1} \to 0$, $(b)$ follows since we proved in the claim above that $\nabla_{\mathbf{w}}\mathcal{P}\left(\overline{\mathbf{w}}^\infty\right)\overline{\mathbf{z}}^\infty \neq 0$ and hence dominates $\boldsymbol{\delta}_3^{(t)}$.

We have shown that $\overline{\mathbf{w}}^\infty = \overline{\gamma}\nabla_{\mathbf{w}}\mathcal{P}\left(\overline{\mathbf{w}}^\infty\right)\overline{\mathbf{z}}^\infty$ for a positive scalar $\overline{\gamma}$, which completes the proof. $\quad\square$

# B Linear Fully Connected Networks: Proof of Theorem 1

**Theorem 1** (Linear fully connected networks). *For any depth $L$, almost all linearly separable datasets $\{\mathbf{x}_n, y_n\}_{n=1}^N$, almost all initializations $\mathbf{w}^{(0)}$, and any bounded sequence of step sizes $\{\eta_t\}_t$, consider the sequence gradient descent iterates $\mathbf{w}^{(t)}$ in eq. (7) for minimizing $\mathcal{L}_{\mathcal{P}_{full}}(\mathbf{w})$ in eq. (4) with exponential loss $\ell(\widehat{y}, y) = \exp(-\widehat{y}y)$ over $L$–layer fully connected linear networks.*

*If (a) the iterates $\mathbf{w}^{(t)}$ minimize the objective, i.e., $\mathcal{L}_{\mathcal{P}_{full}}(\mathbf{w}^{(t)}) \to 0$, (b) $\mathbf{w}^{(t)}$, and consequently $\boldsymbol{\beta}^{(t)} = \mathcal{P}_{full}(\mathbf{w}^{(t)})$, converge in direction to yield a separator with positive margin, and (c) gradients with respect to linear predictors $\nabla_{\boldsymbol{\beta}}\mathcal{L}(\boldsymbol{\beta}^{(t)})$ converge in direction, then the limit direction is given by,*

$$\overline{\boldsymbol{\beta}}^\infty = \lim_{t\to\infty} \frac{\mathcal{P}_{full}(\mathbf{w}^{(t)})}{\|\mathcal{P}_{full}(\mathbf{w}^{(t)})\|} = \frac{\boldsymbol{\beta}^*_{\ell_2}}{\|\boldsymbol{\beta}^*_{\ell_2}\|}, \text{ where } \boldsymbol{\beta}^*_{\ell_2} := \underset{w}{\operatorname{argmin}}\|\boldsymbol{\beta}\|_2^2 \ \text{s.t.} \ \forall n, y_n\langle \mathbf{x}_n, \boldsymbol{\beta}\rangle \geq 1. \quad (8)$$

*Proof.* Recall that for fully connected networks of any depth $L > 0$ with parameters $\mathbf{w} = [\mathbf{w}_l \in \mathbb{R}^{D_{l-1} \times D_l}]_{l-1}^L$, the equivalent linear predictor given by $\mathcal{P}_{full}(\mathbf{w}) = \mathbf{w}_1\mathbf{w}_2\ldots\mathbf{w}_L$ is a homogeneous polynomial of degree $L$.

Let $\mathbf{w}^{(t)} = [\mathbf{w}_l^{(t)} \in \mathbb{R}^{D_{l-1} \times D_l}]_{l=1}^L$ denote the iterates of individual matrices $\mathbf{w}_l$ along the gradient descent path, and $\boldsymbol{\beta}^{(t)} = \mathcal{P}_{full}(\mathbf{w}^{(t)})$ denote the corresponding sequence of linear predictors.

We first introduce the following notation.

1. Let $\overline{\mathbf{w}}^\infty = \lim_{t\to\infty} \frac{\mathbf{w}^{(t)}}{\|\mathbf{w}^{(t)}\|}$ denote the limit direction of the parameters, with component matrices in each layer denoted as $\overline{\mathbf{w}}^\infty = [\overline{\mathbf{w}}_l^\infty]$. Specializing (16) for fully connected networks, we have:

$$\mathbf{w}_l^{(t)} = \overline{\mathbf{w}}_l^\infty g(t) + \boldsymbol{\delta}_{\mathbf{w}_l}^{(t)} g(t), \quad (27)$$

where $g(t) = \|\mathbf{w}^{(t)}\|$ and $\boldsymbol{\delta}_{\mathbf{w}_l}^{(t)} \to 0$.

2. For $0 < l_1 < l_2 \leq L$, denote $\mathbf{w}_{l_1:l_2}^{(t)} = \mathbf{w}_{l_1}^{(t)}\mathbf{w}_{l_1+1}^{(t)}\ldots\mathbf{w}_{l_2}^{(t)}$ and $\overline{\mathbf{w}}_{l_1:l_2}^\infty = \overline{\mathbf{w}}_{l_1}^\infty\overline{\mathbf{w}}_{l_1+1}^\infty\ldots\overline{\mathbf{w}}_{l_2}^\infty$. Using eq. (27), we can check by induction on $l_2 - l_1$ that $\lim_{t\to\infty} \frac{\mathbf{w}_{l_1:l_2}^{(t)}}{g(t)^{l_2-l_1+1}} = \overline{\mathbf{w}}_{l_1:l_2}^\infty$, and hence $\exists\boldsymbol{\delta}_{\mathbf{w}_{l_1:l_2}}^{(t)} \to 0$ such that the following holds,

$$\mathbf{w}_{l_1:l_2}^{(t)} = \overline{\mathbf{w}}_{l_1:l_2}^\infty g(t)^{l_2-l_1+1} + \boldsymbol{\delta}_{\mathbf{w}_{l_1:l_2}}^{(t)} g(t)^{l_2-l_1+1}. \quad (28)$$

3. Let $\mathbf{z}^{(t)} = -\nabla_{\boldsymbol{\beta}}\mathcal{L}(\boldsymbol{\beta}^{(t)})$. Again repeating eq. (17) for fully connected networks, we have for some $\boldsymbol{\delta}_{\mathbf{z}}^{(t)} \to 0$ and $p(t) = \|\mathbf{z}^{(t)}\|$,

$$\mathbf{z}^{(t)} = \overline{\mathbf{z}}^\infty p(t) + \boldsymbol{\delta}_{\mathbf{z}}^{(t)} p(t). \quad (29)$$

4. From Lemma 8, we have that $\exists\{\alpha_n\}_{n\in S_\infty}$ such that $\overline{\mathbf{z}}^\infty = \sum_{n\in S_\infty} \alpha_n y_n\mathbf{x}_n$, where $S_\infty$ are support vectors of $\overline{\boldsymbol{\beta}}^\infty = \lim_{t\to\infty} \frac{\boldsymbol{\beta}^{(t)}}{\|\boldsymbol{\beta}^{(t)}\|} \propto \mathcal{P}_{full}(\overline{\mathbf{w}}^\infty)$.

The proof of Theorem 1 is fairly straight forward from using Lemma 8 and the intermediate results in the proof of Theorem 4.

**Showing KKT conditions for $\widetilde{\boldsymbol{\beta}}^\infty \propto \mathcal{P}_{full}(\overline{\mathbf{w}}^\infty)$.** Using our notation described above, we have $\overline{\mathbf{w}}_{1:L}^\infty = \mathcal{P}_{full}(\overline{\mathbf{w}}^\infty)$. In the following arguments we show that a positive scaling $\widetilde{\boldsymbol{\beta}}^\infty = \gamma\overline{\mathbf{w}}_{1:L}^\infty$ satisfies the following KKT conditions for the optimality of $\ell_2$ maximum margin problem in eq. (8):

$$\exists\{\alpha_n\}_{n=1}^N \quad \text{s.t.} \quad \forall n, y_n\langle\mathbf{x}_n, \boldsymbol{\beta}\rangle \geq 1, \boldsymbol{\beta} = \sum_n \alpha_n y_n\mathbf{x}_n,$$
$$\forall n, \alpha_n \geq 0 \text{ and } \alpha_n = 0, \forall i \notin S := \{i \in [N] : y_n\langle\mathbf{x}_n, \boldsymbol{\beta}\rangle = 1\}. \quad (30)$$

As we saw in proof of Theorem 4, since $\overline{\mathbf{w}}_{1:L}^\infty = \mathcal{P}_{full}(\overline{\mathbf{w}}^\infty)$ has strictly positive margin, using homogeneity of $\mathcal{P}_{full}$, we can scale $\overline{\mathbf{w}}_{1:L}^\infty$ to get $\widetilde{\boldsymbol{\beta}}^\infty = \gamma\overline{\mathbf{w}}_{1:L}^\infty$ with unit margin, *i.e.,*

$\forall n, y_n \langle \mathbf{x}_n, \widetilde{\boldsymbol{\beta}}^\infty \rangle \geq 1$. For dual variables, we again use a positive scaling of $\alpha_n$ from Lemma 8, such that $\overline{\mathbf{z}}^\infty = \sum_{n \in S_\infty} \alpha_n y_n \mathbf{x}_n$. In order to prove the theorem, we need to show that $\widetilde{\boldsymbol{\beta}}^\infty \propto \overline{\mathbf{z}}^\infty$ or equivalently $\overline{\mathbf{w}}_{1:L}^\infty \propto \overline{\mathbf{z}}^\infty$.

Recall that in the proof of Theorem 4, we showed a version of stationarity in the parameter space in eq. (26), repeated below.

$$\overline{\mathbf{w}}^\infty \propto \nabla \mathbf{w} \mathcal{P}(\overline{\mathbf{w}}^\infty) \overline{\mathbf{z}}^\infty. \tag{31}$$

This case in particular includes $\mathcal{P}_{full}$ which is homogeneous with $\nu = L$. We special case the result fully connected network. In particular, for the parameters of the first layer $\mathbf{w}_1$, we have $\mathcal{P}(\mathbf{w}) = \mathbf{w}_1 \mathbf{w}_{2:L}$, where $\mathbf{w}_1 \in \mathbb{R}^{d \times d_1}$ and $\mathbf{w}_{2:L} \in \mathbb{R}^{d_1 \times 1}$. This implies, for any $\mathbf{z}$, $\nabla_{\mathbf{w}_1} \mathcal{P}(\mathbf{w}) \mathbf{z} = \mathbf{z} \mathbf{w}_{2:L}^\top$. Using this along with eq. (31), we get the following expression for some positive scalar $\overline{\gamma}$

$$\overline{\mathbf{w}}_1^\infty = \overline{\gamma} \, \nabla_{\mathbf{w}_1} \mathcal{P}(\overline{\mathbf{w}}^\infty) \overline{\mathbf{z}}^\infty = \overline{\gamma} \, \overline{\mathbf{z}}^\infty \overline{\mathbf{w}}_{2:L}^{\infty \top} \implies \overline{\mathbf{w}}_{1:L}^\infty = \overline{\mathbf{w}}_1^\infty \overline{\mathbf{w}}_{2:L}^\infty = \overline{\gamma} \, \overline{\mathbf{z}}^\infty \cdot \| \overline{\mathbf{w}}_{2:L}^\infty \|^2 \propto \overline{\mathbf{z}}^\infty. \tag{32}$$

Since $\overline{\mathbf{w}}_{1:L}^\infty \propto \widetilde{\boldsymbol{\beta}}^\infty$, we have shown that $\widetilde{\boldsymbol{\beta}}^\infty \propto \overline{\mathbf{z}}^\infty$, which completes our proof of Theorem 1.  □

# C   Linear Convolutional Networks: Proof of Theorem 2–2a

Recall that $L$–layer linear convolutional networks have parameters $\mathbf{w} = [\mathbf{w}_l \in \mathbb{R}^D]_{l-1}^L$. We first recall some complex numbers terminology and properties

1. Complex vectors $\widehat{\mathbf{z}} \in \mathbf{C}^D$ are represented in polar form as $\widehat{\mathbf{z}} = |\widehat{\mathbf{z}}| e^{i \phi_{\widehat{\mathbf{z}}}}$, where $|\widehat{\mathbf{z}}| \in \mathbb{R}_+^D$ and $\phi_{\widehat{\mathbf{z}}} \in [0, 2\pi)^D$ are the vectors with magnitudes and phases, respectively, of components $\widehat{\mathbf{z}}$.
2. For $\widehat{\mathbf{z}} = |\widehat{\mathbf{z}}| e^{i \phi_{\widehat{\mathbf{z}}}} \in \mathbf{C}^D$, the complex conjugate vector is denoted by $\widehat{\mathbf{z}}^* = |\widehat{\mathbf{z}}| e^{-i \phi_{\widehat{\mathbf{z}}}}$.
3. The complex inner product for $\widehat{\mathbf{x}}, \widehat{\boldsymbol{\beta}} \in \mathbf{C}^D$ is given by $\langle \widehat{\mathbf{x}}, \widehat{\boldsymbol{\beta}} \rangle = \sum_d \widehat{\mathbf{x}}[d] \widehat{\boldsymbol{\beta}}^*[d] = \widehat{\mathbf{x}}^\top \widehat{\boldsymbol{\beta}}^*$.
4. Let $\mathcal{F} \in \mathbb{C}^{D \times D}$ denote the discrete Fourier transform matrix with $\mathcal{F}[d, p] = \frac{1}{\sqrt{D}} \omega_D^{dp}$ where recall that $\omega_D = e^{-\frac{2\pi i}{D}}$ is the $D^{\text{th}}$ complex root of unity. Thus, for any $\mathbf{z} \in \mathbb{R}^D$, the representation in Fourier basis is given by $\widehat{\mathbf{z}} = \mathcal{F} \mathbf{z}$. $\mathcal{F}$ and its complex conjugate matrix $\mathcal{F}^*$ also satisfy: $\mathcal{F} \mathcal{F}^* = \mathcal{F}^* \mathcal{F} = I, \mathcal{F} = \mathcal{F}^\top$ and $\mathcal{F}^* = \mathcal{F}^{* \top}$.

Before getting into full proofs of Theorem 2a–2, we also prove the two lemmas (Lemma 3 and Lemma 9) that establish equivalence of dynamics of gradient descent on full dimensional convolutional networks to those on linear diagonal networks (Figure 1c), albeit with complex valued parameters. This makes the analysis of the of convolutional networks simpler and more intuitive.

We begin by proving Lemma 3 which shows the equivalence of representation between convolutional networks and diagonal networks.

**Lemma 3.** *For full-dimensional convolutions, $\boldsymbol{\beta} = \mathcal{P}_{conv}(\mathbf{w})$ is equivalent to*

$$\widehat{\boldsymbol{\beta}} = diag(\widehat{\mathbf{w}}_1) \dots diag(\widehat{\mathbf{w}}_{L-1}) \widehat{\mathbf{w}}_L,$$

*where for $l = 1, 2, \dots, L$, $\widehat{\mathbf{w}}_1 \in \mathbf{C}^D$ are the Fourier coefficients of the parameters $\mathbf{w}_l \in \mathbb{R}^D$.*

*Proof.* First, we state the following properties which follow immediately from definitions:

1. For $\mathbf{x}, \boldsymbol{\beta} \in \mathbb{R}^D$,

$$\langle \mathbf{x}, \boldsymbol{\beta} \rangle = \mathbf{x}^\top \boldsymbol{\beta} = \mathbf{x}^\top \mathcal{F} \mathcal{F}^* \boldsymbol{\beta} = \widehat{\mathbf{x}}^\top \widehat{\boldsymbol{\beta}}^* = \langle \widehat{\mathbf{x}}, \widehat{\boldsymbol{\beta}} \rangle, \tag{33}$$

where recall that the complex inner product is given by $\langle \widehat{\mathbf{x}}, \widehat{\boldsymbol{\beta}} \rangle = \widehat{\mathbf{x}}^\top \widehat{\boldsymbol{\beta}}^*$.

2. We next show the following property

$$\mathcal{F}(\mathbf{h} \star \mathbf{w}) = (\mathcal{F}\mathbf{h}) \odot (\mathcal{F}^* \mathbf{w}) = \widehat{\mathbf{h}} \odot \widehat{\mathbf{w}}^*, \tag{34}$$

where $\odot$ denotes the Hadamard product (elementwise product), *i.e.*, $(a \odot b)[d] = a[d]b[d]$.

The above equation follows from simple manipulations of definitions: recall that $(\mathcal{F}\mathbf{z})[d] = \frac{1}{\sqrt{D}} \sum_{p=0}^{D-1} \mathbf{z}[p] \omega_D^{pd}$ and $\mathbf{h} \star \mathbf{w}$ defined in eq. (2) as $(\mathbf{h} \star \mathbf{w})[d] = \frac{1}{\sqrt{D}} \sum_{k=0}^{D-1} \mathbf{w}[k] \mathbf{h}[(d+k) \bmod D]$.

$$\widehat{\mathbf{h}} \odot \widehat{\mathbf{w}}^*[d] = \widehat{\mathbf{h}}[d]\widehat{\mathbf{w}}^*[d] = \frac{1}{D}\sum_{k=0}^{D-1}\sum_{k'=0}^{D-1}\mathbf{w}[k]\mathbf{h}[k']\omega_D^{(k'-k)d} \stackrel{(a)}{=} \frac{1}{D}\sum_{k=0}^{D-1}\sum_{k'=0}^{D-1}\mathbf{w}[k]\mathbf{h}[k']\omega_D^{((k'-k) \bmod \mathrm{D})d}$$

$$\stackrel{(b)}{=} \frac{1}{\sqrt{D}}\sum_{p=0}^{D-1}\left[\frac{1}{\sqrt{D}}\sum_{k=0}^{D-1}\mathbf{w}[k]\mathbf{h}[(p+k) \bmod D]\right]\omega_D^{pd} = (\mathcal{F}(\mathbf{h}\star\mathbf{w}))[d], \tag{35}$$

where $(a)$ follows as $\omega_D^D = 1$ and in $(b)$ we used the change of variables $p = (k'-k) \bmod D$ (recall our use of modulo operator as $a \bmod D = a - D\lfloor\frac{a}{D}\rfloor$).

Recall from eq. (3) the output of an $L$-layer convolutional network is given by

$$\widehat{y}(\mathbf{x}) = ((((\mathbf{x}\star\mathbf{w}_1)\star\mathbf{w}_2)\ldots)\star\mathbf{w}_{L-1})^\top\mathbf{w}_L = \langle\mathbf{x},\boldsymbol{\beta}\rangle.$$

Denote $\mathbf{h}_{L-1}(\mathbf{x}) = (((\mathbf{x}\star\mathbf{w}_1)\star\mathbf{w}_2)\ldots)\star\mathbf{w}_{L-1}$. By iteratively using eq. (34), we have

$$\mathcal{F}\mathbf{h}_{L-1}(\mathbf{x}) = \mathcal{F}\mathbf{x}\odot\mathcal{F}^*\mathbf{w}_1\odot\mathcal{F}^*\mathbf{w}_2\ldots\odot\mathcal{F}^*\mathbf{w}_{L-1}. \tag{36}$$

Thus, on one hand using the above equation we have,

$$\widehat{y}(\mathbf{x}) = \mathbf{h}_{L-1}(x)^\top\mathbf{w}_L = \mathbf{h}_{L-1}(\mathbf{x}))^\top\mathcal{F}\mathcal{F}^*\mathbf{w}_L = (\mathcal{F}\mathbf{h}_{L-1}(\mathbf{x}))^\top(\mathcal{F}^*\mathbf{w}_L)$$
$$\stackrel{(a)}{=} (\mathcal{F}(\mathbf{x}))^\top(\mathcal{F}^*\mathbf{w}_1\odot\mathcal{F}^*\mathbf{w}_2\ldots\odot\mathcal{F}^*\mathbf{w}_L) \stackrel{(b)}{=} \langle\widehat{\mathbf{x}},\mathcal{F}\mathbf{w}_1\odot\mathcal{F}\mathbf{w}_2\ldots\odot\mathcal{F}\mathbf{w}_L\rangle, \tag{37}$$

where $(a)$ follows from substituting for $\mathcal{F}\mathbf{h}_{L-1}(\mathbf{x})$ from eq. (36) and noting that for any $\{\mathbf{z}_l \in \mathbb{R}^D\}$, $(\mathbf{z}_1\odot\mathbf{z}_2\odot\ldots\mathbf{z}_{L-1})^\top\mathbf{z}_L = \mathbf{z}_1^\top(\mathbf{z}_2\odot\mathbf{z}_3\odot\ldots\mathbf{z}_L)$, and $(b)$ uses the definition of complex inner product $\langle\widehat{\mathbf{x}},\widehat{\boldsymbol{\beta}}\rangle = \widehat{\mathbf{x}}^\top\widehat{\boldsymbol{\beta}}^*$.

Now further using eq. (33) in above equation, we have

$$\langle\mathbf{x},\mathcal{P}_{conv}(\mathbf{w})\rangle = \langle\widehat{\mathbf{x}},\mathcal{F}\mathcal{P}_{conv}(\mathbf{w})\rangle = \widehat{y}(\mathbf{x})$$
$$\implies \langle\widehat{\mathbf{x}},\mathcal{F}\mathcal{P}_{conv}(\mathbf{w})\rangle = \langle\widehat{\mathbf{x}},\mathcal{F}\mathbf{w}_1\odot\mathcal{F}\mathbf{w}_2\ldots\odot\mathcal{F}\mathbf{w}_L\rangle. \tag{38}$$

Thus, for $\boldsymbol{\beta} = \mathcal{P}_{conv}(\mathbf{w})$, we have shown that $\widehat{\boldsymbol{\beta}} = \mathcal{F}\mathcal{P}_{conv}(\mathbf{w}) = \widehat{\mathbf{w}}_1\odot\widehat{\mathbf{w}}_2\ldots\odot\widehat{\mathbf{w}}_L = \mathrm{diag}(\widehat{\mathbf{w}}_1)\mathrm{diag}(\widehat{\mathbf{w}}_2)\ldots\mathrm{diag}(\widehat{\mathbf{w}}_{L-1})\widehat{\mathbf{w}}_L$. $\qquad\square$

For $\widehat{\mathbf{w}} = [\widehat{\mathbf{w}}_l \in \mathbf{C}^D]_{l=1}^L$, let $\mathcal{P}_{diag}(\widehat{\mathbf{w}}) = \mathrm{diag}(\widehat{\mathbf{w}}_1)\mathrm{diag}(\widehat{\mathbf{w}}_2)\ldots\mathrm{diag}(\widehat{\mathbf{w}}_{L-1})\widehat{\mathbf{w}}_L = \widehat{\mathbf{w}}_1\odot\widehat{\mathbf{w}}_2\ldots\odot\widehat{\mathbf{w}}_L$ denote the equivalent parameterization of convolutional network in Fourier domain.

The above lemma shows that optimizing $\mathcal{L}_{\mathcal{P}_{conv}}(\mathbf{w})$ in eq. (4) is equivalent to the following minimization problem in terms of representation,

$$\min_{\widehat{\mathbf{w}}}\widehat{\mathcal{L}}_{\mathcal{P}_{diag}}(\widehat{\mathbf{w}}) := \sum_{n=1}^N\ell(\langle\widehat{\mathbf{x}}_n,\mathcal{P}_{diag}(\widehat{\mathbf{w}})\rangle, y_n) \tag{39}$$

The following lemma further shown that not only the representations of $\mathcal{P}_{conv}(\mathbf{w})$ and $\mathcal{P}_{diag}(\widehat{\mathbf{w}})$ are equivalent, but there corresponding gradient descent updates for problems in eq. (4) and eq. (39) are also equivalent up to Fourier transformations.

**Lemma 9.** *Consider the gradient descent iterates $\mathbf{w}^{(t)} = [\mathbf{w}_l^{(t)}]_{l=1}^L$ from eq. (7) for minimizing $\mathcal{L}_{\mathcal{P}_{conv}}$ in eq. (4) over full dimensional linear convolutional networks. For all $l$, the incremental update directions, $\Delta\mathbf{w}_l^{(t)} := \mathbf{w}_l^{(t+1)} - \mathbf{w}_l^{(t)} = -\eta_t\nabla_{\mathbf{w}_l}\mathcal{L}_{\mathcal{P}_{conv}}(\mathbf{w}^{(t)})$ satisfy the following,*

$$\mathcal{F}\Delta\mathbf{w}_l^{(t)} = \widehat{\mathbf{w}}_l^{(t+1)} - \widehat{\mathbf{w}}_l^{(t)} = -\eta_t\nabla_{\widehat{\mathbf{w}}_l}\widehat{\mathcal{L}}_{\mathcal{P}_{diag}}(\widehat{\mathbf{w}}^{(t)}), \tag{40}$$

*where $\widehat{\mathbf{w}}^{(t)} = \left[\widehat{\mathbf{w}}_l^{(t)}\right]_{l=1}^L$ are the Fourier transformations of $\mathbf{w}^{(t)} = [\mathbf{w}_l^{(t)}]_{l=1}^L$, respectively.*

The above lemma shows that Fourier transformation of the gradient descent iterates $\mathbf{w}^{(t)} = [\mathbf{w}_l^{(t)}]_{l=1}^L$ for $\mathcal{L}_{\mathcal{P}_{conv}}$ in eq. (4) are equivalently obtained by gradient descent on the complex parameters $\widehat{\mathbf{w}}$ for minimizing $\widehat{\mathcal{L}}_{\mathcal{P}_{diag}}$ in eq. (39)

*Proof.* We use the notation $\underset{l'\neq l}{\odot} \widehat{\mathbf{w}}_{l'} = \widehat{\mathbf{w}}_1 \odot \widehat{\mathbf{w}}_2 \ldots \widehat{\mathbf{w}}_{l-1} \odot \widehat{\mathbf{w}}_{l+1} \ldots \odot \widehat{\mathbf{w}}_L$ to denote Hadamard product across all parameters $\widehat{\mathbf{w}}_{l'}$ with $l' \neq l$.

For any $\mathbf{w} = [\mathbf{w}_l]_{l=1}^L$, using eq. (38), we have the following for all $l$,

$$\langle \mathbf{x}, \mathcal{P}_{conv}(\mathbf{w})\rangle = \langle \widehat{\mathbf{x}}, \mathcal{P}_{diag}(\widehat{\mathbf{w}})\rangle = \widehat{\mathbf{x}}^\top (\widehat{\mathbf{w}}_1^* \odot \widehat{\mathbf{w}}_2^* \odot \ldots \widehat{\mathbf{w}}_L^*) = \widehat{\mathbf{w}}_l^{*\top} \left[ \left( \underset{l'\neq l}{\odot} \widehat{\mathbf{w}}_{l'}^* \right) \odot \widehat{\mathbf{x}} \right]. \tag{41}$$

Using the above equation we have,

$$\ell(\langle \mathbf{x}, \mathcal{P}_{conv}(\mathbf{w})\rangle, y_n) = \ell\left( \mathbf{w}_l^\top \, \mathcal{F}^* \left[ \left( \underset{l'\neq l}{\odot} \widehat{\mathbf{w}}_{l'}^* \right) \odot \widehat{\mathbf{x}} \right], y_n \right)$$

$$\implies \nabla_{\mathbf{w}_l} \ell(\langle \mathbf{x}, \mathcal{P}_{conv}(\mathbf{w})\rangle, y_n) \overset{(a)}{=} \ell'(\langle \mathbf{x}, \mathcal{P}_{conv}(\mathbf{w})\rangle, y_n) \mathcal{F}^* \left[ \left( \underset{l'\neq l}{\odot} \widehat{\mathbf{w}}_l^* \right) \odot \widehat{\mathbf{x}} \right] \tag{42}$$

$$= \mathcal{F}^* \left[ \ell'(\langle \widehat{\mathbf{x}}, \mathcal{P}_{diag}(\widehat{\mathbf{w}})\rangle, y_n) \left( \underset{l'\neq l}{\odot} \widehat{\mathbf{w}}_l^* \right) \odot \widehat{\mathbf{x}} \right] = \mathcal{F}^* \nabla_{\widehat{\mathbf{w}}_l} \ell(\langle \widehat{\mathbf{x}}, \mathcal{P}_{diag}(\widehat{\mathbf{w}})\rangle, y_n).$$

where in $(a)$ we use $\ell'(\widehat{y}, y) = \frac{\partial \ell(\widehat{y}, y)}{\partial \widehat{y}}$ and the remaining equalities simply follow from manipulation of derivatives. From above equation, we have the following:

$$\mathcal{F}\Delta\mathbf{w}_l^{(t)} = -\eta_t \mathcal{F} \nabla_{\mathbf{w}_l} \mathcal{L}_{\mathcal{P}_{conv}}(\mathbf{w}^{(t)}) = -\eta_t \mathcal{F} \sum_{n=1}^N \nabla_{\mathbf{w}_l} \ell(\langle \mathbf{x}_n, \mathcal{P}_{conv}(\mathbf{w}^{(t)})\rangle, y_n)$$

$$= -\eta_t \mathcal{F}\mathcal{F}^* \sum_{n=1}^N \nabla_{\widehat{\mathbf{w}}_l} \ell(\langle \widehat{\mathbf{x}}_n, \mathcal{P}_{diag}(\widehat{\mathbf{w}}^{(t)})\rangle, y_n) = -\eta_t \nabla_{\widehat{\mathbf{w}}_l} \widehat{\mathcal{L}}_{\mathcal{P}_{diag}}(\widehat{\mathbf{w}}^{(t)}). \qquad \square$$

## C.1  Proof of Theorem 2–2a

**Theorem 2** (Linear convolutional networks of depth two). *For almost all linearly separable datasets $\{\mathbf{x}_n, y_n\}_{n=1}^N$, almost all initializations $\mathbf{w}^{(0)}$, and any sequence of step sizes $\{\eta_t\}_t$ with $\eta_t$ smaller than the local Lipschitz at $\mathbf{w}^{(t)}$, consider the sequence gradient descent iterates $\mathbf{w}^{(t)}$ in eq. (7) for minimizing $\mathcal{L}_{\mathcal{P}_{conv}}(\mathbf{w})$ in eq. (4) with exponential loss over 2–layer linear convolutional networks.*

*If (a) the iterates $\mathbf{w}^{(t)}$ minimize the objective, i.e., $\mathcal{L}_{\mathcal{P}_{conv}}(\mathbf{w}^{(t)}) \to 0$, (b) $\mathbf{w}^{(t)}$ converge in direction to yield a separator $\overline{\boldsymbol{\beta}}^\infty$ with positive margin, (c) the phase of the Fourier coefficients $\widehat{\boldsymbol{\beta}}^{(t)}$ of the linear predictors $\boldsymbol{\beta}^{(t)}$ converge coordinate-wise, i.e., $\forall d$, $e^{i\phi_{\widehat{\boldsymbol{\beta}}^{(t)}[d]}} \to e^{i\phi_{\widehat{\boldsymbol{\beta}}^\infty}[d]}$, and (d) the gradients $\nabla_{\boldsymbol{\beta}}\mathcal{L}(\boldsymbol{\beta}^{(t)})$ converge in direction, then the limit direction $\overline{\boldsymbol{\beta}}^\infty$ is given by,*

$$\overline{\boldsymbol{\beta}}^\infty = \frac{\boldsymbol{\beta}_{\mathcal{F},1}^*}{\|\boldsymbol{\beta}_{\mathcal{F},1}^*\|}, \text{ where } \boldsymbol{\beta}_{\mathcal{F},1}^* := \underset{\boldsymbol{\beta}}{\arg\min}\|\widehat{\boldsymbol{\beta}}\|_1 \text{ s.t. } \forall n, y_n\langle \boldsymbol{\beta}, \mathbf{x}_n\rangle \geq 1. \tag{9}$$

**Theorem 2a** (Linear Convolutional Networks of any Depth). *For any depth $L$, under the conditions of Theorem 2, the limit direction $\overline{\boldsymbol{\beta}}^\infty = \lim_{t\to\infty} \frac{\mathcal{P}_{conv}(\mathbf{w}^{(t)})}{\|\mathcal{P}_{conv}(\mathbf{w}^{(t)})\|}$ is a scaling of a first order stationary point of the following optimization problem,*

$$\min_{\boldsymbol{\beta}} \|\widehat{\boldsymbol{\beta}}\|_{2/L} \text{ s.t. } \forall n, y_n\langle \boldsymbol{\beta}, \mathbf{x}_n\rangle \geq 1, \tag{10}$$

*where the $\ell_p$ penalty given by $\|z\|_p = \left( \sum_{i=1}^D |z[i]|^p \right)^{1/p}$ (also called the bridge penalty) is a norm for $p = 1$ and a quasi-norm for $p < 1$.*

For the gradient descent iterates $\mathbf{w}^{(t)} = [\mathbf{w}_l^{(t)}]_{l=1}^L$ from eq. (7) denote the sequence of corresponding linear predictors as $\boldsymbol{\beta}^{(t)} = \mathcal{P}_{conv}(\mathbf{w}^{(t)})$. Let $\widehat{\boldsymbol{\beta}}^{(t)} = \mathcal{F}\boldsymbol{\beta}^{(t)}$ and $\widehat{\mathbf{w}}_l^{(t)} = \mathcal{F}\mathbf{w}_l^{(t)}$ denote the Fourier transforms of $\boldsymbol{\beta}^{(t)}$ and $\mathbf{w}_l^{(t)}$, respectively, and let $\widehat{\mathbf{w}}^{(t)} = \left[ \widehat{\mathbf{w}}_l^{(t)} \right]_{l=1}^L$.

Summarizing the results so far, we have $\widehat{\boldsymbol{\beta}}^{(t)} = \widehat{\mathbf{w}}_1^{(t)} \odot \widehat{\mathbf{w}}_2^{(t)} \ldots \odot \widehat{\mathbf{w}}_L^{(t)}$ (from Lemma 3) and $\Delta\widehat{\mathbf{w}}_l^{(t)} := \widehat{\mathbf{w}}_l^{(t+1)} - \widehat{\mathbf{w}}_l^{(t)} = -\eta_t \nabla_{\widehat{\mathbf{w}}_l} \widehat{\mathcal{L}}_{\mathcal{P}_{diag}}(\widehat{\mathbf{w}}^{(t)})$ (from Lemma 9).

We use the following observations/notations

1. Let $\overline{\mathbf{w}}^\infty = \lim\limits_{t\to\infty} \frac{\mathbf{w}^{(t)}}{\|\mathbf{w}^{(t)}\|}$. Denote the Fourier transform of $\overline{\mathbf{w}}^\infty = [\overline{\mathbf{w}}_l^\infty]$ as $\widehat{\overline{\mathbf{w}}}^\infty = [\widehat{\overline{\mathbf{w}}}_l^\infty]$.
   Taking Fourier transforms of eq. (16) which are also applicable here, we have:

$$\widehat{\mathbf{w}}_l^{(t)} = \widehat{\overline{\mathbf{w}}}_l^\infty g(t) + \widehat{\boldsymbol{\delta}}_{\mathbf{w}_l}^{(t)} g(t), \tag{43}$$

   where $g(t) = \|\mathbf{w}^{(t)}\| = \|\widehat{\mathbf{w}}^{(t)}\|$ and $\widehat{\boldsymbol{\delta}}_{\mathbf{w}_l}^{(t)} \to 0$.

2. Denote the negative gradients with respect to $\boldsymbol{\beta}^{(t)}$ as $\mathbf{z}^{(t)} = -\nabla_{\boldsymbol{\beta}} \mathcal{L}(\boldsymbol{\beta}^{(t)})$ and let $\widehat{\mathbf{z}}^{(t)} = \mathcal{F}\mathbf{z}^{(t)}$. From the assumption of Theorem 2-2a, $\lim\limits_{t\to\infty} \frac{\mathbf{z}^{(t)}}{\|\mathbf{z}^{(t)}\|}$ exists. Let $\overline{\mathbf{z}}^\infty = \lim_{t\to\infty} \frac{\mathbf{z}^{(t)}}{\|\mathbf{z}^{(t)}\|}$.
   Denote $\widehat{\overline{\mathbf{z}}}^\infty = \mathcal{F}\overline{\mathbf{z}}^\infty$. We get the following by taking Fourier transform of eq. (17)

$$\widehat{\mathbf{z}}^{(t)} = \widehat{\overline{\mathbf{z}}}^\infty p(t) + \widehat{\boldsymbol{\delta}}_{\mathbf{z}}^{(t)} p(t), \tag{44}$$

   where $p(t) = \|\mathbf{z}^{(t)}\| = \|\widehat{\mathbf{z}}^{(t)}\|$ and $\widehat{\boldsymbol{\delta}}_{\mathbf{z}}^{(t)} \to 0$.

3. From Lemma 8, we have that $\exists \{\alpha_n\}_{n \in S_\infty}$ such that $\lim\limits_{t\to\infty} \frac{\mathbf{z}^{(t)}}{\|\mathbf{z}^{(t)}\|} = \sum\limits_{n \in S_\infty} \alpha_n y_n \mathbf{x}_n$. Thus,

$$\widehat{\overline{\mathbf{z}}}^\infty = \sum_{n \in S_\infty} \alpha_n y_n \widehat{\mathbf{x}}_n. \tag{45}$$

**KKT conditions for optimality**   We want to show that a positive scaling of $\overline{\boldsymbol{\beta}}^\infty \propto \mathcal{P}_{conv}(\overline{\mathbf{w}}^\infty)$, denoted by $\widetilde{\boldsymbol{\beta}}^\infty = \gamma \mathcal{P}_{conv}(\overline{\mathbf{w}}^\infty)$ is a first order stationary point of eq. (10), repeated below,

$$\min_{\boldsymbol{\beta}} \|\widehat{\boldsymbol{\beta}}\|_{2/L} \text{ s.t. } \forall n, \ y_n \langle \boldsymbol{\beta}, \mathbf{x}_n \rangle \geq 1.$$

Recall the KKT conditions discussed in Section 3. The first order stationary points, or sub-stationary points, of (10) are the set of feasible predictors $\boldsymbol{\beta}$ such that $\exists \{\alpha_n \geq 0\}_{n=1}^N$ satisfying the following: $\forall n, \ y_n \langle \mathbf{x}_n, \boldsymbol{\beta} \rangle > 1 \implies \alpha_n = 0$, and

$$\sum_n \alpha_n y_n \widehat{\mathbf{x}}_n \in \partial^\circ \|\widehat{\boldsymbol{\beta}}\|_p, \tag{46}$$

where $\partial^\circ$ denotes the local sub-differential (or Clarke's sub-differential) operator defined as $\partial^\circ f(\boldsymbol{\beta}) = \text{conv}\{\mathbf{v} : \exists (\mathbf{z}_k)_k \text{ s.t. } \mathbf{z}_k \to \boldsymbol{\beta} \text{ and } \nabla f(\mathbf{z}_k) \to \mathbf{v}\}$.

For $p = 1$ and $\widehat{\boldsymbol{\beta}}$ represented in polar form as $\widehat{\boldsymbol{\beta}} = |\widehat{\boldsymbol{\beta}}| e^{i\phi_{\widehat{\beta}}} \in \mathbb{C}^D$, $\|\widehat{\boldsymbol{\beta}}\|_p$ is convex and the local sub-differential is indeed the global sub-differential given by,

$$\partial^\circ \|\widehat{\boldsymbol{\beta}}\|_1 = \{\mathbf{z} : \forall d, \ |\mathbf{z}[d]| \leq 1 \text{ and } \widehat{\boldsymbol{\beta}}[d] \neq 0 \implies \mathbf{z}[d] = e^{i\phi_{\widehat{\beta}}[d]}\}. \tag{47}$$

For $p < 1$, the local sub-differential of $\|\widehat{\boldsymbol{\beta}}\|_p$ is given by,

$$\forall p < 1, \quad \partial^\circ \|\widehat{\boldsymbol{\beta}}\|_p = \{\mathbf{z} : \widehat{\boldsymbol{\beta}}[d] \neq 0 \implies \mathbf{z}[d] = p \, e^{i\phi_{\widehat{\beta}}[d]} |\widehat{\boldsymbol{\beta}}[d]|^{p-1}\}. \tag{48}$$

**Showing KKT conditions for $\widetilde{\boldsymbol{\beta}}^\infty \propto \mathcal{P}_{conv}(\overline{\mathbf{w}}^\infty)$.**   As we showed proof of Theorem 4, since $\mathcal{P}_{conv}(\overline{\mathbf{w}}^\infty)$ has strictly positive margin, using homogeneity of $\mathcal{P}_{conv}$, we can scale $\mathcal{P}_{conv}(\overline{\mathbf{w}}^\infty)$ to get $\widetilde{\boldsymbol{\beta}}^\infty = \gamma \mathcal{P}_{conv}(\overline{\mathbf{w}}^\infty)$ with unit margin, i.e., $\forall n, \ y_n \langle \mathbf{x}_n, \widetilde{\boldsymbol{\beta}}^\infty \rangle \geq 1$. For dual variables, we again use a positive scaling of $\alpha_n$ from Lemma 8, such that $\overline{\mathbf{z}}^\infty = \sum_{n \in S_\infty} \alpha_n y_n \mathbf{x}_n$.

In order to prove the theorem, we need to show that for some positive scalar $\overline{\gamma}$, $\overline{\gamma} \widehat{\overline{\mathbf{z}}}^\infty \in \partial^\circ \|\widehat{\boldsymbol{\beta}}\|_{2/L}$, i.e., satisfies the conditions in eq. (47) and (48), for $L = 2$ and $L > 2$, respectively.

We start from the stationarity condition in the parameter space in eq. (26) of Theorem 4. For some positive scalar $\overline{\gamma}$, we have

$$\overline{\mathbf{w}}^\infty = \overline{\gamma} \nabla_{\mathbf{w}} \mathcal{P}_{conv}(\overline{\mathbf{w}}^\infty) \overline{\mathbf{z}}^\infty. \tag{49}$$

We will now special case the above equation for fully width convolutional networks.

From Lemma 3, we have that for all $\mathbf{w} = [\mathbf{w}_l \in \mathbb{R}^D]$, we have $\mathcal{P}_{conv}(\mathbf{w}) = \mathcal{F}^* \mathcal{P}_{diag}(\mathcal{F}\mathbf{w})$ where $\mathcal{F}$ and $\mathcal{F}^*$ denote discrete Fourier matrix and its inverse in appropriate dimensions. Let $\{e_d\}_{d=1}^D$

denote the standard basis in $\mathbb{R}^D$. We first note that for all $l = 1, 2, \ldots, L$ and for all $d = 1, 2, \ldots, D$, the following holds

$$\mathcal{P}_{conv}(\mathbf{w})[d] = e_d^\top \mathcal{F}^* \mathcal{P}_{diag}(\mathcal{F}\mathbf{w}) = e_d^\top \mathcal{F}^* \left( \odot_{l'=1}^{L-1} \widehat{\mathbf{w}}_{l'} \right) \tag{50}$$

$$= e_d^\top \mathcal{F}^* \left( \prod_{l' \neq l} \mathrm{diag}(\widehat{\mathbf{w}}_{l'}) \right) \mathcal{F}\mathbf{w}_l = \left\langle \mathbf{w}_l, \mathcal{F}^* \left( \prod_{l' \neq l} \mathrm{diag}(\widehat{\mathbf{w}}_{l'}^*) \right) \mathcal{F} e_d \right\rangle. \tag{51}$$

$$\implies \nabla_{\mathbf{w}_l} \mathcal{P}_{conv}(\mathbf{w})[:, d] = \mathcal{F}^* \left( \prod_{l' \neq l} \mathrm{diag}(\widehat{\mathbf{w}}_{l'}^*) \right) \mathcal{F} e_d. \tag{52}$$

This implies, for $l = 1, 2, \ldots, L$ and any $\mathbf{z} \in \mathbb{R}^D$, we have

$$\nabla_{\mathbf{w}_l} \mathcal{P}_{conv}(\mathbf{w}) \mathbf{z} = \sum_d \nabla_{\mathbf{w}_l} \mathcal{P}_{conv}(\mathbf{w})[:, d] \mathbf{z}[d] = \mathcal{F}^* \left( \prod_{l' \neq l} \mathrm{diag}(\widehat{\mathbf{w}}_{l'}^*) \right) \mathcal{F} \mathbf{z}. \tag{53}$$

Substituting the above equation in eq. (49), we have,

$$\widehat{\overline{\mathbf{w}}}_l^\infty = \mathcal{F} \overline{\mathbf{w}}_l^\infty = \overline{\gamma} \mathcal{F} \nabla_{\mathbf{w}_l} \mathcal{P}_{conv}(\overline{\mathbf{w}}^\infty) \overline{\mathbf{z}}^\infty = \overline{\gamma} \left( \odot_{l' \neq l} \widehat{\overline{\mathbf{w}}}_{l'}^{\infty*} \right) \odot \widehat{\overline{\mathbf{z}}}^\infty, \tag{54}$$

where $\widehat{\overline{\mathbf{w}}}_{l'}^{\infty*}$ denotes the complex conjugate of $\widehat{\overline{\mathbf{w}}}_{l'}^\infty$.

Let $\widehat{\overline{\boldsymbol{\beta}}}^\infty = \mathcal{P}_{diag}(\widehat{\overline{\mathbf{w}}}^\infty)$. The above equation, further implies, for all $l$

$$|\widehat{\overline{\mathbf{w}}}_l^\infty|^2 = \widehat{\overline{\mathbf{w}}}_l^{\infty*} \odot \widehat{\overline{\mathbf{w}}}_l^\infty = \overline{\gamma} \widehat{\overline{\boldsymbol{\beta}}}^{\infty*} \odot \widehat{\overline{\mathbf{z}}}^\infty = \overline{\gamma} |\widehat{\overline{\boldsymbol{\beta}}}^\infty| \odot |\widehat{\overline{\mathbf{z}}}^\infty| e^{i(\phi_{\widehat{\overline{\mathbf{z}}}^\infty} - \phi_{\widehat{\overline{\boldsymbol{\beta}}}^\infty})} \tag{55}$$

In eq. (55), since the LHS is a real number, we have that for all $d$ such that $|\widehat{\overline{\boldsymbol{\beta}}}^\infty[d]| > 0$

$$e^{i\phi_{\widehat{\overline{\mathbf{z}}}^\infty}[d]} = e^{i\phi_{\widehat{\overline{\boldsymbol{\beta}}}^\infty}[d]}. \tag{56}$$

Also, by multiplying the LHS of eq. (55) across all $l$ and taking $L$th root over positive scalars, we have for $d = 0, 1, \ldots, D - 1$,

$$\left| \widehat{\overline{\boldsymbol{\beta}}}^\infty[d] \right|^{2/L} = \overline{\gamma} \left| \widehat{\overline{\boldsymbol{\beta}}}^\infty[d] \right| \, |\widehat{\overline{\mathbf{z}}}^\infty[d]|, \tag{57}$$

Finally, let $\gamma$ be a positive scaling of $\overline{\boldsymbol{\beta}}^\infty$ such that $\widetilde{\boldsymbol{\beta}}^\infty = \gamma \overline{\boldsymbol{\beta}}^\infty$ has unit margin. Let $\widehat{\widetilde{\boldsymbol{\beta}}}^\infty = \mathcal{F} \widetilde{\boldsymbol{\beta}}^\infty = \gamma \widehat{\overline{\boldsymbol{\beta}}}^\infty$. Since $\overline{\gamma}$ is arbitrary positive scalar, redefining as $\overline{\gamma} \leftarrow \frac{2}{L} \gamma^{2/L-1} \overline{\gamma}$, we have from eq. (56)-(57),

$$\forall d \text{ s.t. } \left| \widehat{\widetilde{\boldsymbol{\beta}}}^\infty[d] \right| \neq 0, \quad \overline{\gamma} \widehat{\overline{\mathbf{z}}}[d] = e^{i\phi_{\widehat{\widetilde{\boldsymbol{\beta}}}}[d]} \left| \widehat{\widetilde{\boldsymbol{\beta}}}^\infty[d] \right|^{2/L-1} \tag{58}$$

### C.1.1 Case of $L > 2$ or $p = {}^2\!/\!{}_L < 1$

For $p = {}^2\!/\!{}_L < 1$, since $\widehat{\overline{\mathbf{z}}}^\infty = \sum_{n \in S_\infty} \alpha_n y_n \widehat{\mathbf{x}}_n$, eq. (58) is indeed the first order stationarity condition for eq. (10) as described in eq. (11) and (13).

### C.1.2 Case of $L = 2$ or $p = {}^2\!/\!{}_L = 1$

For the case of $p = 1$, in addition to eq. (58), we need to show that $\overline{\gamma} |\widehat{\overline{\mathbf{z}}}^\infty| \leq 1$. From eq. (58), for $L = 2$ we have $\left| \widehat{\widetilde{\boldsymbol{\beta}}}^\infty[d] \right| \neq 0 \implies \overline{\gamma} |\widehat{\overline{\mathbf{z}}}^\infty[d]| = 1$.

We need to further show that $\forall d$ s.t. $\left| \widehat{\widetilde{\boldsymbol{\beta}}}^\infty[d] \right| \propto \left| \widehat{\widetilde{\boldsymbol{\beta}}}^\infty[d] \right| = 0, \overline{\gamma} |\widehat{\overline{\mathbf{z}}}^\infty[d]| \leq 1$.

**Showing** $\forall d, \left|\widehat{\overline{\boldsymbol{\beta}}}^{\infty}[d]\right| = 0 \implies \overline{\gamma}|\widehat{\overline{\mathbf{z}}}^{\infty}[d]| \leq 1$

Using Lemma 9 for for the special case of 2–layer linear convolutional network, for $\forall d$,

$$\begin{aligned}
\Delta\widehat{\mathbf{w}}_1^{(t)}[d] &= \eta_t \widehat{\mathbf{z}}^{(t)}[d]\, \widehat{\mathbf{w}}_2^{(t)*}[d], \\
\Delta\widehat{\mathbf{w}}_2^{(t)}[d] &= \eta_t \widehat{\mathbf{z}}^{(t)}[d]\, \widehat{\mathbf{w}}_1^{(t)*}[d].
\end{aligned} \tag{59}$$

Recall: for $l = 1, 2$, $\frac{\widehat{\mathbf{w}}_l^{(t)}}{g(t)} \to \widehat{\overline{\mathbf{w}}}_l^{\infty}$, $\frac{\widehat{\mathbf{z}}^{(t)}}{p(t)} \to \widehat{\overline{\mathbf{z}}}^{\infty}$, $\widehat{\boldsymbol{\beta}}^{(t)} = \widehat{\mathbf{w}}_1^{(t)} \odot \widehat{\mathbf{w}}_2^{(t)}$ and $\frac{\widehat{\boldsymbol{\beta}}^{(t)}}{g(t)^2} \to \widehat{\overline{\boldsymbol{\beta}}}^{\infty} = \widehat{\overline{\mathbf{w}}}_1^{\infty} \odot \widehat{\overline{\mathbf{w}}}_2^{\infty}$.

Further, from eq. (55), we have $\forall d$, $|\widehat{\overline{\mathbf{w}}}_1^{\infty}[d]|^2 = |\widehat{\overline{\mathbf{w}}}_2^{\infty}[d]|^2$, and hence

$$|\widehat{\overline{\mathbf{w}}}_1^{\infty}[d]| = |\widehat{\overline{\mathbf{w}}}_2^{\infty}[d]| = \sqrt{|\widehat{\overline{\boldsymbol{\beta}}}^{\infty}[d]|}. \tag{60}$$

From the convergence of complex numbers, we have the following:

1. $\forall d$ such that $|\widehat{\overline{\mathbf{z}}}^{\infty}[d]| \neq 0$, we have

$$\frac{|\widehat{\mathbf{z}}^{(t)}[d]|}{p(t)} \to |\widehat{\overline{\mathbf{z}}}^{\infty}[d]| \text{ and } e^{i\phi_{\widehat{\mathbf{z}}^{(t)}[d]}} \to e^{i\phi_{\widehat{\overline{\mathbf{z}}}^{\infty}[d]}}. \tag{61}$$

2. $\forall d$ such that $|\widehat{\overline{\boldsymbol{\beta}}}^{\infty}[d]| \neq 0$, we have $|\widehat{\overline{\mathbf{w}}}_1^{\infty}[d]|, |\widehat{\overline{\mathbf{w}}}_2^{\infty}[d]| \neq 0$, and the following holds

$$\begin{aligned}
\text{for } l = 1, 2, \quad & \frac{|\widehat{\mathbf{w}}_l^{(t)}[d]|}{g(t)} \to |\widehat{\overline{\mathbf{w}}}_l^{\infty}[d]| \text{ and } e^{i\phi_{\widehat{\mathbf{w}}_l^{(t)}[d]}} \to e^{i\phi_{\widehat{\overline{\mathbf{w}}}_l^{\infty}[d]}} \\
& \frac{|\widehat{\boldsymbol{\beta}}^{(t)}[d]|}{g(t)^2} \to |\widehat{\overline{\boldsymbol{\beta}}}^{\infty}[d]| \text{ and } e^{i\phi_{\widehat{\boldsymbol{\beta}}^{(t)}[d]}} \to e^{i\phi_{\widehat{\overline{\boldsymbol{\beta}}}^{\infty}[d]}} = e^{i\phi_{\widehat{\overline{\mathbf{w}}}_1^{\infty}[d]}} \cdot e^{i\phi_{\widehat{\overline{\mathbf{w}}}_2^{\infty}[d]}} \\
& \overline{\gamma}|\widehat{\overline{\mathbf{z}}}^{\infty}[d]| = 1 \text{ and } e^{i\phi_{\widehat{\overline{\mathbf{z}}}^{\infty}[d]}} = e^{i\phi_{\widehat{\overline{\boldsymbol{\beta}}}^{\infty}[d]}},
\end{aligned} \tag{62}$$

where the last equation follows from eq. (56).

3. $\forall d$ such that $|\widehat{\overline{\boldsymbol{\beta}}}^{\infty}[d]| = 0$, from eq. (60), we have $|\widehat{\overline{\mathbf{w}}}_1^{\infty}[d]| = |\widehat{\overline{\mathbf{w}}}_2^{\infty}[d]| = 0$.

In the remainder of the proof, we only consider $d$ with $|\widehat{\overline{\mathbf{z}}}^{\infty}[d]| \neq 0$.

Consider $\mathbf{u}_d^{(t)}$ defined below,

$$\mathbf{u}_d^{(t)} := \widehat{\mathbf{w}}_1^{(t)}[d] \cdot e^{-i\phi_{\widehat{\overline{\mathbf{z}}}^{\infty}[d]}} + \widehat{\mathbf{w}}_2^{(t)*}[d]. \tag{63}$$

Since for $l = 1, 2$, $\mathbf{w}_l^{(t)}/g(t) \to \overline{\mathbf{w}}_l^{\infty}$, we have the following:

$$\lim_{t \to \infty} \frac{\mathbf{u}_d^{(t)}}{g(t)} = \widehat{\overline{\mathbf{w}}}_1^{\infty}[d] \cdot e^{-i\phi_{\widehat{\overline{\mathbf{z}}}^{\infty}[d]}} + \widehat{\overline{\mathbf{w}}}_2^{\infty*}[d] \overset{(a)}{=} \begin{cases} 0 & \text{if } |\widehat{\overline{\boldsymbol{\beta}}}^{\infty}[d]| = 0 \\ e^{-i\phi_{\widehat{\overline{\mathbf{w}}}_2^{\infty}[d]}}\left[|\widehat{\overline{\mathbf{w}}}_1^{\infty}[d]| + |\widehat{\overline{\mathbf{w}}}_2^{\infty}[d]|\right] & \text{if } |\widehat{\overline{\boldsymbol{\beta}}}^{\infty}[d]| > 0 \end{cases}$$

$$\overset{(b)}{=} \begin{cases} 0 & \text{if } |\widehat{\overline{\boldsymbol{\beta}}}^{\infty}[d]| = 0 \\ 2e^{-i\phi_{\widehat{\overline{\mathbf{w}}}_2^{\infty}[d]}}\sqrt{|\widehat{\overline{\boldsymbol{\beta}}}^{\infty}[d]|} & \text{if } |\widehat{\overline{\boldsymbol{\beta}}}^{\infty}[d]| > 0 \end{cases}, \tag{64}$$

where $(a)$ follows from using $e^{i\phi_{\widehat{\overline{\mathbf{z}}}^{\infty}[d]}} = e^{i\phi_{\widehat{\overline{\boldsymbol{\beta}}}^{\infty}[d]}} = e^{i\phi_{\widehat{\overline{\mathbf{w}}}_1^{\infty}[d]}} \cdot e^{i\phi_{\widehat{\overline{\mathbf{w}}}_2^{\infty}[d]}}$ whenever $\overline{\boldsymbol{\beta}}^{\infty}[d] \neq 0$ (from eq. (62)), and $(b)$ follows from eq. (60).

**Step 1.** *Dynamics of $\mathbf{u}_d^{(t)}$:* Now looking at the dynamics of $\mathbf{u}_d$, using eq. (59) we have that

$$\begin{aligned}
\mathbf{u}_d^{(t+1)} &= \mathbf{u}_d^{(t)} + e^{-i\phi_{\widehat{\overline{\mathbf{z}}}^{\infty}[d]}} \cdot \eta_t \widehat{\mathbf{z}}^{(t)}[d]\, \widehat{\mathbf{w}}_2^{(t)*}[d] + \eta_t \widehat{\mathbf{z}}^{(t)*}[d]\, \widehat{\mathbf{w}}_1^{(t)}[d] \\
&= \mathbf{u}_d^{(t)} + \eta_t |\widehat{\mathbf{z}}^{(t)}[d]|\left[e^{i\left(\phi_{\widehat{\mathbf{z}}^{(t)}[d]} - \phi_{\widehat{\overline{\mathbf{z}}}^{\infty}[d]}\right)}\widehat{\mathbf{w}}_2^{(t)*}[d] + e^{-i\left(\phi_{\widehat{\mathbf{z}}^{(t)}[d]} - \phi_{\widehat{\overline{\mathbf{z}}}^{\infty}[d]}\right)} \cdot \widehat{\mathbf{w}}_1^{(t)}[d] \cdot e^{-i\phi_{\widehat{\overline{\mathbf{z}}}^{\infty}[d]}}\right]
\end{aligned}$$

Additionally, since $e^{i\phi_{\widehat{\mathbf{z}}^{(t)}[d]}} \to e^{i\phi_{\widehat{\widetilde{\mathbf{z}}}^\infty[d]}}$, we can write $e^{\pm i\left(\phi_{\widehat{\mathbf{z}}^{(t)}[d]} - \phi_{\widehat{\widetilde{\mathbf{z}}}^\infty[d]}\right)} = 1 + \boldsymbol{\delta}_{1,d}^{(t)} \pm i\boldsymbol{\delta}_{2,d}^{(t)}$ where $\boldsymbol{\delta}_{1,d}^{(t)}, \boldsymbol{\delta}_{2,d}^{(t)} \to 0$ are real scalars. Substituting in above equation and rearranging the terms, we have

$$
\begin{aligned}
\mathbf{u}_d^{(t+1)} &= \left[1 + \eta_t |\widehat{\mathbf{z}}^{(t)}[d]|(1 + \boldsymbol{\delta}_{1,d}^{(t)})\right] \mathbf{u}_d^{(t)} + i\boldsymbol{\delta}_{2,d}^{(t)}\eta_t |\widehat{\mathbf{z}}^{(t)}[d]| \left[\widehat{\mathbf{w}}_2^{(t)*}[d] - \widehat{\mathbf{w}}_1^{(t)}[d] \cdot e^{-i\phi_{\widehat{\widetilde{\mathbf{z}}}^\infty[d]}}\right] \\
&\overset{(a)}{:=} \left[1 + \eta_t |\widehat{\mathbf{z}}^{(t)}[d]|(1 + \boldsymbol{\delta}_{1,d}^{(t)})\right] \mathbf{u}_d^{(t)} + \eta_t |\widehat{\mathbf{z}}^{(t)}[d]| \boldsymbol{\tau}_d^{(t)},
\end{aligned}
\tag{65}
$$

where in $(a)$ we define $\boldsymbol{\tau}_d^{(t)} = i\boldsymbol{\delta}_{2,d}^{(t)}\left[\widehat{\mathbf{w}}_2^{(t)*}[d] - \widehat{\mathbf{w}}_1^{(t)}[d] \cdot e^{-i\phi_{\widehat{\widetilde{\mathbf{z}}}^\infty[d]}}\right]$.

The following intermediate lemma is proved in Appendix C.1.3.

**Lemma 10.** *Consider $\boldsymbol{\tau}_d^{(t)}$ in eq. (65). For all $d$ such that $\widehat{\widetilde{\mathbf{z}}}^\infty[d] \neq 0$, $\mathbf{u}_d^{(t)} \to \infty$ and $\frac{\boldsymbol{\tau}_d^{(t)}}{\mathbf{u}_d^{(t)}} \to 0$.*

Using the above lemma, we have $\boldsymbol{\delta}_{3,d}^{(t)} \to 0$ such that $\boldsymbol{\tau}_d^{(t)} = \boldsymbol{\delta}_{3,d}^{(t)}\mathbf{u}_d(t)$. Additionally, since $\frac{|\widehat{\mathbf{z}}^{(t)}[d]|}{p(t)} \to |\widehat{\widetilde{\mathbf{z}}}^\infty[d]|$, there exists $\boldsymbol{\delta}_{4,d}^{(t)} \to 0$ such that $|\widehat{\mathbf{z}}^{(t)}[d]| = |\widehat{\widetilde{\mathbf{z}}}^\infty[d]|p(t) + \boldsymbol{\delta}_{4,d}^{(t)}p(t)$. Substituting these representations in eq. (65), we have the following dynamics for $\mathbf{u}_d(t)$,

$$
\begin{aligned}
\mathbf{u}_d^{(t+1)} &= \left[1 + \eta_t p(t)\left(|\widehat{\widetilde{\mathbf{z}}}^\infty[d]| + \boldsymbol{\delta}_{4,d}^{(t)}\right)\left(1 + \boldsymbol{\delta}_{1,d}^{(t)} + \boldsymbol{\delta}_{3,d}^{(t)}\right)\right] \mathbf{u}_d^{(t)} \\
&\overset{(a)}{:=} \left[1 + \eta_t p(t)\left(|\widehat{\widetilde{\mathbf{z}}}^\infty[d]| + \boldsymbol{\delta}_d^{(t)}\right)\right] \mathbf{u}_d^{(t)},
\end{aligned}
\tag{66}
$$

where in $(a)$ we have accumulated all diminishing terms into $\boldsymbol{\delta}_d^{(t)} = \boldsymbol{\delta}_{4,d}^{(t)}\left(1 + \boldsymbol{\delta}_{1,d}^{(t)} + \boldsymbol{\delta}_{3,d}^{(t)}\right) + |\widehat{\widetilde{\mathbf{z}}}^\infty[d]|\left(\boldsymbol{\delta}_{1,d}^{(t)} + \boldsymbol{\delta}_{3,d}^{(t)}\right) \to 0$.

**Step 2.** *Remainder of the proof:* We now prove our theorem by looking the following quantity: For any $d, d'$ with $\widehat{\widetilde{\mathbf{z}}}^\infty[d], \widehat{\widetilde{\mathbf{z}}}^\infty[d'] \neq 0$, define $\kappa_{d,d'}^{(t)} = \left|\frac{\mathbf{u}_d^{(t)}}{\mathbf{u}_{d'}^{(t)}}\right|$.

We will show that whenever $|\widehat{\widetilde{\mathbf{z}}}^\infty[d]| > |\widehat{\widetilde{\mathbf{z}}}^\infty[d']|$, we get $\kappa_{d,d'}^{(t)} \to \infty$. Along with eq. (64), this would imply that $\lim_{t\to\infty} \kappa_{d,d'}^{(t)} = \sqrt{\frac{|\widehat{\widetilde{\boldsymbol{\beta}}}^\infty[d]|}{|\widehat{\widetilde{\boldsymbol{\beta}}}^\infty[d']|}} = \infty$. Hence, for any $d, d'$ with $\widehat{\widetilde{\boldsymbol{\beta}}}^\infty[d] = 0$ and $\widehat{\widetilde{\boldsymbol{\beta}}}^\infty[d'] \neq 0$, we have $\overline{\gamma}|\widehat{\widetilde{\mathbf{z}}}^\infty[d]| \leq \overline{\gamma}|\widehat{\widetilde{\mathbf{z}}}^\infty[d']|$. Moreover from eq.(57)), we know that $\overline{\gamma}|\widehat{\widetilde{\mathbf{z}}}^\infty[d']| = 1$ for all $d'$ with $\widehat{\widetilde{\boldsymbol{\beta}}}^\infty[d'] \neq 0$. This implies $\forall d, \overline{\gamma}|\widehat{\widetilde{\mathbf{z}}}^\infty[d]| \leq 1$ and concludes the proof.

*Showing* $|\widehat{\widetilde{\mathbf{z}}}^\infty[d]| > |\widehat{\widetilde{\mathbf{z}}}^\infty[d']| \implies \kappa_{d,d'}^{(t)} \to \infty$:

For any $2\epsilon > 0$, let $|\widehat{\widetilde{\mathbf{z}}}^\infty[d]| - |\widehat{\widetilde{\mathbf{z}}}^\infty[d']| = 2\epsilon > 0$. We note that the since the loss $\mathcal{L}(\boldsymbol{\beta}^{(t)}) \to 0$, norm of the gradient $p(t) = \|\mathbf{z}^{(t)}\| = \|\widehat{\mathbf{z}}^t\| \to 0$. Hence, for any finite step size sequence $\{\eta_t\}$, there exists $t_1$ such that $\forall t \geq t_1$ and $\forall d$, $\eta_t p(t)\left(|\widehat{\widetilde{\mathbf{z}}}^\infty[d]| + |\boldsymbol{\delta}_d^{(t)}|\right) < 0.5$ and the following inequalities hold,

$$
\kappa_{d,d'}^{(t+1)} = \left|\frac{\mathbf{u}_d^{(t+1)}}{\mathbf{u}_{d'}^{(t+1)}}\right| = \left|\frac{\left(1 + \eta_t\left(|\widehat{\widetilde{\mathbf{z}}}^\infty[d]| + \boldsymbol{\delta}_d^{(t)}\right)p(t)\right)\mathbf{u}_d^{(t)}}{\left(1 + \eta_t\left(|\widehat{\widetilde{\mathbf{z}}}^\infty[d']| + \boldsymbol{\delta}_d^{(t)}\right)p(t)\right)\mathbf{u}_{d'}^{(t)}}\right|
\tag{67}
$$

$$
\geq \frac{\left(1 + \eta_t\left(|\widehat{\widetilde{\mathbf{z}}}^\infty[d]| - |\boldsymbol{\delta}_d^{(t)}|\right)p(t)\right)}{\left(1 + \eta_t\left(|\widehat{\widetilde{\mathbf{z}}}^\infty[d']| + |\boldsymbol{\delta}_{d'}^{(t)}|\right)p(t)\right)}\kappa_{d,d'}^{(t)}
\tag{68}
$$

$$
\overset{(a)}{\geq} \left(1 + \eta_t\left(|\widehat{\widetilde{\mathbf{z}}}^\infty[d]| - |\boldsymbol{\delta}_d^{(t)}|\right)p(t)\right)\left(1 - \eta_t\left(|\widehat{\widetilde{\mathbf{z}}}^\infty[d']| + |\boldsymbol{\delta}_{d'}^{(t)}|\right)p(t)\right)\kappa_{d,d'}^{(t)}
\tag{69}
$$

$$
\overset{(c)}{\geq} \left(1 + \eta_t\left(2\epsilon + \boldsymbol{\delta}_{d,d'}^{(t)}\right)p(t)\right)\kappa_{d,d'}^{(t)},
\tag{70}
$$

where in $(a)$ follows from using $1/(1+x) \geq (1-x)$ for $x < 1$ since $\eta_t p(t) \left( |\widehat{\overline{\mathbf{z}}}^\infty[d]| + |\boldsymbol{\delta}_d^{(t)}| \right) < 0.5$ for all $t \geq t_1$, and in $(c)$, we absorbed all $o(p(t))$ terms as $\boldsymbol{\delta}_{d,d'}^{(t)} p(t)$ for $\boldsymbol{\delta}_{d,d'}^{(t)} \to 0$ and used $|\widehat{\overline{\mathbf{z}}}^\infty[d]| - |\widehat{\overline{\mathbf{z}}}^\infty[d']| = 2\epsilon > 0$.

Since $\boldsymbol{\delta}_{d,d'}^{(t)} \to 0$, for large enough $t_2$ and $t \geq t_2$, we have $|\boldsymbol{\delta}_{d,d'}^{(t)}| < \epsilon$. Thus, for all $t \geq \max\{t_1, t_2\}$,

$$\kappa_{d,d'}^{(t+1)} \geq (1 + \eta_t \epsilon p(t)) \, \kappa_{d,d'}^{(t)}. \tag{71}$$

Further, from the conditions of the theorem, for almost all initializations, $|\widehat{\mathbf{w}}_l^{(0)}[d]| > 0$ for all $d$. For step sizes $\{\eta_t\}$ smaller than the local Lipschitz constant, for all finite $t' < \infty$, we also have $|\mathbf{w}_l^{(t')}[d]| > 0$. Moreover from Lemma 10, we have that $|\mathbf{u}_d^{(t)}|, |\mathbf{u}_{d'}^{(t)}| \to \infty$ and hence $\exists t_3$ such that $\forall t \geq t_3, |\mathbf{u}_d^{(t)}| > 0$, but for any finite $t' < \infty$, $|\mathbf{u}_{d'}^{(t')}| < \infty$. Thus, for $t_0 = \max\{t_1, t_2, t_3\}$, using the above observations, we have that $\kappa_{d,d'}^{(t_0)} = \left| \frac{\mathbf{u}_d^{(t_0)}}{\mathbf{u}_{d'}^{(t_0)}} \right| > 0$.

Now, using eq. (71), for all $t \geq t_0$,

$$\kappa_{d,d'}^{(t+1)} \geq (1 + \eta_t \epsilon p(t))\kappa_{d,d'}^{(t)} = \left( \prod_{u=t_0}^{t} (1 + \eta_u \epsilon p(u)) \right) \kappa_{d,d'}^{(t_0)} \text{ and } \kappa_{d,d'}^{(t_0)} > 0. \tag{72}$$

Finally, we show the following claim:

**Claim 2.** *For any finite $t_0$, finite step-sizes $\{\eta_t\}$, and any $\epsilon > 0$, we have $\prod_{u=t_0}^{t} (1 + \eta_u \epsilon p(u)) \to \infty$.*

*Proof.* Let $\mu = \max_d |\widehat{\overline{\mathbf{z}}}^\infty[d]| + \max_{t > t_0} |\boldsymbol{\delta}_d^{(t)}| < \infty$. From eq. (66), we have that for all $d$,

$$|\mathbf{u}_d^{(t+1)}| \leq (1 + \mu \eta_t p(t))|\mathbf{u}_d^{(t)}| \leq |\mathbf{u}_d^{(t_0)}| \prod_{u=t_0}^{t} (1 + \mu \eta_u p(u)) \leq |\mathbf{u}_d^{(t_0)}| \exp \left( \sum_{u=t_0}^{t} \mu \eta_u p(u) \right).$$

Moreover, we have $\mathbf{u}_d^{(t)} \to \infty$ for at least one $d$, and for any finite step sizes and finite $t_0$, $|\mathbf{u}_d^{(t_0)}| < \infty$. This then implies that for some $\mu < \infty$, $\exp \left( \sum_{u=t_0}^{t} \mu \eta_u p(u) \right) \to \infty \implies \sum_{u=t_0}^{t} \eta_u p(u) \to \infty$. Thus, for any $\epsilon > 0$, we also have $\prod_{u=t_0}^{t} (1 + \epsilon \eta_u p(u)) \geq \epsilon \sum_{u=t_0}^{t} \eta_u p(u) \to \infty$. $\square$

From eq. (72) and above claim, we conclude that for all $d, d'$, if $|\widehat{\overline{\mathbf{z}}}^\infty[d]| > |\widehat{\overline{\mathbf{z}}}^\infty[d']|$, then $\kappa_{d,d'}^{(t)} \to \infty$.

This completes the proof of the theorem. $\square$

### C.1.3   Proof of Lemma 10

**Lemma 10.** *Consider $\boldsymbol{\tau}_d^{(t)}$ in eq. (65). For all $d$ such that $\widehat{\overline{\mathbf{z}}}^\infty[d] \neq 0$, $\mathbf{u}_d^{(t)} \to \infty$ and $\frac{\boldsymbol{\tau}_d^{(t)}}{\mathbf{u}_d^{(t)}} \to 0$.*

*Proof.* Recalling $\boldsymbol{\tau}_d^{(t)}$ from eq. (65) and $\mathbf{u}_d^{(t)}$ from eq. (63), we have the following:

$$\frac{\boldsymbol{\tau}_d^{(t)}}{\mathbf{u}_d^{(t)}} = i\boldsymbol{\delta}_{2,d}^{(t)} \frac{\widehat{\mathbf{w}}_2^{(t)*}[d] - \widehat{\mathbf{w}}_1^{(t)}[d] \cdot e^{-i\phi_{\widehat{\overline{\mathbf{z}}}^\infty}[d]}}{\widehat{\mathbf{w}}_1^{(t)}[d] \cdot e^{-i\phi_{\widehat{\overline{\mathbf{z}}}^\infty}[d]} + \widehat{\mathbf{w}}_2^{(t)*}[d]} = i\boldsymbol{\delta}_{2,d}^{(t)} \frac{1 - \frac{|\widehat{\mathbf{w}}_1^{(t)}[d]|}{|\widehat{\mathbf{w}}_2^{(t)}[d]|} \cdot e^{-i\phi_{\widehat{\overline{\mathbf{z}}}^\infty}[d] + i\phi_{\widehat{\boldsymbol{\beta}}^{(t)}}[d]}}{1 + \frac{|\widehat{\mathbf{w}}_1^{(t)}[d]|}{|\widehat{\mathbf{w}}_2^{(t)}[d]|} \cdot e^{-i\phi_{\widehat{\overline{\mathbf{z}}}^\infty}[d] + i\phi_{\widehat{\boldsymbol{\beta}}^{(t)}}[d]}} \tag{73}$$

For all $d$ if $\widehat{\overline{\boldsymbol{\beta}}}^\infty[d] = \widehat{\overline{\mathbf{w}}}_1^\infty[d] \cdot \widehat{\overline{\mathbf{w}}}_2^\infty[d] \neq 0$, the it is straightforward to see that $\frac{|\widehat{\mathbf{w}}_1^{(t)}[d]|}{|\widehat{\mathbf{w}}_2^{(t)}[d]|} = \frac{|\widehat{\mathbf{w}}_1^{(t)}[d]|/g(t)}{|\widehat{\mathbf{w}}_2^{(t)}[d]|/g(t)} \to \frac{|\widehat{\overline{\mathbf{w}}}_1^\infty[d]|}{|\widehat{\overline{\mathbf{w}}}_2^\infty[d]|} = 1$ (from eq. (60)), and also that $e^{-i\phi_{\widehat{\overline{\mathbf{z}}}^\infty}[d] + i\phi_{\widehat{\boldsymbol{\beta}}^{(t)}}[d]} \to e^{-i\phi_{\widehat{\overline{\mathbf{z}}}^\infty}[d] + i\phi_{\widehat{\overline{\boldsymbol{\beta}}}^\infty}[d]} = 1$ (from eq. (62)). This along with eq. (73) gives us $\frac{\boldsymbol{\tau}_d^{(t)}}{\mathbf{u}_d^{(t)}} \to 0$.

Moreover, since $|\widehat{\boldsymbol{\beta}}^{(t)}[d]| \to \infty$, we have $|\widehat{\mathbf{w}}_2^{(t)}[d]|$ or $|\widehat{\mathbf{w}}_2^{(t)}[d]| \to \infty$. Further, using $\mathrm{e}^{-\mathrm{i}\phi_{\widehat{\mathbf{z}}^\infty}[d]+\mathrm{i}\phi_{\widehat{\boldsymbol{\beta}}^{(t)}[d]}} \to 1$, we have $|\mathbf{u}_d^{(t)}| = |\widehat{\mathbf{w}}_2^{(t)}[d]| + |\widehat{\mathbf{w}}_1^{(t)}[d]|\mathrm{e}^{-\mathrm{i}\phi_{\widehat{\mathbf{z}}^\infty}[d]+\mathrm{i}\phi_{\widehat{\boldsymbol{\beta}}^{(t)}[d]}} \to \infty$.

We now only need to show that these results also hold for $d$ such that $\overline{\widehat{\boldsymbol{\beta}}}^\infty[d] = 0$. Recall from the assumptions of the theorem that even when $\overline{\widehat{\boldsymbol{\beta}}}^\infty[d] = 0$, $\exists \phi_{\widehat{\boldsymbol{\beta}}^\infty[d]} \in [0, 2\pi)$ such that $\mathrm{e}^{\mathrm{i}\phi_{\widehat{\boldsymbol{\beta}}^{(t)}[d]}} \to \mathrm{e}^{\mathrm{i}\phi_{\widehat{\boldsymbol{\beta}}^\infty[d]}}$. We now prove the lemma by showing the following steps for $d$ such that $\overline{\widehat{\boldsymbol{\beta}}}^\infty[d] = 0$. :

Step 1. Show $\frac{|\widehat{\mathbf{w}}_1^{(t)}[d]|}{|\widehat{\mathbf{w}}_2^{(t)}[d]|} \to 1$.

Step 2. Show $\mathrm{Re}(\mathrm{e}^{-\mathrm{i}\phi_{\widehat{\mathbf{z}}^\infty[d]}+\mathrm{i}\phi_{\widehat{\boldsymbol{\beta}}^\infty[d]}}) = 2\cos\left(\phi_{\widehat{\mathbf{z}}^\infty[d]} - \phi_{\widehat{\boldsymbol{\beta}}^\infty[d]}\right) \geq 0$.

**Proof of lemma assuming Step 1 and Step 2 hold**  The above steps would imply that in eq. (73),

- the denominator satisfies

$$\left|1 + \frac{|\widehat{\mathbf{w}}_1^{(t)}[d]|}{|\widehat{\mathbf{w}}_2^{(t)}[d]|} \cdot \mathrm{e}^{-\mathrm{i}\phi_{\widehat{\mathbf{z}}^\infty[d]}+\mathrm{i}\phi_{\widehat{\boldsymbol{\beta}}^{(t)}[d]}}\right| \to \left|1 + \mathrm{e}^{-\mathrm{i}\phi_{\widehat{\mathbf{z}}^\infty[d]}+\mathrm{i}\phi_{\widehat{\boldsymbol{\beta}}^\infty[d]}}\right|$$

$$\geq \left|1 + \mathrm{Re}(\mathrm{e}^{-\mathrm{i}\phi_{\widehat{\mathbf{z}}^\infty[d]}+\mathrm{i}\phi_{\widehat{\boldsymbol{\beta}}^\infty[d]}})\right| \geq 1. \tag{74}$$

- the numerator satisfies

$$\left|\boldsymbol{\delta}_{2,d}^{(t)}\left(1 - \frac{|\widehat{\mathbf{w}}_1^{(t)}[d]|}{|\widehat{\mathbf{w}}_2^{(t)}[d]|} \cdot \mathrm{e}^{-\mathrm{i}\phi_{\widehat{\mathbf{z}}^\infty[d]}+\mathrm{i}\phi_{\widehat{\boldsymbol{\beta}}^{(t)}[d]}}\right)\right| \leq |\boldsymbol{\delta}_{2,d}^{(t)}|\left|1 + \frac{|\widehat{\mathbf{w}}_1^{(t)}[d]|}{|\widehat{\mathbf{w}}_2^{(t)}[d]|}\right| \to 0. \tag{75}$$

These eqs. along with eq. (73) in turn prove the lemma, *i.e.*, $\frac{\boldsymbol{\tau}_d^{(t)}}{\mathbf{u}_d^{(t)}} \to 0$ and $|\mathbf{u}_d^{(t)}| \to \infty$.

**Showing Step 1 and Step 2**

**Step** 1. *Show* $\frac{|\widehat{\mathbf{w}}_1^{(t)}[d]|}{|\widehat{\mathbf{w}}_2^{(t)}[d]|} \to 1$.

From the dynamics of $\widehat{\mathbf{w}}_l^{(t)}[d]$ from eq. (59), we have the following,

$$|\widehat{\mathbf{w}}_1^{(t+1)}[d]|^2 = |\widehat{\mathbf{w}}_1^{(t)}[d]|^2 + \eta_t\widehat{\mathbf{z}}^{(t)}[d] \cdot \widehat{\boldsymbol{\beta}}^{(t)*}[d] + \eta_t\widehat{\mathbf{z}}^{(t)*}[d] \cdot \widehat{\boldsymbol{\beta}}^{(t)}[d] + \eta_t^2|\widehat{\mathbf{z}}^{(t)}[d]|^2|\widehat{\mathbf{w}}_2^{(t)}[d]|^2$$

$$|\widehat{\mathbf{w}}_2^{(t+1)}[d]|^2 = |\widehat{\mathbf{w}}_2^{(t)}[d]|^2 + \eta_t\widehat{\mathbf{z}}^{(t)}[d] \cdot \widehat{\boldsymbol{\beta}}^{(t)*}[d] + \eta_t\widehat{\mathbf{z}}^{(t)*}[d] \cdot \widehat{\boldsymbol{\beta}}^{(t)}[d] + \eta_t^2|\widehat{\mathbf{z}}^{(t)}[d]|^2|\widehat{\mathbf{w}}_1^{(t)}[d]|^2 \tag{76}$$

Note that since $|\widehat{\mathbf{z}}^{(t)}[d]|^2 \to 0$ and $\eta_t$ are finite, we have that $\exists t_1$ such that for all $t \geq t_1$, $\eta_t|\widehat{\mathbf{z}}^{(t)}[d]|^2 \leq 1$. From the above equation, we have the following for $t \geq t_1$,

$$\left||\widehat{\mathbf{w}}_1^{(t+1)}[d]|^2 - |\widehat{\mathbf{w}}_2^{(t+1)}[d]|^2\right| = \left|\left(1 - \eta_t^2|\widehat{\mathbf{z}}^{(t)}[d]|^2\right)\left(|\widehat{\mathbf{w}}_1^{(t)}[d]|^2 - \widehat{\mathbf{w}}_2^{(t)}[d]|^2\right)\right|$$

$$\overset{(a)}{=} \left(\prod_{u=t_1}^t \left(1 - \eta_u^2|\widehat{\mathbf{z}}^{(u)}[d]|^2\right)\right)\left||\widehat{\mathbf{w}}_1^{(t_1)}[d]|^2 - \widehat{\mathbf{w}}_2^{(t_1)}[d]|^2\right| \tag{77}$$

$$\leq \left||\widehat{\mathbf{w}}_1^{(t_1)}[d]|^2 - \widehat{\mathbf{w}}_2^{(t_1)}[d]|^2\right| < \infty,$$

where $(a)$ follows from iterating over $t$ and using $|\widehat{\mathbf{z}}^{(t)}[d]|^2 \leq 1$ for $t \geq t_1$.

Since $|\widehat{\boldsymbol{\beta}}^{(t)}[d]| = |\widehat{\mathbf{w}}_1^{(t)}[d]| \cdot |\widehat{\mathbf{w}}_2^{(t)}[d]| \to \infty$, at least one of $|\widehat{\mathbf{w}}_1^{(t)}[d]|, |\widehat{\mathbf{w}}_2^{(t)}[d]|$ must diverge. Without loss of generality, let $|\widehat{\mathbf{w}}_2^{(t)}[d]| \to \infty$. Let $c(t) := |\widehat{\mathbf{w}}_1^{(t)}[d]|^2 - |\widehat{\mathbf{w}}_2^{(t)}[d]|^2$ with $|c(t)| < \infty$. We have

$$\frac{|\widehat{\mathbf{w}}_1^{(t)}[d]|^2}{|\widehat{\mathbf{w}}_2^{(t)}[d]|^2} = 1 + \frac{c(t)}{|\widehat{\mathbf{w}}_2^{(t)}[d]|^2} \overset{(a)}{\to} 1, \tag{78}$$

where the convergence in $(a)$ follows since $|c(t)| < \infty$ (from eq. (76)) and $|\widehat{\mathbf{w}}_2^{(t)}[d]| \to \infty$.

**Step** 2. *Show* $Re(\mathrm{e}^{-\mathrm{i}\phi_{\widehat{\widehat{\mathbf{z}}}^\infty[d]}+\mathrm{i}\phi_{\widehat{\widehat{\boldsymbol{\beta}}}^\infty[d]}}) = 2\cos\left(\phi_{\widehat{\widehat{\mathbf{z}}}^\infty[d]} - \phi_{\widehat{\widehat{\boldsymbol{\beta}}}^\infty[d]}\right) \geq 0.$

Note that from Step 1 above, we have that $\frac{|\widehat{\mathbf{w}}_1^{(t)}[d]|^2}{|\widehat{\mathbf{w}}_2^{(t)}[d]|^2} \to 1$, which implies $\frac{|\widehat{\mathbf{w}}_1^{(t)}[d]|^2+|\widehat{\mathbf{w}}_2^{(t)}[d]|^2}{2|\widehat{\boldsymbol{\beta}}^{(t)}[d]|} = \frac{|\widehat{\mathbf{w}}_1^{(t)}[d]|^2+|\widehat{\mathbf{w}}_2^{(t)}[d]|^2}{2|\widehat{\mathbf{w}}_1^{(t)}[d]|\cdot|\widehat{\mathbf{w}}_2^{(t)}[d]|} \to 1.$ Thus, there exists $\boldsymbol{\delta}_{1,d}^{(t)} \to 0$, such that

$$|\widehat{\mathbf{w}}_1^{(t)}[d]|^2 + |\widehat{\mathbf{w}}_2^{(t)}[d]|^2 = 2|\widehat{\boldsymbol{\beta}}^{(t)}[d]| \cdot (1 + \boldsymbol{\delta}_{1,d}^{(t)}). \tag{79}$$

Also, from eq. (44), there exists $\boldsymbol{\delta}_{2,d}^{(t)} \to 0$, such that

$$\widehat{\mathbf{z}}^{(t)}[d] = \widehat{\widehat{\mathbf{z}}}^\infty[d]p(t) + \boldsymbol{\delta}_{2,d}^{(t)}p(t), \text{ with } p(t) = \|\widehat{\mathbf{z}}^{(t)}\| \to 0. \tag{80}$$

Using the above representations, along with eq. (59), we have the following,

$$\widehat{\boldsymbol{\beta}}^{(t+1)}[d] = \widehat{\boldsymbol{\beta}}^{(t)}[d] + \eta_t \widehat{\mathbf{z}}^{(t)}[d]\left[|\widehat{\mathbf{w}}_1^{(t)}[d]|^2 + |\widehat{\mathbf{w}}_2^{(t)}[d]|^2 + \eta_t \widehat{\mathbf{z}}^{(t)}[d] \cdot \widehat{\boldsymbol{\beta}}^{(t)*}[d]\right]$$

$$\stackrel{(a)}{=} \widehat{\boldsymbol{\beta}}^{(t)}[d] + 2\eta_t p(t)|\widehat{\boldsymbol{\beta}}^{(t)}[d]|\left(\widehat{\widehat{\mathbf{z}}}^\infty[d] + \boldsymbol{\delta}_{2,d}^{(t)}\right)\left[1 + \boldsymbol{\delta}_{1,d}^{(t)} + \tfrac{1}{2}\eta_t \widehat{\mathbf{z}}^{(t)}[d]\mathrm{e}^{-\mathrm{i}\phi_{\widehat{\boldsymbol{\beta}}^{(t)}[d]}}\right]$$

$$\stackrel{(b)}{:=} \widehat{\boldsymbol{\beta}}^{(t)}[d] + 2\eta_t p(t)|\widehat{\boldsymbol{\beta}}^{(t)}[d]|\left[\widehat{\widehat{\mathbf{z}}}^\infty[d] + \boldsymbol{\delta}_{3,d}^{(t)}\right], \tag{81}$$

where $(a)$ follows from substituting eqs. (79)-(80), and $(b)$ follows from using $|\widehat{\mathbf{z}}^{(t)}[d]| \leq p(t) \to 0$ and defining $\boldsymbol{\delta}_{3,d}^{(t)} = \boldsymbol{\delta}_{2,d}^{(t)}\left[1 + \boldsymbol{\delta}_{1,d}^{(t)} + \tfrac{1}{2}\eta_t \widehat{\mathbf{z}}^{(t)}[d]\mathrm{e}^{-\mathrm{i}\phi_{\widehat{\boldsymbol{\beta}}^{(t)}[d]}}\right] + \widehat{\widehat{\mathbf{z}}}^\infty[d]\left[\boldsymbol{\delta}_{1,d}^{(t)} + \tfrac{1}{2}\eta_t \widehat{\mathbf{z}}^{(t)}[d]\mathrm{e}^{-\mathrm{i}\phi_{\widehat{\boldsymbol{\beta}}^{(t)}[d]}}\right] \to 0.$

Denote $\Delta_d = \phi_{\widehat{\widehat{\boldsymbol{\beta}}}^\infty[d]} - \phi_{\widehat{\widehat{\mathbf{z}}}^\infty[d]}$. Additionally, from the assumption in the theorem, we have $\mathrm{e}^{\mathrm{i}\phi_{\widehat{\boldsymbol{\beta}}^{(t)}[d]}} \to \mathrm{e}^{\mathrm{i}\phi_{\widehat{\widehat{\boldsymbol{\beta}}}^\infty[d]}}$, hence there exists $\boldsymbol{\delta}_{4,d}^{(t)} \to 0$ such that $\mathrm{e}^{\mathrm{i}\phi_{\widehat{\boldsymbol{\beta}}^{(t)}[d]}-\mathrm{i}\phi_{\widehat{\widehat{\mathbf{z}}}^\infty[d]}} = \mathrm{e}^{\mathrm{i}\Delta_d}(1 + \boldsymbol{\delta}_{4,d}^{(t)}).$

Now, from the above equation, for any $t_0$ and $t \geq t_0$, we derive the updates for $|\widehat{\boldsymbol{\beta}}^{(t)}[d]|$,

$$|\widehat{\boldsymbol{\beta}}^{(t+1)}[d]|^2 = |\widehat{\boldsymbol{\beta}}^{(t)}[d]|^2\left(\mathrm{e}^{\mathrm{i}\phi_{\widehat{\boldsymbol{\beta}}^{(t)}[d]}} + 2\eta_t p(t)\left[\widehat{\widehat{\mathbf{z}}}^\infty[d] + \boldsymbol{\delta}_{3,d}^{(t)}\right]\right)\left(\mathrm{e}^{-\mathrm{i}\phi_{\widehat{\boldsymbol{\beta}}^{(t)}[d]}} + 2\eta_t p(t)\left[\widehat{\widehat{\mathbf{z}}}^{\infty*}[d] + \boldsymbol{\delta}_{3,d}^{(t)*}\right]\right)$$

$$\stackrel{(a)}{=} |\widehat{\boldsymbol{\beta}}^{(t)}[d]|^2\left[1 + 2\eta_t p(t)\left(|\widehat{\widehat{\mathbf{z}}}^\infty[d]|\left(\mathrm{e}^{\mathrm{i}\Delta_d}(1 + \boldsymbol{\delta}_{4,d}^{(t)}) + \mathrm{e}^{-\mathrm{i}\Delta_d}(1 + \boldsymbol{\delta}_{4,d}^{(t)*})\right) + \boldsymbol{\delta}_{5,d}^{(t)}\right)\right]$$

$$\stackrel{(b)}{=} |\widehat{\boldsymbol{\beta}}^{(t)}[d]|^2\left[1 + 4\eta_t p(t)\left(|\widehat{\widehat{\mathbf{z}}}^\infty[d]|\cos(\Delta_d) + \boldsymbol{\delta}_{6,d}^{(t)}\right)\right]$$

$$\stackrel{(c)}{=} |\widehat{\boldsymbol{\beta}}^{(t_0)}[d]|^2\left[\prod_{u=t_0}^{t}\left(1 + 4\eta_u p(u)\left(|\widehat{\widehat{\mathbf{z}}}^\infty[d]|\cos(\Delta_d) + \boldsymbol{\delta}_{6,d}^{(u)}\right)\right)\right]$$

$$\stackrel{(d)}{\leq} |\widehat{\boldsymbol{\beta}}^{(t_0)}[d]|^2\exp\left(\sum_{u=t_0}^{t} 4\eta_u p(u)\left(|\widehat{\widehat{\mathbf{z}}}^\infty[d]|\cos(\Delta_d) + \boldsymbol{\delta}_{6,d}^{(u)}\right)\right), \tag{82}$$

where in $(a)$ we used $\mathrm{e}^{\mathrm{i}\phi_{\widehat{\boldsymbol{\beta}}^{(t)}[d]}-\mathrm{i}\phi_{\widehat{\widehat{\mathbf{z}}}^\infty[d]}} = \mathrm{e}^{\mathrm{i}\Delta_d}(1 + \boldsymbol{\delta}_{4,d}^{(t)})$ and collected all $o(p(t))$ terms into $\boldsymbol{\delta}_{5,d}^{(t)} = \tfrac{1}{2}\mathrm{e}^{\mathrm{i}\phi_{\widehat{\boldsymbol{\beta}}^{(t)}[d]}}\boldsymbol{\delta}_{3,d}^{(t)*} + 2p(t)\widehat{\widehat{\mathbf{z}}}^\infty[d]\left[\widehat{\widehat{\mathbf{z}}}^{\infty*}[d] + \boldsymbol{\delta}_{3,d}^{(t)*}\right] + \boldsymbol{\delta}_{3,d}^{(t)}\left(\mathrm{e}^{-\mathrm{i}\phi_{\widehat{\boldsymbol{\beta}}^{(t)}[d]}} + 2p(t)\left[\widehat{\widehat{\mathbf{z}}}^{\infty*}[d] + \boldsymbol{\delta}_{3,d}^{(t)*}\right]\right) \to 0$ (since $p(t), \boldsymbol{\delta}_{3,d}^{(t)} \to 0$); in $(b)$ we defined $\boldsymbol{\delta}_{6,d}^{(t)} = \tfrac{1}{2}\boldsymbol{\delta}_{4,d}^{(t)*}\mathrm{e}^{\mathrm{i}\Delta_d} + \tfrac{1}{2}\boldsymbol{\delta}_{3,d}^{(t)}\mathrm{e}^{-\mathrm{i}\Delta_d} + \boldsymbol{\delta}_{5,d}^{(t)} \to 0$; $(c)$ is obtained by iterating over $t$; and $(d)$ follows from using $(1 + x) \leq \exp(x)$.

If possible, let $\cos(\Delta_d) = -2\epsilon < 0$. Since $|\boldsymbol{\delta}_{6,d}^{(t)}| \to 0$, and for finite step sizes $\eta_t p(t) \to 0$, $\exists t_0$ such that for all $t \geq t_0$, $|\boldsymbol{\delta}_{6,d}^{(t)}| < \epsilon|\widehat{\widehat{\mathbf{z}}}^\infty[d]|$ and $\exp\left(-4\epsilon|\widehat{\widehat{\mathbf{z}}}^\infty[d]|\eta_t p(t)\right) \leq 1$. From eq. (82), we now have

$$|\widehat{\boldsymbol{\beta}}^{(t+1)}[d]|^2 \leq |\widehat{\boldsymbol{\beta}}^{(t_0)}[d]|^2\exp\left(-4\epsilon|\widehat{\widehat{\mathbf{z}}}^\infty[d]|\sum_{u=t_0}^{t}\eta_u p(u)\right) \leq |\widehat{\boldsymbol{\beta}}^{(t_0)}[d]|^2.$$

Finally, for any finite step sizes and finite $t_0$, we have $|\widehat{\beta}^{(t_0)}[d]|^2 < \infty$ and this creates a contradiction since the LHS in the above equation diverges, $|\widehat{\beta}^{(t+1)}[d]|^2 \to \infty$. Hence, in order for the updates in eq. (82) to lead to a divergent $|\widehat{\beta}^{(t+1)}[d]|$, we necessarily require that $\cos\left(e^{\mathrm{i}\Delta_d}\right) = \mathrm{Re}(e^{-\mathrm{i}\phi_{\widehat{\mathbf{z}}^\infty[d]} + \mathrm{i}\phi_{\widehat{\boldsymbol{\beta}}^\infty[d]}}) = 2\cos\left(\phi_{\widehat{\mathbf{z}}^\infty[d]} - \phi_{\widehat{\boldsymbol{\beta}}^\infty[d]}\right) \geq 0$.

This completes the proof of the lemma. $\qquad\square$

# D  Computing $\mathcal{R}_{\mathcal{P}}(\boldsymbol{\beta})$: Proofs of Lemmas in Section 5

In this appendix we prove the lemmas in Section 5 that compute the form of induced bias of linear networks in the space of predictors. Recall that for linear predictors parameterized as $\boldsymbol{\beta} = \mathcal{P}(\mathbf{w})$, $\mathcal{R}_{\mathcal{P}}(\boldsymbol{\beta}) = \min_{\mathbf{w}:\mathcal{P}(\mathbf{w})=\boldsymbol{\beta}} \|\mathbf{w}\|_2^2$.

**Lemma 5.** *For fully connected networks of any depth $L > 0$,*

$$\mathcal{R}_{\mathcal{P}_{full}}(\boldsymbol{\beta}) = \min_{\mathbf{w}:\mathcal{P}_{full}(\mathbf{w})=\boldsymbol{\beta}} \|\mathbf{w}\|_2^2 = L\|\boldsymbol{\beta}\|_2^{2/L} = monotone(\|\boldsymbol{\beta}\|_2).$$

*Proof.* Recall that for fully connected networks of any depth $L > 0$ with parameters $\mathbf{w} = [\mathbf{w}_l \in \mathbb{R}^{D_{l-1} \times D_l}]_{l-1}^L$, the equivalent linear predictor given by $\mathcal{P}_{full}(\mathbf{w}) = \mathbf{w}_1 \mathbf{w}_2 \ldots \mathbf{w}_L$.

We first show that $\mathcal{R}_{\mathcal{P}_{full}}(\boldsymbol{\beta}) \geq L\|\boldsymbol{\beta}\|_2^{2/L}$.
Let $\mathbf{w}^\star(\boldsymbol{\beta}) = [\mathbf{w}_l^\star(\boldsymbol{\beta})]_{l=1}^L$ be the minimizer of $\min_{\mathbf{w}:\mathcal{P}_{full}(\mathbf{w})=\boldsymbol{\beta}} \|\mathbf{w}\|_2^2$, so that $\boldsymbol{\beta} = \mathcal{P}_{full}(\mathbf{w}^\star(\boldsymbol{\beta})) = \mathbf{w}_1^\star(\boldsymbol{\beta}) \cdot \mathbf{w}_2^\star(\boldsymbol{\beta}) \ldots \mathbf{w}_L^\star(\boldsymbol{\beta})$ and $\mathcal{R}_{\mathcal{P}_{full}}(\boldsymbol{\beta}) = \|\mathbf{w}^\star(\boldsymbol{\beta})\|_2^2 = \sum_{l=1}^L \|\mathbf{w}_l^\star(\boldsymbol{\beta})\|_2^2$. We then have,

$$\|\boldsymbol{\beta}\|_2^{2/L} = \|\mathbf{w}_1^\star(\boldsymbol{\beta}) \cdot \mathbf{w}_2^\star(\boldsymbol{\beta}) \ldots \mathbf{w}_L^\star(\boldsymbol{\beta})\|_2^{2/L} \leq \|\mathbf{w}_1^\star(\boldsymbol{\beta})\|_2^{2/L} \|\mathbf{w}_2^\star(\boldsymbol{\beta})\|_2^{2/L} \ldots \|\mathbf{w}_L^\star(\boldsymbol{\beta})\|_2^{2/L}$$

$$\overset{(a)}{\leq} \frac{1}{L}\sum_{l=1}^L \|\mathbf{w}_l^\star(\boldsymbol{\beta})\|_2^2 = \frac{1}{L}\mathcal{R}_{\mathcal{P}_{full}}(\boldsymbol{\beta}), \tag{83}$$

where $(a)$ follows as arithmetic mean is greater than the geometric mean.

Next, we show that $\mathcal{R}_{\mathcal{P}_{full}}(\boldsymbol{\beta}) \leq L\|\boldsymbol{\beta}\|_2^{2/L}$.
Given any unit norm vectors $\mathbf{z}_l \in \mathbb{R}^{D_l}$ for $l = 1, 2, \ldots, L$, consider $\overline{\mathbf{w}} = [\overline{\mathbf{w}}_l]$, defined as

$$\overline{\mathbf{w}}_l = \begin{cases} \|\boldsymbol{\beta}\|_2^{1/L} \frac{\boldsymbol{\beta}}{\|\boldsymbol{\beta}\|_2} \mathbf{z}_1^\top & \text{if } l = 1 \\ \|\boldsymbol{\beta}\|_2^{1/L} \mathbf{z}_{l-1} \mathbf{z}_l^\top & \text{if } l = 2, 3, \ldots, L-1 \\ \|\boldsymbol{\beta}\|_2^{1/L} \mathbf{z}_{L-1} & \text{if } l = L \end{cases}$$

This ensures that $\mathcal{P}_{full}(\overline{\mathbf{w}}) = \overline{\mathbf{w}}_1 \overline{\mathbf{w}}_2 \ldots \overline{\mathbf{w}}_L = \boldsymbol{\beta}$ and $\|\overline{\mathbf{w}}\|_2^2 = L\|\boldsymbol{\beta}\|_2^{2/L}$, and hence

$$\mathcal{R}(\boldsymbol{\beta}) = \min_{\mathbf{w}:\mathcal{P}_{full}(\mathbf{w})=\boldsymbol{\beta}} \|\mathbf{w}\|_2^2 \leq \|\overline{\mathbf{w}}\|_2^2 = L\|\boldsymbol{\beta}\|_2^{2/L}. \tag{84}$$

Combining eq. (83) and eq. (84), we get $\mathcal{R}_{\mathcal{P}_{full}}(\boldsymbol{\beta}) = L\|\boldsymbol{\beta}\|_2^{2/L}$ $\qquad\square$

The proofs of the lemmas for computing $\mathcal{R}_{\mathcal{P}}(\mathbf{w})$ for diagonal and convolutional networks are similar to those of fully connected network.

**Lemma 6.** *For a depth–$L$ diagonal network with parameters $\mathbf{w} = [\mathbf{w}_l \in \mathbb{R}^D]_{l-1}^L$, we have*

$$\mathcal{R}_{\mathcal{P}_{diag}}(\boldsymbol{\beta}) = \min_{\mathbf{w}:\mathcal{P}_{diag}(\mathbf{w})=\boldsymbol{\beta}} \|\mathbf{w}\|_2^2 = L\|\boldsymbol{\beta}\|_{2/L}^{2/L} = monotone(\|\boldsymbol{\beta}\|_{2/L}).$$

*Proof.* Recall that for an $L$–layer linear diagonal networks with parameters $\mathbf{w} = [\mathbf{w}_l \in \mathbb{R}^D]_{l-1}^L$, the equivalent linear predictor is given by $\mathcal{P}_{diag}(\mathbf{w}) = \mathrm{diag}(\mathbf{w}_1)\mathrm{diag}(\mathbf{w}_2)\ldots\mathrm{diag}(\mathbf{w}_{L-1})\mathbf{w}_L$.

Let $\mathbf{w}^\star(\boldsymbol{\beta}) = [\mathbf{w}_l^\star(\boldsymbol{\beta})]_{l=1}^L$ be the minimizer of $\min_{\mathbf{w}:\mathcal{P}_{diag}(\mathbf{w})=\boldsymbol{\beta}}\|\mathbf{w}\|_2^2$, so that $\boldsymbol{\beta} = \mathcal{P}_{diag}(\mathbf{w}^\star(\boldsymbol{\beta}))$ and $\mathcal{R}_{\mathcal{P}_{diag}}(\boldsymbol{\beta}) = \|\mathbf{w}^\star(\boldsymbol{\beta})\|_2^2$. We then have,

$$\sum_{d=0}^{D-1}|\boldsymbol{\beta}[d]|^{2/L} = \sum_{d=0}^{D-1}\prod_{l=1}^L|\mathbf{w}_1^\star(\boldsymbol{\beta})[d]|^{2/L} \overset{(a)}{\leq} \frac{1}{L}\sum_{d=0}^{D-1}\sum_{l=1}^L|\mathbf{w}_1^\star(\boldsymbol{\beta})[d]|^2$$
$$= \frac{1}{L}\|\mathbf{w}^\star(\boldsymbol{\beta})\|_2^2 = \frac{1}{L}\mathcal{R}_{\mathcal{P}_{diag}}(\boldsymbol{\beta}), \tag{85}$$

where $(a)$ again follows as arithmetic mean is greater than the geometric mean.

Similar to the case of fully connected networks, we now choose $\overline{\mathbf{w}} = [\overline{\mathbf{w}}_l]$ that satisfies $\mathcal{P}_{diag}(\overline{\mathbf{w}}) = \boldsymbol{\beta}$ and $\|\overline{\mathbf{w}}\|_2^2 = L\|\boldsymbol{\beta}\|_{2/L}^{2/L}$. This would ensure that,

$$\mathcal{R}_{\mathcal{P}_{diag}}(\boldsymbol{\beta}) = \min_{\mathbf{w}:\mathcal{P}_{diag}(\mathbf{w})=\boldsymbol{\beta}}\|\mathbf{w}\|_2^2 \leq \|\overline{\mathbf{w}}\|_2^2 = L\|\boldsymbol{\beta}\|_{2/L}^{2/L}.$$

We can check that these properties are satisfied by choosing $\overline{\mathbf{w}}$ as follows: for $d = 0, 1, \ldots D-1$, let $\overline{\mathbf{w}}_1[d] = \text{sign}(\boldsymbol{\beta}^{(d)})|\boldsymbol{\beta}^{(d)}|^{1/L}$ and $\overline{\mathbf{w}}_l[d] = |\boldsymbol{\beta}^{(d)}|^{1/L}$ for $l = 2, 3, \ldots, L$.

Combining this argument with eq. 85 concludes the proof. $\qquad\square$

For convolutional networks, the argument is the exactly the same as that for diagonal network adapted for complex vectors.

**Lemma 7.** *For a depth–$L$ convolutional network with parameters $\mathbf{w} = [\mathbf{w}_l \in \mathbb{R}^D]_{l-1}^L$, we have*

$$\mathcal{R}_{\mathcal{P}_{conv}}(\boldsymbol{\beta}) = \min_{\mathbf{w}:\mathcal{P}_{conv}(\mathbf{w})=\boldsymbol{\beta}}\|\mathbf{w}\|_2^2 = L\|\widehat{\boldsymbol{\beta}}\|_{2/L}^{2/L} = \textit{monotone}(\|\widehat{\boldsymbol{\beta}}\|_{2/L}).$$

*Proof.* Denote the Fourier basis coefficients of $\mathbf{w}_l \in \mathbb{R}^D$ and $\boldsymbol{\beta} = \mathcal{P}_{conv}(\mathbf{w}) \in \mathbb{R}^D$ in polar form as

$$\widehat{\mathbf{w}}_l = |\widehat{\mathbf{w}}_l|e^{\mathrm{i}\phi_{\widehat{\mathbf{w}}_l}} \in \mathbb{C}^D, \quad \widehat{\boldsymbol{\beta}} = |\widehat{\boldsymbol{\beta}}|e^{\mathrm{i}\phi_{\widehat{\boldsymbol{\beta}}}} \in \mathbb{C}^D,$$

where $|\widehat{\mathbf{w}}_l|, |\widehat{\boldsymbol{\beta}}| \in \mathbb{R}_+^D$ and $\phi_{\widehat{\mathbf{w}}_l}, \phi_{\widehat{\boldsymbol{\beta}}} \in [0, 2\pi)^D$ are the vectors with magnitudes and phases, respectively, of $\widehat{\mathbf{w}}_l, \widehat{\boldsymbol{\beta}}$.

From Lemma 3, the Fourier basis representation of $\boldsymbol{\beta} = \mathcal{P}_{conv}(\mathbf{w})$ is given by

$$\widehat{\boldsymbol{\beta}} = \text{diag}(\widehat{\mathbf{w}}_1)\text{diag}(\widehat{\mathbf{w}}_2)\ldots\text{diag}(\widehat{\mathbf{w}}_{L-1})\widehat{\mathbf{w}}_L = \mathcal{P}_{diag}(\widehat{\mathbf{w}}),$$

where we have overloaded the notation $\mathcal{P}_{diag}$ to denote the mapping of diagonal networks in complex vector fields, and $\widehat{\mathbf{w}} = [\widehat{\mathbf{w}}_l]_{l=1}^L$. We thus have for $d = 0, 1, \ldots, D-1$,

$$|\widehat{\boldsymbol{\beta}}[d]| = \prod_{l=1}^L|\widehat{\mathbf{w}}_l[d]|, \quad \text{and} \quad \phi_{\widehat{\boldsymbol{\beta}}}[d] = \left(\sum_{l=1}^L\phi_{\widehat{\mathbf{w}}_l}[d]\right) \bmod 2\pi.$$

From orthonormality of discrete Fourier transformation, we have for all $\mathbf{w}$, $\|\mathbf{w}\|_2^2 = \|\widehat{\mathbf{w}}\|_2^2$. Thus,

$$\mathcal{R}_{\mathcal{P}_{conv}}(\boldsymbol{\beta}) = \min_{\mathbf{w}:\mathcal{P}_{conv}(\mathbf{w})=\boldsymbol{\beta}}\|\mathbf{w}\|_2^2 = \min_{\widehat{\boldsymbol{\beta}}:\widehat{\boldsymbol{\beta}}=\mathcal{P}_{\text{diag}}(\widehat{\mathbf{w}})}\|\widehat{\mathbf{w}}\|_2^2. \tag{86}$$

We can now adapt the proof of diagonal networks here. Let $\widehat{\mathbf{w}}^\star(\boldsymbol{\beta}) = [\widehat{\mathbf{w}}_l^\star(\boldsymbol{\beta}) \in \mathbf{C}^D]_{l=1}^L$ be the minimizer of $\min_{\widehat{\mathbf{w}}:\widehat{\boldsymbol{\beta}}=\mathcal{P}_{\text{diag}}(\widehat{\mathbf{w}})}\|\widehat{\mathbf{w}}\|_2^2$, so that $\widehat{\boldsymbol{\beta}} = \mathcal{P}_{diag}(\widehat{\mathbf{w}}^\star(\boldsymbol{\beta}))$ and $\mathcal{R}_{\mathcal{P}_{conv}}(\boldsymbol{\beta}) = \|\widehat{\mathbf{w}}^\star(\boldsymbol{\beta})\|_2^2$, and

$$\sum_{d=0}^{D-1}|\widehat{\boldsymbol{\beta}}[d]|^{2/L} = \sum_d\prod_{l=1}^L|\widehat{\mathbf{w}}_1^\star(\boldsymbol{\beta})[d]|^{2/L} \leq \frac{1}{L}\sum_d\sum_{l=1}^L|\widehat{\mathbf{w}}_1^\star(\boldsymbol{\beta})[d]|^2$$
$$= \frac{\|\widehat{\mathbf{w}}^\star(\boldsymbol{\beta})\|_2^2}{L} = \frac{1}{L}\mathcal{R}_{\mathcal{P}_{conv}}(\boldsymbol{\beta}). \tag{87}$$

Similar to the diagonal networks, we can choose the parameters in the Fourier domain $\widehat{\overline{\mathbf{w}}} = [\widehat{\overline{\mathbf{w}}}_l \in \mathbb{C}^D]$ to ensure that $\mathcal{P}_{diag}(\widehat{\overline{\mathbf{w}}}) = \widehat{\boldsymbol{\beta}}$ and $\|\widehat{\overline{\mathbf{w}}}\|_2^2 = L\|\widehat{\boldsymbol{\beta}}\|_{2/L}^{2/L}$ as follows: for $d = 0, 1, \ldots D - 1$, let

$$\widehat{\overline{\mathbf{w}}}_1[d] = \phi_{\widehat{\boldsymbol{\beta}}}[d] \, |\widehat{\boldsymbol{\beta}}[d]|^{1/L} \text{ and } \widehat{\overline{\mathbf{w}}}_l[d] = |\widehat{\boldsymbol{\beta}}[d]|^{1/L}, \forall l > 1.$$

This gives us

$$\mathcal{R}_{\mathcal{P}_{conv}}(\boldsymbol{\beta}) = \min_{\mathbf{w}: \mathcal{P}_{diag}(\widehat{\mathbf{w}}) = \widehat{\boldsymbol{\beta}}} \|\widehat{\mathbf{w}}\|_2^2 \leq \|\widehat{\overline{\mathbf{w}}}\|_2^2 \leq L\|\widehat{\boldsymbol{\beta}}\|_{2/L}^{2/L}.$$

Combining this with eq. 87 concludes the proof. $\qquad\square$

## E    Background Results

**Theorem 11** (Stolz–Cesaro theorem, proof in Theorem 1.22 of Muresan [2009]). *Assume that $\{a_k\}_{k=1}^\infty$ and $\{b_k\}_{k=1}^\infty$ are two sequences of real numbers such that $\{b_k\}_{k=1}^\infty$ is strictly monotonic and diverging (i.e., monotone increasing with $b_k \to \infty$, or monotone decreasing with $b_k \to -\infty$). Additionally, if $\lim_{k\to\infty} \frac{a_{k+1}-a_k}{b_{k+1}-b_k} = L$ exists, then $\lim_{k\to\infty} \frac{a_k}{b_k}$ exists and is equal to L.*