[Reviews · NeurIPS 2018]

Reviewer 1



The paper considers the problem of formalizing the implicit bias of gradient descent on fully connected linear/convolutional networks with an exponential loss. Building on the recent work by Soudry et al. which considered a one layer neural network with no activation the paper generalizes the analysis to networks with greater depth (with no activations) and the exponential loss. The two main networks considered by the authors and the corresponding results are as follows. Linear Fully Connected Networks - In this setting the authors show that gradient descent in the limit converges to a predictor which in direction is the max margin predictor. This behaviour is the same as what was established in the earlier paper of Soudry et al for one layer neural networks. In particular the paper shows that depth has no effect in this setting on the implicit bias of gradient descent. Convolutional Network - The paper further considers the case of convolutional networks and works in the model where every layer has the same width as the input and the convolutional filter has width equal to that of the layer. Note that they work with the one dimensional convolutional filter. For this model the authors show that the solution converges in direction to a critical point of the problem - minimize the 2/L norm of the fourier coefficients of the predictor given the predictor classifies perfectly. In particular this says that as the depth of the network increases the classifier learned is encouraged to be sparse in the fourier domain. Characterizing implicit bias of gradient descent has attracted a lot of recent attention as it seems to be the source for the auto regularization effect observed in neural net training. While the work of Soudry et al was the first step in this direction, this paper makes the above two very interesting generalizations of that work. In particular the characterization obtained for linear convolutional networks is illuminating and could be of great significance breeding further research. The paper achieves the result by first showing that for any network which implements a homogenous polynomial in terms of its parameters, we have that the solution converges in direction to a perfect classifier with the smallest l_2 norm. Note that this result is in the parameter space. The two results then follow by understanding the minimum l2 norm solution in terms of the predictor it corresponds to. The interesting thing here is that if the network is assumed to be diagonal then the depth plays a role in the implicit bias. The result for convolutional case follows by noting that colvolutional networks correspond to diagonal networks in the setup considered by the authors. The paper is well written (modulo a couple of typos), considers an interesting problem and provides an illuminating result. The discussion section of the paper is strong with multiple future directions. I think the paper would be a great addition to the program. Typos - I believe there are typos in the definition of fourier coefficients (line 52 and line 140) and the definition of the convolution line 70.

Reviewer 2



In this paper, the authors study the implicit bias induced by gradient descent on linear fully connected neural network and linear convolutional neural network. The authors show that convolutional parametrization of the NN has a radically different biased when trained with gradient descent, even if the fully connected linear NN and the linear convolutional NN defines the exact model class. Even though the linear convolutional NN are not explicitly regularised their solutions are biased to have sparsity in the frequency domain (Theorem 2), and this bias change with respect to the depth of the network (Theorem 2a). The paper is well written, clear and to the point. Although it is clear that this work allows us to better understand the implicit “regularisation” role of optimization on some specific form NN, because the subject of this paper is not close to my personal research area, it is hard for me to evaluate the Significance of this work.

Reviewer 3



The paper under review proves implicit regularization properties of gradient descent applied to multi-layer linear networks. The main results establish that the gradient descent algorithm applied to a multilayer linear network converges towards a solution which is characterized by the structure of the network. I think that the results are timely and interesting for the NIPS community. The presentation of the paper needs to be improved. There are several typos and many sequences which are not clear. Moreover, some parts of the proofs and the theorems are not sufficiently detailed in my opinion. Some examples: line 374: the following assumption is made in almost every statement: "the incremental updates w(t+1) − w(t) converges in direction", but the authors never discuss when this property holds in practice. It would be interesting to know whether there are e.g. choices of the stepsizes ensuring that this assumption holds. line 378: \Delta w_t is not defined line 383: why lim t→\infty of \gamma^\nu y_n /||w^(t)||^\nu > 0? line 409: in order to understand equation (17), in my opinion it would be helpful to anticipate the formula after line 409 before (17)